# Single-cell and spatially resolved interactomics of tooth-associated keratinocytes in periodontitis

Quinn T. Easter [1], Bruno Fernandes Matuck[1], Germán Beldorati Stark [2], Catherine L. Worth[3], Alexander V. Predeus [4], Brayon Fremin[3], Khoa Huynh[5], Vaishnavi Ranganathan[6], Zhi Ren [7], Diana Pereira [8], Brittany T. Rupp[1], Theresa Weaver[1], Kathryn Miller[9], Paola Perez [10], Akira Hasuike [1,11], Zhaoxu Chen [12], Mandy Bush[13], Xufeng Qu[14], Janice Lee [15], Scott H. Randell [16], Shannon M. Wallet[17,18], Inês Sequeira [8], Hyun Koo[7], Katarzyna M. Tyc[5,14], Jinze Liu[5,14], Kang I. Ko[12], Sarah A. Teichmann [4,19] & Kevin M. Byrd [1,10,17] ✉

Periodontitis affects billions of people worldwide. To address relationships of periodontal niche cell types and microbes in periodontitis, we generated an integrated single-cell RNA sequencing (scRNAseq) atlas of human periodontium (34-sample, 105918-cell), including sulcular and junctional keratinocytes (SK/JKs). SK/JKs displayed altered differentiation states and were enriched for effector cytokines in periodontitis. Single-cell metagenomics revealed 37 bacterial species with cell-specific tropism. Fluorescence in situ hybridization detected intracellular *16 S* and mRNA signals of multiple species and correlated with SK/JK proinflammatory phenotypes in situ. Cell-cell communication analysis predicted keratinocyte-specific innate and adaptive immune interactions. Highly multiplexed immunofluorescence (33-antibody) revealed peri-epithelial immune foci, with innate cells often spatially constrained around JKs. Spatial phenotyping revealed immunosuppressed JK-microniches and SK-localized tertiary lymphoid structures in periodontitis. Here, we demonstrate impacts on and predicted interactomics of SK and JK cells in health and periodontitis, which requires further investigation to support precision periodontal interventions in states of chronic inflammation.

Periodontal diseases—i.e., periodontitis—affect billions of people every year across the globe[1]. They are most often characterized by dysregulated, chronic inflammation of the periodontium, typically caused by polybacterial dysbiosis, though other systemic conditions have been associated with periodontitis symptomology[2]. If left untreated, the result of chronic periodontitis is tooth loss[3]. Periodontal diseases are associated with >60 systemic diseases, including cardiovascular diseases, type 2 diabetes, Alzheimer's disease, and inflammatory bowel diseases[4]. Precision medicine approaches for these diseases—including

diagnostics, prognostics, and biologics—have had minimal success to date; however, early identification and treatment may improve oral and overall health[5]. A limited understanding of cell subpopulations and their cell states, either supporting niche maintenance or contributing to its breakdown, inhibits precision approaches in periodontitis, and there is an unmet need to elucidate cell-specific and cooperative cell plasticity in periopathogenesis. Despite decades of data supporting tooth-associated keratinocyte heterogeneity[6], functional cell annotation remains incomplete.

Structural immunity[7] i.e., "immune functions of non-hematopoietic cells", can be studied using single-cell and spatial genomic approaches, and recent work in this field suggests that in certain contexts, "every cell is an immune cell"[8]. In periodontitis, this concept has been demonstrated between fibroblasts and innate immune cells[9]. Other structural cell types such as epithelial cells—specifically keratinocytes—have been implicated in regulating autoimmune and infectious disease immune responses[10,11]. Furthermore, while epithelial cells and/or keratinocytes (KCs) may passively permit microbial infection via tissue barrier breakdown[12], keratinocytes via their stem/progenitor cells at the stromal tissue interface may play important immune education roles before, during, and after barrier breakdown; alternatively, keratinocyte stem/progenitor cell rewiring through immune-like memory of past immune and microial interactions may sensitize these cells to proinflammatory profiles with or without chronic challenge[13].

To address whether and how tooth-associated epithelial cell types may play active structural immune roles in periodontitis, we created an integrated periodontitis atlas of human tissues[9,14–16] using the open-source single-cell analysis toolkit Cellenics hosted by Parse Biosciences (https://scp.biomage.net; see Methods), describing 17 total and functionally annotating 5 new gingival keratinocyte subpopulations within the human gingival epithelium at a single-cell level. Tooth-facing sulcular keratinocytes (SKs) and tooth-interfacing junctional keratinocytes (JKs) were classified into their stem/progenitor cells and their differentiated progeny using *KRT19* and other niche-specific marker expression. We made this data publicly available at CZI CELLxGENE and validated these cell identities using multiplexed fluorescence in situ hybridization (mISH). Transcriptionally, SK and JK microniches respond differently in disease but generally alter cell differentiation and upregulate effector cytokines ("epi-kines/keratokines") along stem-cell-to-progeny trajectories.

Since keratinocytes function at the interface of microbial challenge and related immune effects, we combined unmapped reads of our integrated atlas and found that SKs and JKs harbor numerous bacterial reads from known periodontitis-associated pathogens ("periopathogens"). mISH in vivo pinpointed cell-specific signatures of highly positive *16S*+ cells, revealing regional SK/JKs phenotypes mostly associated with *16S* signal within tissue microniches. To understand the effects of this phenotype, we used cell-cell communication analysis to predict these SK/JKs preferentially regulate innate and adaptive immune cell subpopulations, even in health. Using highly multiplexed immunofluorescence assays (mIF; 33-antibody), we found adaptive and innate immune cell compartmentalization between SK and JK niches, including potential immune suppression phenotypes at areas closest to the tooth surface where gram-negative bacteria aggregate into immunostimulatory biofilms in disease—i.e., the peri-JK microniche. The JK-microniche was most often PD-L1+ in contrast to SK-localized tertiary lymphoid structures, which more often contained mixed active (ICOS+) and exhausted (PD-1+) populations in periodontitis. Here, we demonstrate impacts on and predicted interactomics of SK and JK cells in health and periodontitis, suggesting their linked barrier integrity maintenance, immunostimulatory microbial interaction endurance, and immune cell tropism coordination roles require deeper investigation to support precision periodontal interventions in states of chronic periodontal inflammation.

## Results

### Generation and analysis of a first draft integrated periodontitis atlas

The tooth is supported by diverse cell types (Fig. 1a). Hundreds of diseases affect teeth; however, the cell-specific contribution to these diseases remains limitedly explored[17]. Though not exhaustively explored here, recent murine[18–20] and human studies have focused on the single-cell RNA sequencing (scRNAseq) of the tooth-supporting periodontium (mineralized tissues: alveolar bone, cementum; soft tissues: gingiva, periodontal ligament)[21], but some cell types like keratinocytes are minimally annotated despite their known heterogeneity[22]. We analyzed 4 human scRNAseq datasets[9,14–16] (34 samples, 3 states [20 health, 4 gingivitis, 10 periodontitis]; Fig. 1b) using Cellenics®. Metadata was harmonized (Supplementary Data 1)[23] and the location noted for 27 of the 34 samples to establish a common coordinate framework (CCF) for these and future periodontium studies[24] (Supplementary Fig. 1a). Some harvest sites were unknown (~20%); others were taken from broader characterizations than a single tooth site (i.e., "anterior", meaning incisors and canines or "posterior", meaning premolar and molars).

All scRNAseq samples were reprocessed, filtered, and integrated (Fig. 1b; see Methods). Cells were broadly annotated at Tier 1 resolution (epithelial, stromal, endothelial, neural, and immune). We focused on gingival epithelial heterogeneity (Fig. 1c) within the distinct transitional zone between non-keratinized alveolar mucosal (AM, if present), attached gingival (AG), gingival margin (GM), and sulcular and junctional keratinocytes (SK/JKs)[22]. Red blood cells (*HBA*+/*HBB*+) and cycling cells were filtered out, and each study was further annotated (Tier 2; Fig. 1d). Integrating data enabled the harmonized cell annotation of 32 cell types across datasets (Tier 3; Fig. 1e, f); marker genes were determined for each (Fig. 1g; Supplementary Data 1). All keratinocytes were broadly marked by *KRT5/KRT14*. SK/JKs also expressed higher *FDCSP*, *ODAM*, and keratins *KRT7* and *KRT19*[6]. At this level of resolution, epithelial cells could be classified into 7 different types, including SK/JKs and other epithelial-resident cells i.e., Merkel cells, melanocytes, and Langerhans cells. While no single study contained all the cell types annotated in this atlas, these four studies were similar when comparing relative cell proportions (Supplementary Fig. 1b). Cellenics® data was exported to CELLxGENE for public use (https://cellxgene.cziscience.com/collections/71f4bccf-53d4-4c12-9e80-e73bfb89e398; Supplementary Fig. 1c).

### Transcriptomic analysis reveals immune roles for and effects on keratinocytes in periodontitis

Recent work supports the idea that "structural" cells (i.e., neural, adipose, muscle, vasculature, fibroblasts, epithelial cells) can support the tailoring of individual immune responses to niche-specific challenges ("structural immunity"[7]). In periodontitis, this has been relatively unexplored. Individually, fibroblasts in single-cell[9] and spatial transcriptomic[25] studies have shown direct innate immune curation roles via effector cytokines and other ligands predicted to interact with innate and adptive immune cell populations. To ask how structural cell types may play roles in periodontitis progression, we analyzed periodontitis versus healthy cells in pseudobulk RNAseq analyses, generating differentially expressed gene (DEG) lists (Fig. 1h; Supplementary Fig. 1d–f), Supplementary Data 1) for all cells and specifically keratinocytes, fibroblasts, and vasculature endothelial cells. We observed structural populations comprising ~2/3 of the up- and downregulated DEGs from the "all-cell" pseudobulk experiment, supporting potential immune roles in periodontitis for multiple structural cell types.

The immunoregulatory role of keratinocytes in periodontitis has been specifically studied[12,26] and the innate immune population residency near JKs previously shown[27]. In diseased keratinocytes, our DEG analysis revealed upregulation of *CXCL1, CXCL3, CXCL8, CXCL13, CCL20, CSF3, IL1A, IL1B,* and *IL36G* and receptors *IL1R1* and *IL7R* in periodontitis. Furthermore, *CXCL1, CXCL8, IL1A,* and *IL1B* had a greater log fold change in keratinocytes compared to all cells (Fig. 1h), and *CXCL17, CCL20,* and *CSF3* were upregulated in keratinocytes (compared to Supplementary Fig. 1) and also shared with primarily fibroblasts and/or endothelial cells. We used g:Profiler (see Methods) to understand altered disease state pathways in keratinocytes, and using functional profiling of GO pathways, cell signaling and bacteria/dysbiosis responses were highlighted as the most upregulated (Supplementary

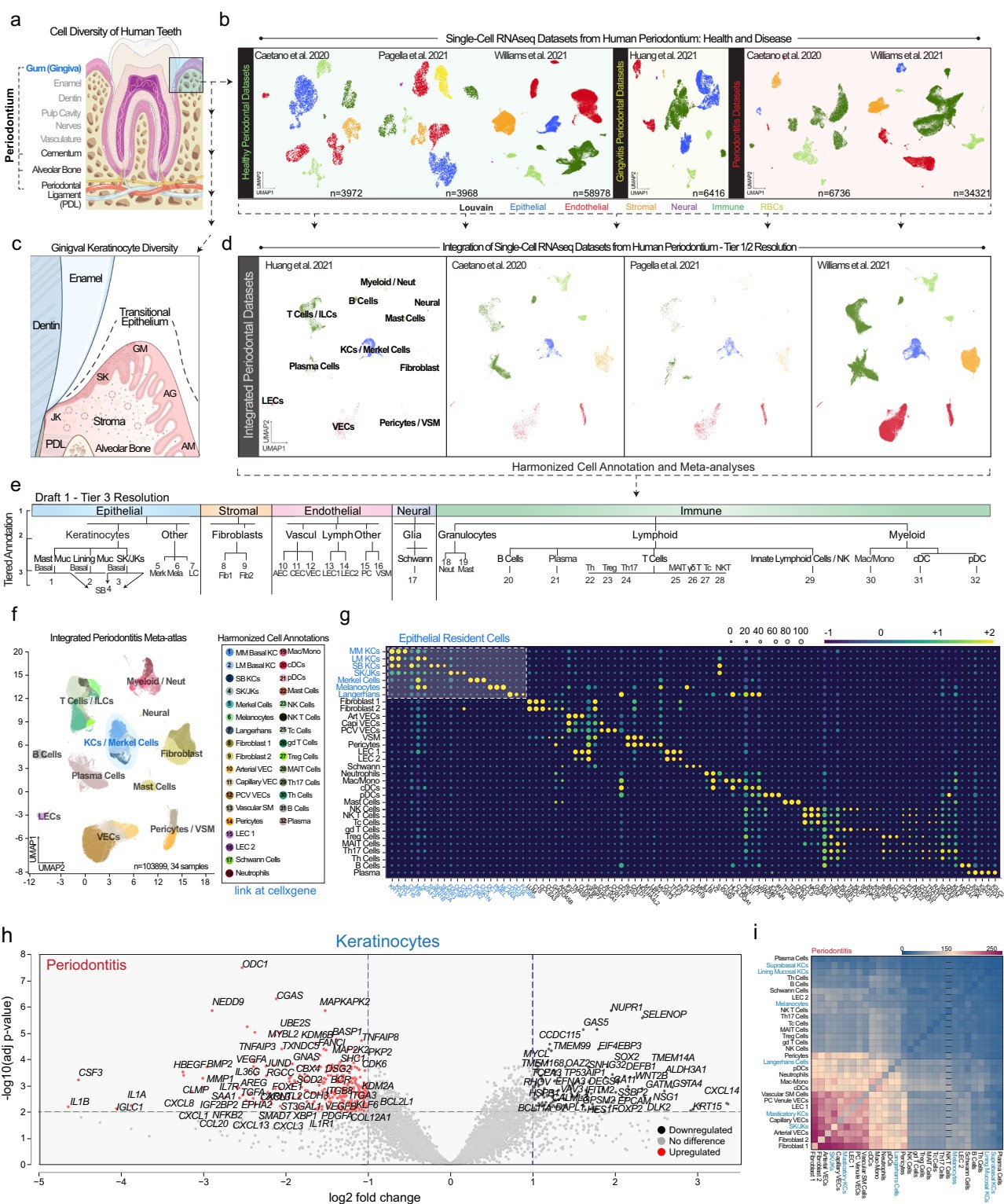

Fig. 2a). Immune response upregulation also appeared to come at the expense of terminal differentiation, development, and translation activity (Supplementary Fig. 2b). This phenomenon may help reshape the periodontal niche through immune recruitment, either individually or in concert with other epithelial and structural cell types of the periodontal niche.

Using the DEG data, the "reactome" of diseased keratinocytes further suggested active immune roles for these cells (Supplementary Fig. 2c). To understand this comprehensively, we used CellPhoneDB (see Methods) to investigate relative cell-cell communication patterns

of Tier 3-level cell identities via receptor-ligand pairs (Fig. 1i; Supplementary Data 2). We found expected fibroblast-vasculature and fibroblast-immune interactions. As predicted, we also discovered many interactions between SK/JKs and many other cell types, suggesting regional epithelia-stromal-immune communication axes in periodontitis. While some other keratinocytes are predicted to signal to other cell types in disease states, this occurs prominently for SK/JKs compared to other keratinocytes, melanocytes, and Merkel cells. Overall, this focused our investigation on understanding these tooth-associated keratinocytes in health and disease states.

**Fig. 1 | An integrated periodontitis atlas reveals important oral keratinocyte population roles in immune signaling. a** Specialized tissues support human teeth, including the periodontium, consisting of 1) gingiva (blue/box: epithelia; stroma), 2) periodontal ligament, and 3) mineralized tissues (cementum; alveolar bone). **b** Four single-cell RNA sequencing (scRNAseq) datasets were reprocessed for broad cell class comparison between studies. **c** The gingival epithelial attachment is a specialized transitional epithelium example, changing from non-keratinized alveolar mucosa (AM; if present), to keratinized attached gingiva (AG), altering expression profiles at the gingival margin (GM), then specializing in gingival sulcus and junctional epithelial keratinocytes (SK/JKs). **d** Each study was first integrated using Harmony and assigned Tier 1 cell type annotations. **e** Harmonized tier annotation was performed between epithelial, stromal, endothelial, neural, and immune cell populations. **f** Integrated UMAP and cell assignments and **g** cell signatures (Supplementary Data 1) were generated. Epithelial cells (blue) are highlighted. The entire dataset was uploaded to publicly-available CELLxGENE (cellxgene.cziscience.com/). SKs and JKs were grouped in the Tier 3 analysis as co-expressing Keratin 14 (*KRT14*) and Keratin 19 (*KRT19*). **h** Pseudobulk analysis of some differentially expressed genes (DEGs) in periodontitis using all keratinocytes

(Tier 3 annotations) in volcano plots; full list, Supplementary Data 1. **i** Using Cell-PhoneDB, all Tier 3 cell types were analyzed for inferred receptor-ligand interactions; most frequent–bottom left. SK/JKs appear uniquely expressive of effector cytokines/other ligands compared to other keratinocytes (Supplementary Data 2). Abbreviations: ILCs Innate Lymphoid Cells, KCs Keratinocytes, VECs Vascular Endothelial Cells, VSM Vascular Smooth Muscle, LECs Lymphatic Endothelial Cells, Neut Neutrophils, Mast Muc, MM Masticatory (Keratinized) Mucosa, Lining Muc, LM Lining (Non-Keratinized) Mucosa, SB Suprabasal (Differentiated) Keratinocytes, Fib Fibroblast, AECs Arterial Endothelial Cells, PCV Postcapillary Venule, VECs Venule Endothelial Cells, Mac/Mono Macrophage/Monocytes, cDCs Conventional Dendritic Cells, pDC Plasmacytoid Dendritic Cells, Tc Cytotoxic T Cells, gdT Gamma Delta T Cells, Treg Regulatory T Cells, MAIT Mucosal Associated Invariant T Cells, Th Helper T Cells. Illustration from (**a**) created with BioIcons (image hosted at https://bioicons.com; tooth icon by Servier https://smart.servier.com/, licensed under CC-BY 3.0 Unported https://creativecommons.org/licenses/by/3.0/); illustration from (**c**) created with BioRender (https://www.biorender.com/). *n* = 34-sample, 105918-cells.

## Redefining human gingival keratinocyte subpopulations for niche-specific analysis in periodontitis

To validate the KRT19/*KRT19* spatial localization, adult gingivae were harvested, and the orientation was preserved to feature both oral-facing and tooth-facing keratinocytes (Fig. 2a–c). Immunofluorescence validated KRT19 as the definitive SK/JK marker (Fig. 2b, c). Each of these regions within the entire gingiva revealed similar proportions of Ki67+ cycling cells, highlighting the need to understand SK/JK epithelial stem/progenitor cells in humans as previously done with mice[28] (Fig. 2c). We subclustered keratinocytes from our integrated atlas (~8500 cells, Fig. 1) and identified that *KRT19*-high expressing cells clustered together (Fig. 2d, Supplementary Fig. 3a). We generated marker lists for each population (Fig. 2e; Supplementary Data 1). Using a custom 12-plex in situ hybridization (ISH) assay (RNAscope) designed from single-cell signatures with built-in negative/low controls, we found *CXCL14* expression in keratinized basal cells (AG) in contrast to no expression in SK/JK *KRT19*+ cells (Fig. 2f; Supplementary Fig. 3b). Other markers enriched in oral-facing keratinocytes included *NPPC*, *PAPPA*, and *NEAT1* but *SAA1/2*, *IL18*, and *RHCG* in tooth-facing SK/JKs. Using primary human gingival keratinocytes (HGKs), we discovered the persistence of KRT14+ and KRT14+/KRT19+ cells that could enter the cell cycle (Ki67+; Fig. 2h). This mixed primary culture model was maintained at passage (P) 5, including differentiated progeny from both stem/progenitor cell types (Fig. 2i). Using ISH, we also found cell subpopulation-specific markers such as SK/JKs (*SOX6*+/*FDCSP*+) and oral-facing AG keratinocytes (*SOX6*+/*LGR6*+; Fig. 2j). This established multiple lines of evidence for distinct SK/JK epithelial stem cells and their progeny in vivo as well as the potential for new in vitro models to understand their cell subpopulation-specific activity and disease and/or stress responses.

## Murine gingival Krt19 subpopulations exist around molars but are less common

Despite the frequent use of mouse models of periodontitis[29], the junctional niche similarity between mice and humans is debated[30,31]. We were curious whether these SK/JK cells and their signatures were conserved between human and mouse. To address this, we performed scRNAseq of adult mouse gingiva (Supplementary Fig. 4a, b). Subclustering epithelial cells revealed a subpopulation of *Krt19*+ keratinocytes. When compared to basal epithelial stem/progenitors (*Krt14*+) and differentiated (*Krt1*+, *Krt4*+) keratinocytes, *Krt19*+ cells were mostly *Krt14*-positive (Supplementary Fig. 4b). ISH analysis of healthy mouse gingiva around molar (M) 1 and M2 readily showed pockets of *Krt19* expression, suggesting some conservation (Supplementary Fig. 4c). These rare *Krt19* cells also upregulated *Cxcl1*, *Cxcl8*, and *Cxcl17*, distinct from other *Krt14*-high cells (Population 7, yellow box: Supplementary

Fig. 4d), suggesting some mirroring of human cell-cell communication activity between SK/JKs and other cell types in mice. Some other mixed cell subpopulations expressed some interleukins (Population 6, pink box: *Il1a*, *Il1b*, *Il1rn*, *Il18*). When examining the healthy keratinocytes, we observed similar expression signatures in SK/JK cells. When using the same human signatures from Fig. 2, mouse subpopulations failed to cleanly segregate using human signatures, supporting only some heterogeneity of murine gingival keratinocytes. Overall, this data suggested that further investigation of cross-species cell type analyses is needed for mouse periodontium between other animal models and humans[32].

## Periodontitis affects SK/JK stem/progenitor differentiation to upregulate inflammatory signatures

Confirmation of SK/JK cell identities led us to refine cell annotations using new markers (Fig. 3a) and consider how SK/JKs are uniquely affected in periodontitis. To better understand SK/JK heterogeneity, we subclustered out the *KRT19*-high cells from the integrated atlas, annotating subpopulations to include basal (stem/progenitors) and their differentiated progeny (suprabasal [SB] keratinocytes; Fig. 3b, c). We next looked at the cell-specific gene upregulation patterns to gain insight into these cells in periodontitis (Supplementary Data 1). Despite their adjacency, we found 28.5% of shared SK/JK gene upregulation in periodontitis (Fig. 3d). We found basal and SB JKs—*infrequently* basal and SB SKs—generally upregulated effector cytokines *CXCL1*, *CXCL3*, *CXCL6*, *CXCL8*, *IL1A*, *IL1B*, and *IL36G* (*CXCL1* and *IL1B* by SB JKs compared to SB SKs (Fig. 3e)). This finding uncovered potential therapeutic avenues for periodontitis niche restoration to health, since keratinocytes closest to the tooth surface drove most effector cytokine expression compared to all keratinocytes.

Next, we subclustered basal and SB SKs and JKs and found that these distinct cell identities formed unique clusters using partition-based graph abstraction (PAGA; see Methods and Supplementary Fig. 5a–e). We sought to understand how gene expression pattern changes revealed the differentiation and proinflammatory changes in SK/JKs. Along the differentiation trajectory in pseudotime, we predicted altered differentiation trajectories i.e., *JUND* increased and *SPRR3* decreased in both JKs and SKs yet *TGM* decreased in JKs and *SPINK7* decreased in SKs (Fig. 3f, g). The largest difference occurred in cell signaling changes between JKs and SKs (Fig. 3h, i). *CXCL1*, *CXCL3*, *CXCL8*, *CSF3*, *IL1A*, *IL1B*, and *IL1RN* all increased in JKs in periodontitis as predicted by our atlas/DEG analyses; alternatively, SKs were relatively less active compared to JKs. Overall, SK/JK stem cells and their progeny appeared to differently upregulate proinflammatory, immunoregulatory, and innate immune chemotactic signatures at the expense of healthy cell differentiation and epithelial barrier maintenance.

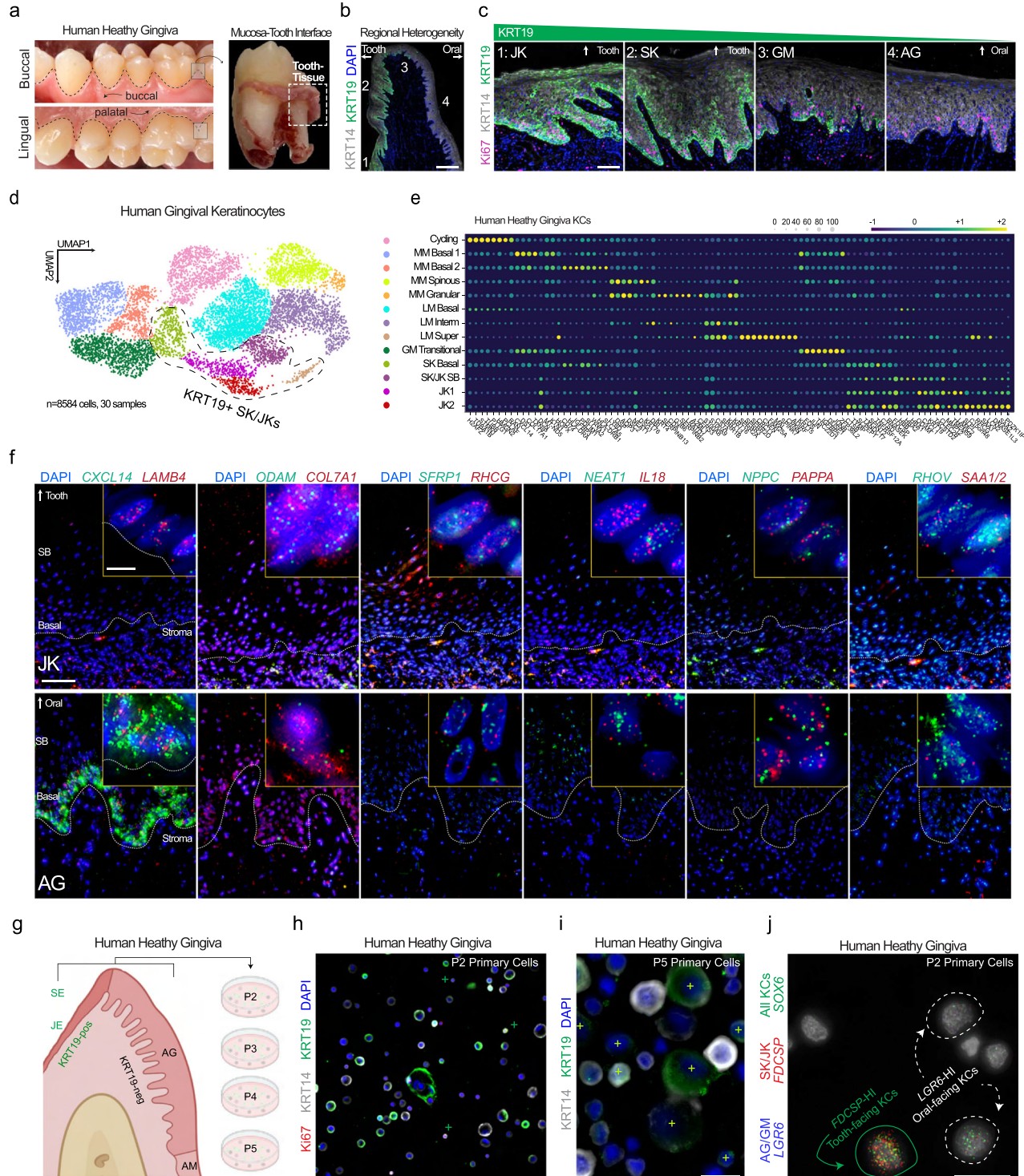

## Polybacterial interactions of human keratinocytes are diverse and frequent in periodontitis

Considering polymicrobial dysbiosis drives periodontitis in susceptible hosts[33], we hypothesized these SK/JK phenotypes for altered differentiation and immune cell communication partially arose from periodontal pathogen ("periopathogen") interactions. We first used a *16S* rRNA ISH probe common to all bacteria[34] in healthy and periodontitis tissues (Fig. 4a). In health, *16S* was frequently detected in suprabasal keratinocytes; however, in periodontitis, we noted higher counts of *16S*+ basal keratinocytes and stromal cells, with generally higher burdens in JKs. We wanted to reveal cell-specific, species-specific bacterial burden and adapted the "Single-cell Analysis of Host-Microbiome Interactions"

pipeline (SAHMI; see Methods) to identify sparse bacterial reads from single-cell datasets (Fig. 4b; Supplementary Data 2). We first looked at healthy samples, identifying few cell-microbial associations (Fig. 4c); yet, in diseased samples, we observed many more bacterial reads per cell. Well-known periopathogen *Porphyromonas gingivalis*[35] had the largest increase–nearly 200-fold–in keratinocytes and larger in other immune cell populations (Fig. 4d, e). Other well-known periopathogens *Leptotrichia sp*[36]., *Treponema (T.) denticola, T. medium, T. vincentii*[37], and recently-associated periopathogen *Pseudomonas aeruginosa*[38] each had >2-fold increases, suggesting cell-specific enrichment for oral microbes and/or their outer membrane vesicles (OMVs) containing bacterial nucleic acids in/on human cells may increase in disease states[39].

**Fig. 2 | Gingival keratinocyte diversity is molecularly defined, spatially distinct, and preserved in vitro. a** To validate keratinocyte heterogeneity, healthy human gingival tissues were preserved on the tooth surface after extraction and fixed. **b** IHC revealed the gradual transition from tooth-facing JKs and SKs to GM, AG, and AM keratinocytes using KRT19; **c** KRT19-high and -low epithelial stem cells proliferate in the basal layer in health. **d** All keratinocytes (KRT14⁺) were subclustered from the integrated periodontitis meta-atlas (30 samples, 8584 cells) and assigned annotations (**e**) based on Louvain clustering. Cell signatures for these populations are plotted and included in Supplementary Data 1. **f** Using these signatures, we used a custom 12-plex ISH panel to reveal heterogeneity in keratinocyte populations (AG, GM, and JK here; SK and AM as in Fig. 1). Markers such as *CXCL14* and *NEAT1* marked the AG basal epithelium in the opposite pattern of KRT19 protein of the SK/JK cells; *ODAM, RHCG, IL18,* and *SAA1/2* marked SK and JK cells in sequencing and in situ (See

Supplementary Fig. 3 for sulcular, marginal, and alveolar mucosa imaging). **g** Primary human gingival keratinocytes were cultured over multiple passages. KRT19-high (marked by+) basal and larger suprabasal keratinocytes are found in mixed populations at (**h**) first passage and over (**i**) multiple passages. **j** Using RNA ISH and additional markers, cell subpopulations that were defined in vivo such as AG (*LGR6⁺*) and SK/JK (*FDCSP⁺*) can be identified in vitro, suggesting a heterogeneous 2D model of tooth-facing and oral-facing keratinocytes can be utilized for future assays and that these markers are more likely cell identities than cell states. Abbreviations: P Passage; see Fig. 1 legend. Sequential sections from samples were used (*n* = 3 health). Scale bars: **b** 100 μm; **c** 50 μm; **f** 25 μm; **i, j** 10 μm. Illustration from (**g**) created with BioRender.com. For this figure, *n* = 30-sample, 8554-cell for scRNAseq; *n* = 3 for tissues; *n* = 2 for unique primary cell lines.

---

*P. gingivalis, F. nucleatum,* and *T. denticola* have been found intracellularly in keratinocytes, fibroblasts, and endothelial cells[40]. We hypothesized that SK/JKs may harbor these microbes—or microbial fragments—intracellularly. Using healthy and diseased keratinocytes, we analyzed the average number of cells harboring at least one bacterial read per barcode, finding enrichment of key periopathogens in 0.5-2% of all barcodes (Fig. 4f). We then created an ISH panel against 2 of the detected periopathogens (*P. gingivalis, fimA; Fusobacterium, fadA*), with the *16S* rRNA probe as a control. We used tissues first profiled by multiplex IF in periodontitis (see below) and re-probed for microbial mRNA targets (Fig. 4g, h), revealing intracellular signals of both periopathogen targets in 3D—occasionally in the *same* JK epithelial stem cells (Fig. 4h). Whether this is truly intracellular pathogenesis in each case in situ or possibly OMVs containing *16S* or mRNAs remains to be confirmed[39]. At this periodontal niche, >500 species have been detected in humans[41], thus the SK/JKs at the interkingdom interaction "front line" likely play important roles in regulating host-microbe interactions.

## Polybacterial interactions are interlinked with keratokine expression in junctional keratinocytes

We predicted that polybacterial interactions are likely more common than previously appreciated and performed a metagenomic reannotation of the periodontitis atlas (Fig. 1). In this cell-agnostic atlas, single (monobacterial), and multiple species (polybacterial) reads were found in every major cluster (Fig. 5a). Focusing on how this impacted keratinocytes, we found chemokine signatures such as *CXCL1/3/8/17* and *CCL20* that had been found in previous analyses, in both periodontitis and health (Fig. 5b, c). Furthermore, *CXCL1/3/8* and *CCL20* can attract innate (neutrophils) and adaptive immune cell populations (dendritic and T cells), suggesting potential immunoregulatory roles in chronic disease states.

We hypothesized that the observed increase in SK/JK immunoregulatory roles (Figs. 3, 4) was potentially related to host-microbial interactions at a single cell level; we wanted to validate these polybacterial signatures in keratinocytes with spatial context. Because we consistently saw these cell-specific signatures, we termed this epithelial cell signaling response as "epi-kines/keratokines" i.e., cytokine upregulation by keratinocytes in response to challenge. To investigate single-cell, polybacterial interactions in situ, we created a custom 11-keratokine panel with one *16S* probe and performed three consecutive imaging rounds, aligning the individual images of the same tissue sections to simultaneously assess all 12 probes at single-cell and spatial resolution (Fig. 5c–e; see Methods). Using *16S* revealed lineage tracing-like patterns of polybacterial interactions in epithelial stem cells and their differentiated progeny in AG, GM, SK, and JK regions (Fig. 5f), reminiscent of lineage patterns in mouse oral epithelial stem/progenitor cells[28,42,43]. This is important because these cell types persist in the basal layer and give rise to differentiated suprabasal progeny for many cell divisions, with polybacterial interactions potentially impacting subsequent stem and suprabasal cell generations.

Considering periopathogen increases in biofluids like blood (i.e., bacteremia[44]) and saliva[45] as well as epithelial stem/progenitor longevity in the basal layer[46], this could be a body-wide phenomenon associated with exposures over the lifespan.

When qualitatively assessing cytokine expression, there appeared to be a positive correlation with host expression and *16S⁺* signal (Fig. 5g). To quantitate this, we segmented stroma and basal/suprabasal keratinocytes at the four ROIs (Fig. 5f), subdividing each tissue into 12 segments around epithelial transition regions. We generated custom scripts to run 12-plex ISH analyses from these ROIs. Simultaneous keratokine comparison revealed highly infected stromal cells and keratinocytes displayed cell-specific phenotypes (Fig. 5c). Other markers were enriched in suprabasal cells or the stroma specifically. *IL1A, IL1B, CXCL1,* and *CXCL3* displayed some correlation with highly infected cells (Fig. 5c)—mostly suprabasal cells, likely explaining the differentiation effects predicted for these cells in periodontitis (Fig. 3).

Broadly, when comparing health and periodontitis, we found upregulation of keratokines by higher RNA transcript numbers per cell (Fig. 5d). IL1 superfamily members *IL1A, IL1B,* and *IL36G* were upregulated in periodontitis in *both* the basal and suprabasal layers of the JK; further, though previously reported, we observed concomitant expression of these genes in the peri-junctional stroma. These observations held in *CXCL1* and *CXCL3* expression—both innate immune cell chemokines. We downsampled and reordered keratokine expression heatmaps by *16S* burden in the JK and distant GM regions (Fig. 5h, i). *16S*-high burden directly correlated with high *CXCL1, CXCL3, CXCL8,* and *IL1A,* with *IL6* additionally corresponding to basal and suprabasal layer *16S* burden—in either ROI. This was more pronounced for C-C and C-X-C motif chemokines in either region. Consistently, single-cell *16S* signals were most clearly associated with broad effector cytokine expression in situ.

## Tooth-facing keratinocytes are predicted to support immune response coordination for innate and adaptive immune cell types

Having confirmed cell identities of SKs and JKs and gained some insight into their linkage of differentiation, immune signaling, and microbial interactions, we further annotated innate and adaptive immune cell populations using CellTypist (see Methods and Supplementary Fig. 6a–c; Supplementary Data 1). We noticed that some DEGs in keratinocytes targeted both adaptive and innate immune cells. We became interested to understand their targets through receptor-ligand analysis. We utilized CellChat to understand cell-cell interactions via receptor-ligand interactions (see Methods), considering innate and adaptive populations separately. In health, keratinocyte receptor-ligand interactions between the same and heterogeneous subpopulations were high, as evidenced by larger "nodes". This phenomenon diminished in periodontitis. Immune cell communication patterns sharply increased, with larger nodes for multiple T and dendritic cell subpopulations (Fig. 6a). We next investigated information flow from keratinocytes to innate and adaptive cells (Fig. 6b). Healthy and diseased keratinocytes

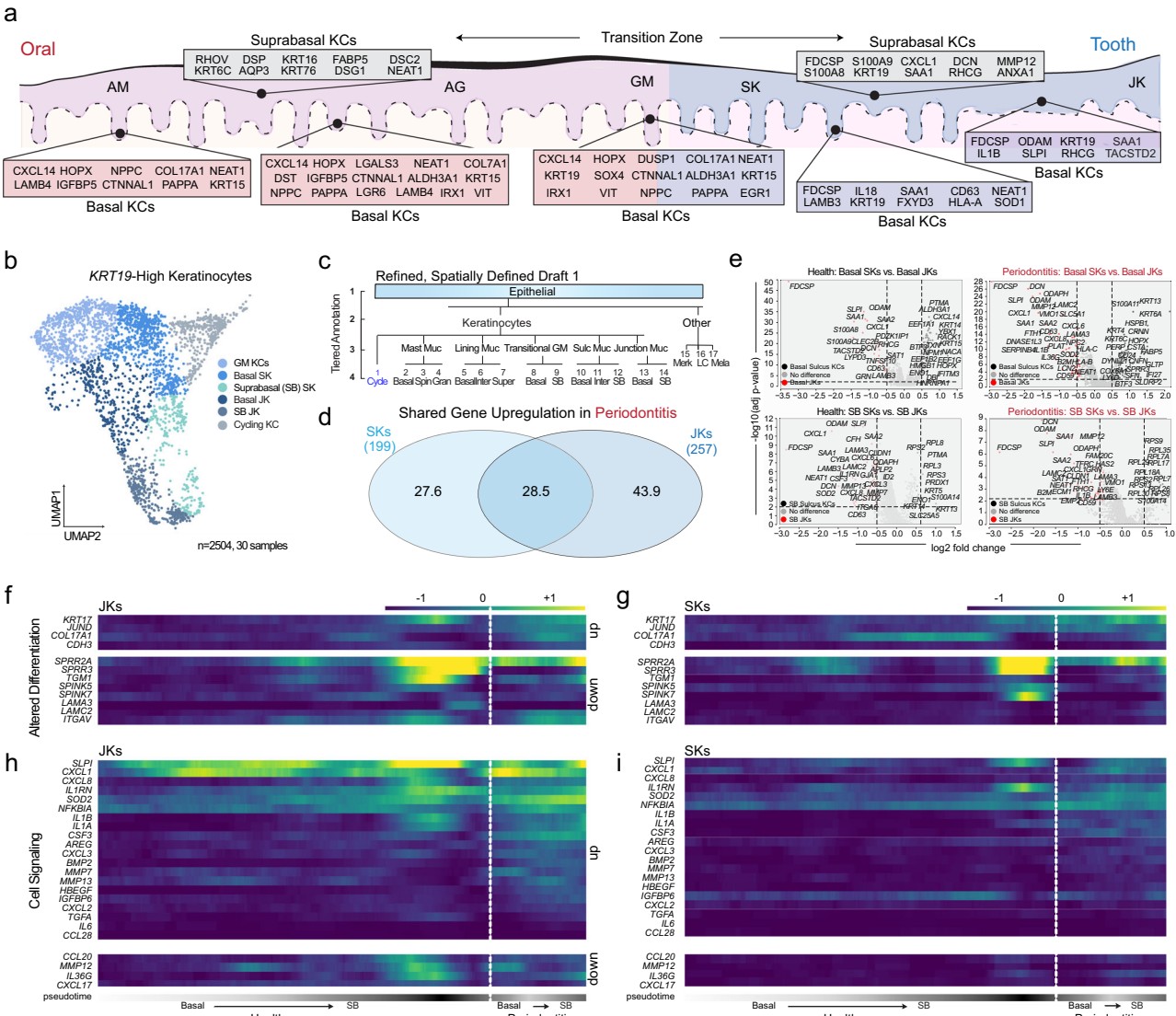

**Fig. 3 | Increased proinflammatory profiles coincide with altered differentiation patterns of tooth-associated keratinocytes. a** Due to single-cell annotation and in situ validation (Fig. 2; Supplementary Fig. 3), a draft model of the oral-to-tooth transition zone in humans is presented, with basal and suprabasal keratinocyte markers uniquely identifying the alveolar mucosa (AM), attached gingiva (AG), gingival margin (GM), sulcular epithelium (SK) and junctional epithelium (JK). **b** These markers allowed for *KRT19*-high keratinocyte (KC) cell subclustering for the first time (2504 cells in total), including gingival margin keratinocytes (GM), sulcular keratinocytes (SK), and junctional keratinocytes (JK). **c** A more granular draft (Tier 4) annotation of epithelial cells of the gingival attachment. **d** Assaying differentially expressed genes in periodontitis, SKs and JKs only share about a quarter of upregulated genes. JKs displayed nearly 125 unique upregulated genes in diseased cells. **e** Further analysis of basal and suprabasal (differentiating) SK and JK keratinocytes revealed unique cell signatures between basal and suprabasal cell

types. This full list is included in Supplementary Data 1. **f–i** To understand SK and JK developmental progression and cell state alterations, we used partitioned-based graph abstraction (PAGA). **f** Examining the basal to suprabasal transition, JKs display altered gene expressed comparing health to disease cell types, including broader expression of *KRT17* and more expression of *JUND*, *COL17A1*, and *CDH3*. Key differentiation genes such as SPRR family members were also downregulated. **g** JKs displayed robust cell signaling and inflammatory phenotypes, which appear exacerbated in disease states. **h** In SKs, differentiation-related genes were uniquely expressed compared to JKs in health but also appeared altered in disease along the basal to suprabasal trajectory. **i** SKs appeared generally less reactive compared to JKs in disease, with lower overall expression of effector molecules such as *CXCL1, CXCL8, IL1A, IL1B*, and *IL1RN*. Abbreviations: Merk Merkel Cells, LC langerhans cells, Mela melanocytes, Muc mucosa, SB suprabasal Keratinocytes; for others, see Fig. 1 legend. For this figure, *n* = 30-sample, 2504-cell for scRNAseq.

showed a preference for cell adhesion (NECTIN, COLLAGEN, junctional adhesion molecule [JAM], LAMININ) and other pathways such as APP, CXCL, and MIF pathways. In disease, we observed a preference for cell signaling (TGFB, TIGIT, CCL, CD45, and EGF), suggesting cell–cell communication shifts in disease, supporting a coordinated response, likely in conjunction with mucosal barrier defects.

With potential keratinocyte-adaptive and keratinocyte-innate axes identified, we next plotted predicted interactions between basal and SB JKs and SKs to understand cell-cell communication changes at the individual gene level for innate and adaptive populations. In the

innate plots, JK signaling to macrophages, neutrophils, and type 1 classical dendritic cells (cDC1s) dramatically increased; communication by GM KCs decreased. JK basal cell signaling decreased; differentiated SB cell signaling generally increased (Fig. 6c). In the adaptive plots, basal JK and GM KC receptor-ligand signaling also decreased, with most differentiated/SB signaling to natural killer T (NK T), tissue-resident memory T (TRM), gamma delta T (gd T), and cytotoxic T cells. We observed little change in basal/SB SK signaling (Fig. 6d).

We quantified the confidence of receptor-ligand interaction between tooth-associated keratinocytes and immune cells. MIF,

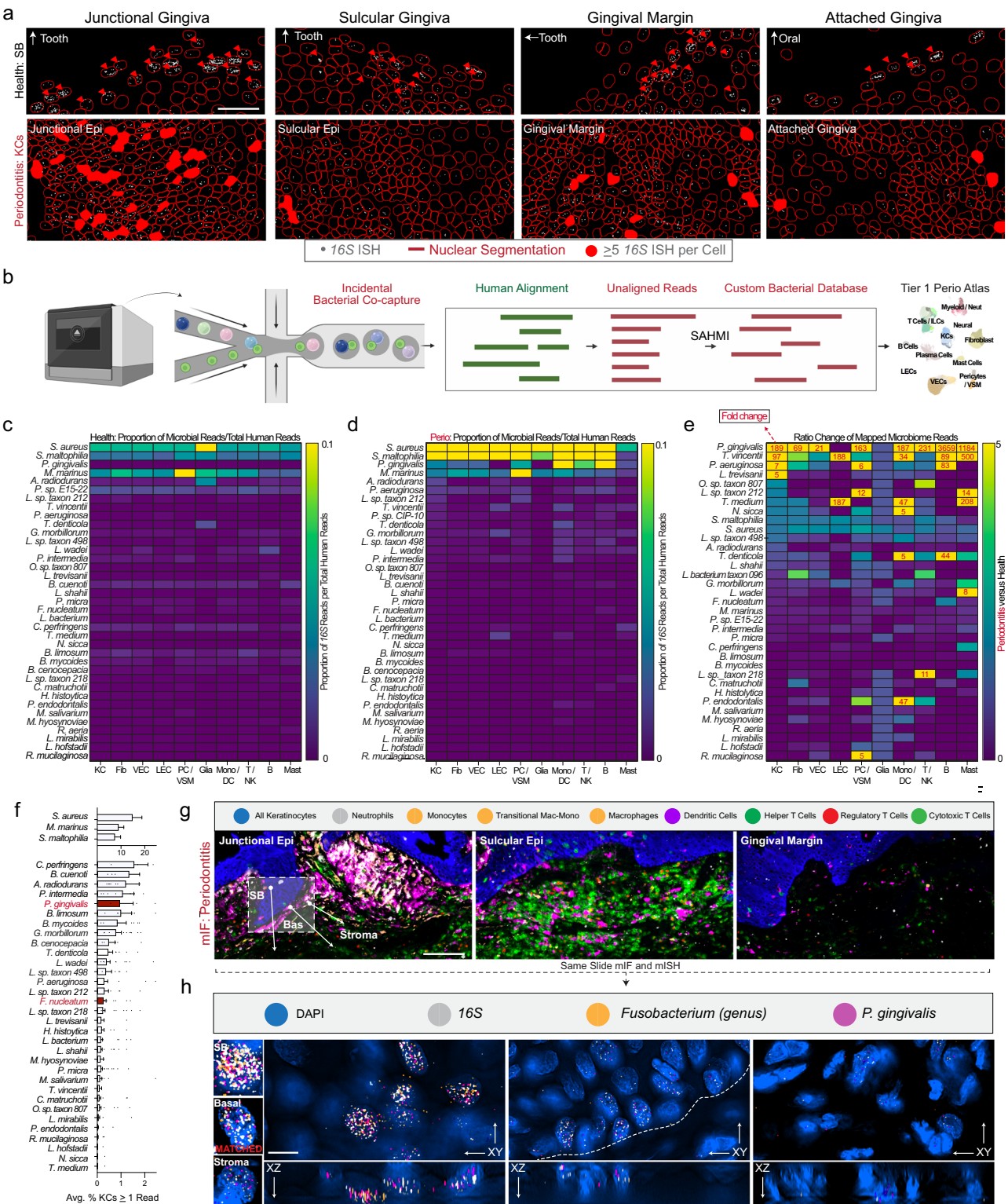

associated with periodontal disease progression[47], was a ligand (via CXCR4 and CD44/CD74) in both the innate and adaptive analyses; predicted interactions for MIF were strongest in JKs for neutrophils, macrophages, and NK and gd T cells (Fig. 6e). We also found decreases in several innate and adaptive cell interactions between structural markers (Fig. 6f; Supplementary Data 2). Overall, basal JK interactions decreased, suggesting these cells are especially affected in periodontitis; however, why these keratinocytes might be affected—and the significance of the remaining interactions—over nearby cells remained unclear.

## Spatial proteomic analysis of periodontitis reveals distinct peri-epithelial immune microenvironments

Informed by our atlas and downstream analysis of SK/JKs at the interface of immune and microbial challenges, we designed a highly multiplexed immunofluorescence (mIF) assay (33-antibody) across healthy and periodontitis samples to understand how disease states affect immune cell patterning around these keratinocytes (Fig. 7a, Supplementary Fig. 7a, b). For whole-slide analysis, we segmented images using StarDist (Fig. 7b, see Methods). In periodontitis, we

**Fig. 4 | Single-cell metagenomic reanalysis of keratinocytes reveals perio-pathogen signals concentrated in tooth-associated keratinocytes. a** Using *16S* rRNA FISH, we segmented cells using StarDist in health and disease tissues. We found bacterial signals primarily focused on the most terminally differentiated suprabasal keratinocytes across each region of the oral- and tooth-associated keratinocytes. In disease, epithelial barrier integrity appeared compromised: we observed more bacterial-associated stromal and epithelial stem cells—especially in SKs and JKs. **b** Using a modified single-cell analysis of host-microbiome interactions (SAHMI) pipeline and a custom Kraken2 database, we used unmapped reads from our integrated single-cell periodontitis meta-atlas. **c** Using a broad Tier 1 annotation of cell types revealed 37 distinct species captured from inside of or membrane-bound to Keratinocytes (KC), Fibroblasts (Fib), Vascular Endothelial (VEC), Lymphatic Endothelial (LEC), Pericyte/Vascular Smooth Muscle (PC/VSM), Glial (Glia), Monocyte/Dendritic (Mono/DC), T/Natural Killer (T/NK), B (B), and Mast Cells (Mast). Using microbial per averaged total reads per human health cell class, we found low read counts across most bacterial species. **d** In periodontitis, we found large associated read shifts, often in well-known periodontal pathogen species ("periopathogen"; i.e., *P. gingivalis, T. vincentii, P. aeruginosa* [*P. sp. CIP-10*]). **e** Performing a ratio analysis of (**d**) over (**c**), we found dramatic increases in many bacteria—especially in known periopathogens. **f** Focusing on all keratinocytes, we found variable bacterial numbers (0.1%-15% of all KCs). **g** Utilizing broad cell classification of our multiplex immunofluorescence data (mIF; Figs. 7, 8), we showed the innate versus adaptive immune disease foci differ between *16S*-high regions in situ. **h** Using the same mIF slides and targets predicted from our SAHMI pipeline, we applied in situ hybridization against *16S* and two common periopathogen mRNA (*P. gingivalis, fimA; Fusobacterium, fadA*). We found polybacterial *16S* and mRNA signals in some epithelial stem cells and their progeny using Nyquist-optimized, three-dimensional imaging. Abbreviations: Fig. 1 legend. Scale bars: **a** 100 μm; **g** 50 μm; **h** 10 μm. Samples from Figs. 7 and 8 were used (*n* = 6, 3 health, 3 periodontitis). Illustration from (**b**) created with BioRender.com. *n* = 30-sample, 8554-cell for scRNAseq; *n* = 3 for tissues.

consistently found concentrated CD45⁺ adaptive immune cells near SK cells; we also found isolated expression of KRT19-high cells in the keratinized mucosa (AG) uniquely attracting CD45⁺ immune cells (inset, Fig. 7b). When cell identities were classified (Supplementary Fig. 7c), peri-junctional niches consistently revealed higher innate immune cell concentrations in periodontitis (MPO⁺-neutrophils, CD14/CD68⁺-macrophages, CD56⁺-natural killer cells, CD11c⁺-dendritic cells), whereas the sulcular region revealed distinct adaptive immune foci (CD8⁺-cytotoxic T cells, CD4⁺-helper T cells, FOXP3⁺-regulatory T cells, and CD20⁺-B cells) (Fig. 7c, d).

Knowing these peri-junctional niches were enriched for *16S* signal, we wondered if SKs and JKs may tolerate and/or support unique immune cell-cell interactions in situ. Using multiple protein markers of the same four ROIs (Fig. 7e), cells were assigned tiered identities (Fig. 7f; see Methods). Proportionally, tissue-wide, immune cell ratios shifted to favor dendritic, macrophage, cytotoxic T, and B cells. Considering local neighborhoods, the sulcus supported more immune-immune predicted "interactions" within cellular neighborhoods, favoring both innate and adaptive immune cell types; however, the junction supported interactions between CD14/CD68⁺ transitioning monocytes/macrophages, CD68⁺ macrophages, and MPO⁺ neutrophils (Fig. 7g).

Using a graph network of immune cell nodes and cell identities (Fig. 8a), we wanted to understand how phenotypes of these immune cells were shaped in the microniche, including cell states and immune checkpoint expression (Supplementary Fig. 7c). We observed few immune cells and little activation (ICOS) or immune exhaustion marker expression (PD-1, PD-L1; Fig. 8b). Phenotypic analysis revealed an immunosuppressed JK-microniche (PD-L1⁺ epithelial and stromal cells) and evidence for SK-localized tertiary lymphoid structures containing mixed active (ICOS⁺) and exhausted (PD-1⁺) populations in periodontitis (Fig. 8c). Few papers have described oral tertiary lymphoid structures as a common feature in periodontitis[48,49]. We quantified this by the same ROIs using single protein markers and manual thresholding to determine the positivity of by converting 0-255 auxiliary units of fluorescence to a binary classification of either negative ("0") or positive ("1"; see: Methods). This revealed proportionally more immune infiltrate in peri-oral-facing stroma (Fig. 8d) and higher innate peri-junctional and adaptive peri-sulcular immune foci frequency in disease (Fig. 8e, f). Notably, the proportion of PD-L1⁺ cells was statistically significantly higher in disease when analyzing JKs compared to SKs (Fig. 8e, f). Thus, the periodontal niche may potentially support immunosuppressed microenvironments through PD-L1/PD-1 interactions nearest to the tooth-JK interface in some disease states.

Assigned cell nodes were assessed for all 10 included cell states, including cell cycling (Ki67), immune activation (GZMB, IFNG, Galectin-3, HLA-A, ICOS), immune tolerance (IDO1), immune memory (CD45RO), and immune exhaustion (PD-1, PD-L1). This extended to cell states of CD3⁺ T cells, which displayed mixed phenotypes of both ICOS⁺ and PD-1⁺ in peri-sulcular foci (Fig. 8c). Assessing cell states, peri-junctional immune cells expressed more GZMB, IFN-G/IFN-γ, Galectin-3, and HLA-A in disease compared to peri-sulcular foci (Fig. 8g–i).

## Discussion

Precision medicine implicitly promises earlier disease intervention, improved clinical outcomes, and generally improved quality of life through extending both life- and healthspan[50]. To date, periodontitis has not benefited from this promise for several reasons: 1) complex host genetics[51], 2) polymicrobial heterogeneity[52] along an antero-posterior axis (likely caused by swallowing[53]), 3) systemic effects on the periodontium from other chronic/genetic diseases[54], and 4) intra-patient "asymmetric burst" and linear "flare" patterns of disease progression[55]. Our study highlights the lack of precision periodontal medicine advances likely arise from an incomplete understanding of this complex niche at a single-cell and spatial biological level. It also underscores that integrated and interkingdom analyses will be necessary to reveal the multi-layered impacts of chronic dysbiosis in susceptible hosts to overcome that lack of progress[56].

Our first step to address this was to create an integrated atlas with harmonized cell annotations and meta-atlas annotation for microbial reads of just one tissue niche in the human body. More studies have been published in the oral cavity, including related to this niche; each will be integrated in future drafts of integrated oral and craniofacial atlases[17]. Here, we focused these analyses just on keratinocytes, but other important cell types were not discovered in high enough numbers for further analyses. As evidenced here and even with large-scale efforts for human scRNAseq i.e., the Tabula Sapiens project using ~500k-cell and 24 body sites[57], more work remains to be done identifying and validating new cell types in human tissues across the body. Furthermore, spatial biology is in its ascendency, helping to determine new cell identities and states[58] and pinpoint cell locations and functions in health and disease[17].

Since this is one of the first studies to use spatial omics assays in the oral tissues, significantly more work remains to understand each niche in more detail to support future precision initiatives. Here, in just one human oral cavity niche in one disease type, multiple microniches/neighborhoods were identified around newly described niche-resident cells[17]. Even before fully understanding disease states, much remains to be understood in adult homeostasis, aging, and development, especially in the oral cavity. The limitation of this study is that it has likely missed as-of-yet undiscovered cell types or states in periodontitis. For example, some rare[59] or difficult-to-sequence cell types like eosinophils[60] are underrepresented in these single-cell atlases, making validation difficult until enrichment for sequencing or targeting through in situ approaches[61].

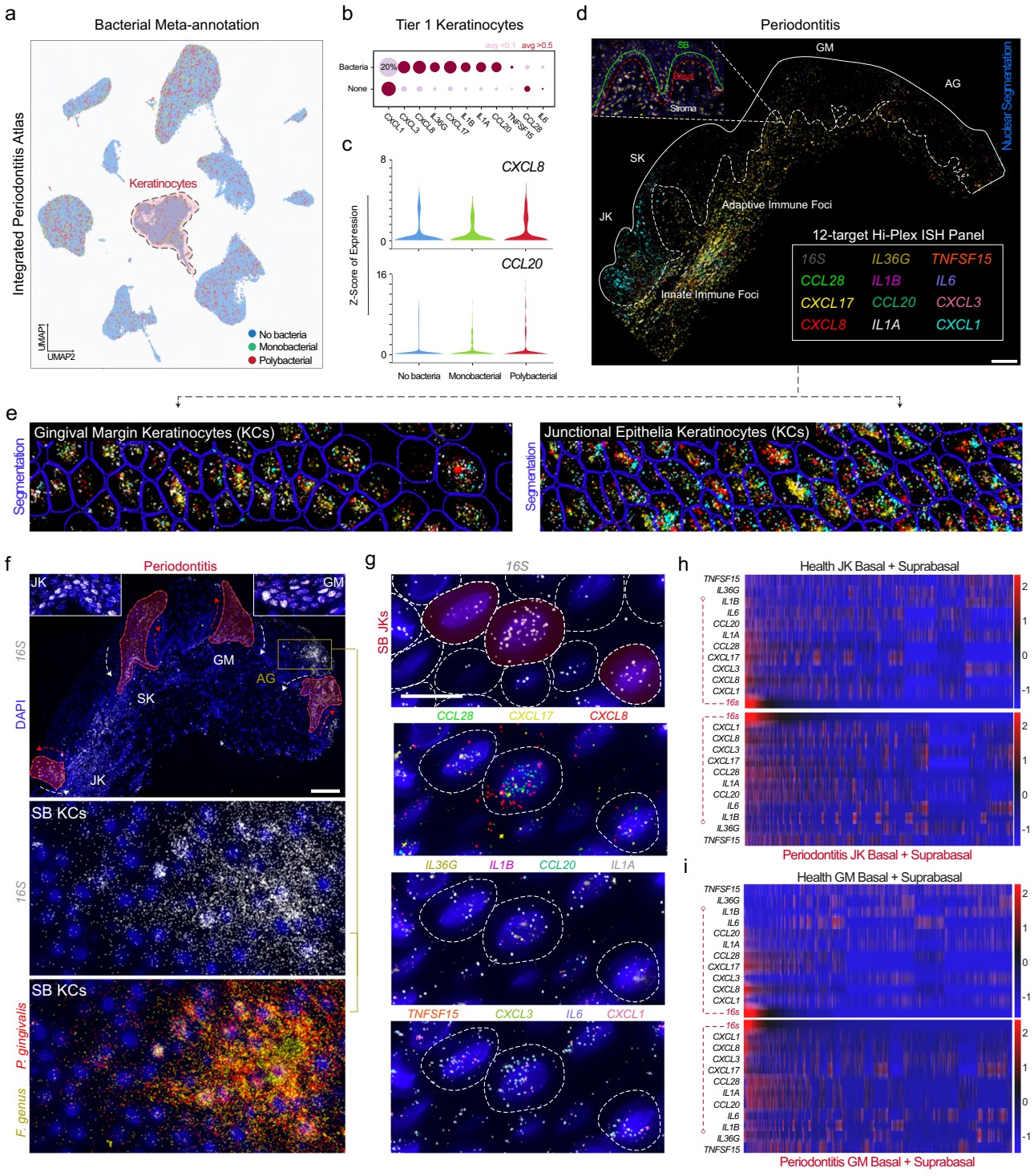

In our atlas, tooth-associated keratinocytes (SK/JKs) represent a rare epithelial cell population (1.1%; stem/progenitor and SB fractions much lower); however, even rarer are epithelial-resident *KRT20*⁺/ *ATOH1*⁺ Merkel cells (0.02%) and *MLANA*⁺ melanocytes (0.1%)—not found in high enough proportions even when combining four studies. Basophils, eosinophils, innate lymphoid cells, and mineralized tissues (osteoblasts, osteoclasts, osteocytes, cementocytes, etc.) were not annotated here; peripheral nervous system contributions via myelinating or non-myelinating Schwann cells were also undersampled (0.05%). Thus, we refer to this as a "draft v1" of the atlas; with more datasets and multimodal sampling of this niche, more will be refined, annotated, and learned that can benefit precision approaches through

digital "tooth" modeling of druggable targets at a single-cell level via available therapeutics (i.e., heart; drug2cell[62]). Adapting this in a spatial context will also be necessary to overcome these precision approach challenges for periodontal diseases.

This original study shows the "spatial" nature of periodontal disease, due to tissue orientation challenges and microniche breakdown in disease states. Here, we link historical knowledge and approaches to annotate new keratinocytes and describe their functions. This study singularly focuses on keratinocytes; more work will be forthcoming about other structural immune influences. As a first focus, we knew that the oral/gingival epithelium is comprised of *KRT14*/KRT14-high epithelial cells, yet some KCs are also *KRT19*/KRT19-high in the gingival

**Fig. 5 | Polybacterial interactions with gingival keratinocytes are highly specific in situ and can be niche- and disease-state agnostic. a** Metagenomic "reannotation" of the integrated periodontitis atlas showed most cells have no bacterial reads. Some (~15%) have 1 read (Monobacterial); a minority have 2 or greater reads (Polybacterial). **b** Disease-agnostic analysis of keratinocyte immune signatures revealed chemokine and interleukin signatures potentially related to bacterial signal—even in health. This was visualized using (**b**) dot plots and (**c**) violin plots. **d** 12-plex custom ISH panel of 11 human effector cytokine mRNA targets and *16S* were overlaid using Warpy. **e** All 11 cytokines are shown simultaneously in GM and JK. Without *16S* overlaid, each cytokine had a distinct patterning. Some i.e., *IL1B*, were broadly expressed in epithelia and stroma. Others i.e., *CXCL3* and *CXCL8* appeared to be cell-specific and enriched in JK over GM keratinocytes. **f** There appeared to be polybacterial patterns in disease that spread to all epithelial regions, including in terminally differentiated attached gingiva keratinocytes i.e., keratinized mucosa. Epithelial stem cell *16S* signal was found in each region (insets).

**g** Even in regions with sparse *16S* signal (intracellular, red overlay), phenotypes appeared highly specific to cells with the highest *16S* per-cell counts. **h, i** Considering keratinocytes at a cell-specific level, we found that *16S* alone is positively associated with most cytokines in health and disease states whether in (**h**) JK or (**i**) GM. Assessing both JKs and GM, all keratinocytes were plotted on a normalized heatmap relative to *16S* expression. We quantified that *CXCL8*, *CXCL17*, *CCL20*, *CCL28*, *IL1A*, and *IL1B* are associated with microbial burden in healthy GM keratinocytes. In JKs, nearly all cytokines were positively associated with microbial burden in heath, suggesting that some bacteria may have cell-specific effects in vivo. Scale bars: (**a, d**) 100 μm (insets; 50 μm); (**b, c**) 25 μm. Abbreviations: ISH In situ hybridization; also see Fig. 1 legend. Illustration from (**a**) created with BioRender.com. Imaging was performed on sequential sections of samples used in Figs. 4, 7, and 8. For this figure, *n* = 34-sample, 105918-cell for scRNAseq; *n* = 6 for tissues: 3 health, 3 periodontitis.

"pocket"[6]. Previous studies showed common cell ontology class representation in each scRNAseq dataset; using Cellenics® enabled the integration and collaborative, harmonized annotation of these datasets as well as KC discovery and validation. It will be important to design future studies with both Tier 1 and Tier 2 clinical metadata, including detailed descriptions of sample origins (CCF[24]) for dataset harmonization to allow for host impact discovery of this chronic disease. Importantly, future studies using gingiva around primary, early erupted succedaneous teeth, and aged periodontal tissues in older adults will help us understand the origin and maintenance of SK and JKs over time. Here, our study suggests that JKs express more odontogenic markers compared to SKs, which is consistent with recent literature[63].

Our study highlights another challenge to precision periodontal medicine with body-wide implications in many diseases with more validation[64]. After cell annotation and spatial validation, our multiomic toolkits gave us insight into cellular programming shifts in single-cell, polybacterial interaction phenotypes only when combined. Utilizing scRNAseq, (m)IHC, (m)ISH, and cell culture, we linked altered differentiation patterns and upregulated keratokines in these new cells to a specific host-microbe-interaction cell state. While these new cell types signaled many cell types, JKs predominantly signaled macrophages and neutrophils (*CXCL1*, *CXCL3*, *CXCL8*), whereas SKs predominantly signaled T/NK and B cells (*CCL20*, *CCL28*), correlating with the polybacterial interaction phenotype—likely with structural immunity correlations with peri-junctional and -sulcular stromal foci and perivascular microniches, which warrants further investigation. Overall, the diseased microniche at the tooth-soft tissue interface appears to be defined by a layered combination of unqiue microbial insults, immunoresponsive and immunostimulatory epithelial cell identities and cell states, and regionalized immunophenotypes. Noting the vast complexity of this disease at a single-cell and spatial level will necessitate multimodal assays of more participants considering disease severity (Stage/Grade), sex, age, genetic ancestry, and other known risk factors i.e., smoking and poorly controlled diabetes.

Though some studies suggest tissue and/or tumor-specific microbiomes exist[65], ours could be discerned at a cell-specific and niche-specific level. While polymicrobial interactions are not thought to occur between multiple viruses within single cells (i.e., superinfection exclusion[66]), polymicrobial infection phenomena are a known cause of multiple inflammatory diseases across the body[67]. Recent work has implicated numerous bacteria such as periopathogen *F. nucleatum* in colorectal and oral cancers[34] with similar immunomodulatory effects from epithelial cell invasion[68]; similar to our periodontitis findings, *Porphyromonas*, *Streptococcus*, and *Leptotrichia* genera co-occur in colorectal cancers, further linking oral-systemic distal sites with epithelial barrier breakdown, stromal immune regulation, immune trafficking to stress sites, and microbial dysbiosis.

We termed this observed phenomenon "polybacterial interactions" because we observed and analyzed these phenotypes at a single-cell and spatial level in vivo with 3D imaging. However, we acknowledge that some or all these signals may be extracellular bacterial aggregates[69] or intracellular OMVs containing bacterial nucleic acids and/or proteins. This is also a limitation of the current study. While many diseases present with polymicrobial infections generally, this has been historically described at a tissue level across bacteria, viruses, fungi, and parasites[70]. While *P. gingivalis* had broad cell tropism in disease using single-cell metagenomics (Fig. 4), *Treponema sp.* appeared to have lymphatic endothelial cell (LEC) tropism, with other enrichment patterns including *P. endodontalis* in pericytes/vascular smooth muscle cells (PC/VCM) and monocytes/dendritic cells and *R. mucilaginosa* in PC/VSM. Our finding of polybacterial interactions in tissues further raises new questions about current long-term periodontitis treatment effectiveness. Understanding host–microbe interactomics through this lens will allow true restoration of the tissue niche considering new strategies that combine biofilm removal, antimicrobials therapy, host-directed immunomodulation, and frequent maintenance of this susceptible niche over the lifespan.

## Methods

### Ethics statement

This research complies with all relevant ethical regulations. Studies using human gingival biopsies were approved by the University of Pennsylvania (IRB #6; Protocol #844933; Lead PI: KIK) and the National Institutes of Health/National Institute of Dental, Oral and Craniofacial Research (NCT #01805869; Lead PI: JL [NIDCR]). Studies using murine samples were approved by the University of North Carolina at Chapel Hill (American Association for Laboratory Animal Science: IACUC ID #20-041.0-B) and the Queen Mary University of London (Animal Welfare & Ethical Review Body ID #P48019841).

### Human integrated periodontitis atlas and mouse keratinocyte atlas generation and analysis

**Human single-cell data reprocessing.** Raw fastq files for the previously published single-cell RNA sequencing projects were downloaded and processed using scripts available here: https://github.com/cellgeni/reprocess_public_10x. Briefly, series metadata was collected using the GEO soft family file. Following this, ENA web API was used to obtain information about the format in which raw data is available for every run (SRR/ERR), as well as to infer the sample-to-run relationships. Raw read files were then downloaded in one of the three formats: 1) SRA read archive; 2) submitter-provided 10X BAM file; 3) gzipped paired-end fastq files. SRA archives were converted to fastq using fastq-dump utility from NCBI SRA tools v2.11.0 using "-F --split-files" options. BAM files were converted to fastq using 10X bamtofastq utility v1.3.2. Following this, raw reads were mapped and quantified using the STARsolo algorithm. STAR version 2.7.10a_alpha_220818

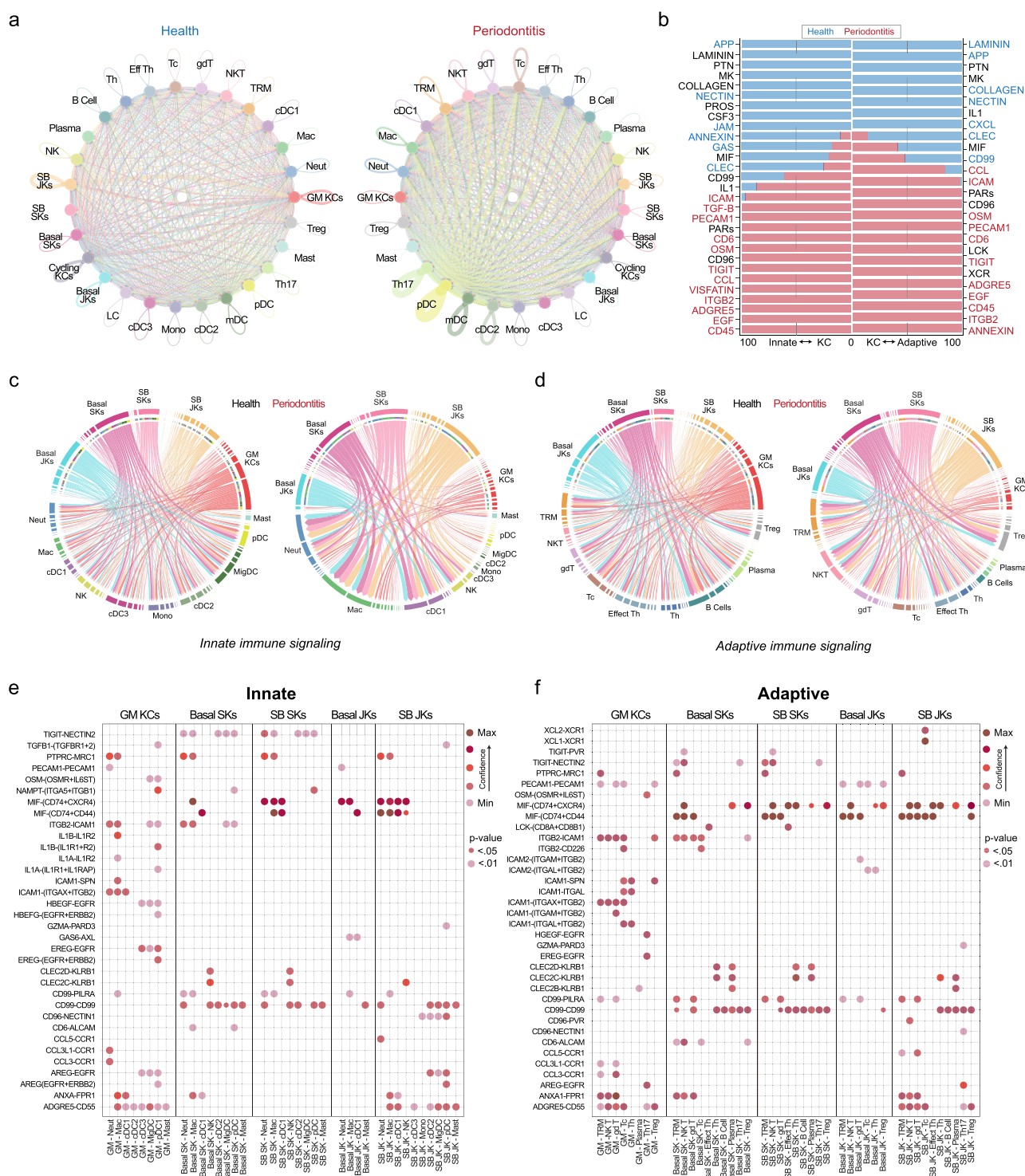

compiled from source files with the "-msse4.2" flag was used for all samples. Wrapper scripts documented in https://github.com/cellgeni/STARsolo/ were used to auto-detect 10x kit versions, appropriate whitelists, and other relevant sample characteristics. The human reference genome and annotation exactly matching Cell Ranger 2020-A was prepared as described by 10x Genomics: https://support.10xgenomics.com/single-cell-gene-expression/software/release-notes/build#header. For 10x samples, the STARsolo command was optimized to generate the results maximally like Cell Ranger v6. Namely, "--soloUMIdedup 1MM_CR --soloCBmatchWLtype

1MM_multi_Nbase_pseudocounts --soloUMIfiltering MultiGeneUMI_CR --clipAdapterType CellRanger4 --outFilterScoreMin 30" were used to specify UMI collapsing, barcode collapsing, and read clipping algorithms. For paired-end 5' 10x samples, options "--soloBarcodeMate 1 --clip5pNbases 39 0" were used to clip the adapter and perform paired-end alignment. For cell filtering, the EmptyDrops algorithm employed in Cell Ranger v4 and above was invoked using "--soloCellFilter EmptyDrops_CR" options. Options "--soloFeatures Gene GeneFull Velocyto" were used to generate both exon-only and full-length (pre-mRNA) gene counts, as well as RNA velocity output matrices.

**Fig. 6 | Cell–cell communication between keratinocytes and immune cells is predicted to occur through innate and adaptive cell-specific programs.**
**a** CellChat was used to understand cell signaling pathways in health and disease considering tooth-associated keratinocytes (basal versus suprabasal; SB and junctional versus sulcular keratinocytes; JK/SKs). Circle plots highlight the significant receptor-ligand interactions between any cell populations, including same-cell type signaling interactions (i.e., JK-JK, SK-SK, etc.). The proportion of interactions increased across more detailed immune cell type annotations (Tier 4 annotations; see Supplementary Fig. 6). **b** Relative information flow in health and disease showed a preference for cell adhesion (NECTIN, COLLAGEN, JAM, LAMININ) and other pathways such as APP, CXCL, and MIF pathways. In disease, more preference for cell signaling pathways is preferred, such as TGFB, TIGIT, CCL, CD45, and EGF. Value of 0 red signifies the pathway is not enriched in periodontitis; value of 100

red signifies highly enriched. Innate (**c**) and adaptive (**d**) immune cell communication was measured gene-by-gene using a chord diagram for visualizing cell-cell communication. **e**, **f** Dot plots showed upregulated signaling pathways in periodontitis at the level of predicted receptor–ligand interactions (y-axis) based on tooth-associated keratinocytes (x-axis). Innate cells appeared to potentially interact with junctional keratinocytes (JK) via CD99-CD99, CD99-PILRA, GAS6-AXL, and MIF-(CD74 + CXCR4/CD44). Adaptive cells appeared to potentially interact via similar pathways. Unique to adaptive cells-JK signaling include XCL2-XCR1; unique to innate cells, AREG-EGFR. Abbreviations: Cycle; KCs Cycling Keratinocytes, Spin Spinous Layer, Granular Granular Layer, Inter Intermediate Layer, Super Superficial; Merk Merkel Cells, Mela Melanocytes, LC Langerhans Cells, MigDC Migratory Dendritic Cells, Mast Mast Cells, also see Fig. 1 legend. Illustration from (**a**) created with BioRender.com. For this figure, $n = 34$-sample, 46835-cell for scRNAseq.

**Cellenics® database generation and subclustering.** The single-cell RNA-seq dataset was processed, analyzed and visualized using the Cellenics® community instance (https://scp.biomage.net/) hosted by Biomage (https://biomage.net/), accessed between May 2022 and February 2024. The team of Cellenics included Alex Pickering, Iva Babukova, Pol Alvarez Vecino, Martin Fosco, Anugerah Erlaut, Germán Beldorati Stark, Sara Castellano, Stefan Babukov, Vicky Morrison, Adam Kurkiewicz, Dana Vuzman, and Peter Kharchenko. The tool itself is Cellenics®, an open-source single-cell analysis toolkit from Harvard Medical School: https://github.com/hms-dbmi-cellenics. Pre-filtered count matrices were uploaded to Cellenics®. Barcodes were then filtered in a series of four sequentially applied steps. Barcodes with less than 500 UMIs were filtered out. Dead and dying cells were removed by filtering out barcodes with a percentage of mitochondrial reads above 15%. To filter outliers, a robust linear model was fitted to the relationship between the number of genes with at least one count and the number of UMIs of each barcode using the MASS package (v. 7.3-56)[71]. The expected number of genes was predicted for each barcode using the fitted model with a tolerance level of $1 - \alpha$, where $\alpha$ is 1 divided by the number of droplets in each sample. Droplets outside the upper and lower boundaries of the prediction interval were filtered out. Finally, the probability of droplets containing more than one cell was calculated using the scDblFinder R package v. 1.11.3[72]. Barcodes with a doublet score greater than 0.5 were filtered out. After filtering, each sample contained between 300 and 8000 high-quality barcodes and was input into the integration pipeline. In the first integration step, data were log-normalized, and the top 2000 highly variable genes were selected based on the variance stabilizing transformation (VST) method. Principal-component analysis (PCA) was performed, and the top 40 principal components, explaining 95.65% of the total variance, were used for batch correction with the Harmony R package[73]. Clustering was performed using Seurat's implementation of the Louvain method. To visualize results, a Uniform Manifold Approximation and Projection (UMAP) embedding was calculated, using Seurat's wrapper around the UMAP package[74]. To identify cluster-specific marker genes, cells of each cluster were compared to all other cells using the presto package implementation of the Wilcoxon rank-sum test[73]. Keratinocytes were subset from the full experiment by extracting manually annotated barcodes and filtering the Seurat object. The subset samples were subsequently input into the Biomage-hosted instance of Cellenics®. Filtering steps were disabled since the data was already filtered. The data was subjected to the same integration pipeline as the full experiment. All cells were manually annotated using available literature and CellTypist[75].

**Transfer of Cellenics® data to CELLxGENE.** Annotated cell-level data were downloaded from Cellenics in the form of an .rds file containing a Seurat object. The data was converted by exporting count matrices and metadata from R and loading them using Scanpy version 1.9.3 (https://scanpy.readthedocs.io/). Additional metadata (e.g., age, sex, self-reported ethnicity) from the original datasets were matched to the closest entries in the respective ontology, per CELLxGENE

contribution guidelines (https://cellxgene.cziscience.com/docs/032_Contribute%20and%20Publish%20Data). The final CELLxGENE dataset can be found at https://cellxgene.cziscience.com/collections/71f4bccf-53d4-4c12-9e80-e73bfb89e398.

**DEG analysis using Cellenics® and g:Profiler[76].** Cells (all, keratinocytes, fibroblasts, vascular clusters; Supplementary Fig. 1) were grouped using lasso tools to allow for the pseudobulk RNA sequencing analyses using these Tier 1 annotations. Differentially expressed gene (DEG) lists and volcano plots (volcano plot statistical significance measured as a $p$ value [i.e., ANOVA] log2 fold change) were generated in Cellenics® and exported as .csv files and uploaded to the g:Profiler website (https://biit.cs.ut.ee/gprofiler/gost). g:Profiler is part of the ELIXIR Recommended Interoperability Resources that support FAIR principles. A complete list of those resources can be found: https://elixir-europe.org/platforms/interoperability/rirs. g:Profiler assesses Gene Ontology and pathways from KEGG Reactome and WikiPathways. DEGs were uploaded to the query section and were first analyzed using g:GOSt multi-query Manhattan plots. These data were further analyzed for the results tab (GO:MF, GO:CC, GO:BP, KEGG, REAC, TF, MIRNA, HPA, CORUM, HP, WP). Data from Supplementary Fig. 2 are an incomplete display of all the g:Profiler data. DEGs are included in Supplementary Data 1 for further analysis.

**CellPhoneDB[77] and CellChat[78].** The total number of ligand–receptor interactions between Tier 3 cell types was calculated for healthy and periodontal disease using *CellPhoneDB* (version 3.1.0). The Tier 3 annotated AnnData object was subsetted and separate AnnData objects were saved for healthy and periodontal disease, respectively. Metadata tables containing the cell barcodes as indices were also exported. *CellPhoneDB* was then run as follows: cellphonedb method statistical_analysis metadata.tsv AnnData.h5ad --iterations = 10 --counts-data hgnc_symbol --threads = 2. The *CellPhoneDB* results were filtered by removing those interactions with a $P$ value > 0.05. Results were visualized using a modified form of *CellPhoneDB*'s *plot_cpdb_heatmap* function to allow for re-ordering of cell types. Cell-cell interactions between receptors in Tier 4 keratinocyte subtypes and ligands in innate and adaptive immune subtypes were further explored using the R package *CellChat* (version 1.6.1 using the cell-cell interaction database). The Tier 4 annotated AnnData object was subsetted and separate expression matrices exported for healthy and periodontal disease, together with their respective metadata tables. These were used to create Seurat objects for health and periodontal disease, which served as inputs to *CellChat*. Analyses were performed using the log-transformed normalized gene counts with default parameters and using the human Cell-ChatDB. Cell type composition differences were accounted for when calculating communication probabilities. Data from health and disease were compared to identify significant changes.

**Partition-based graph abstraction (PAGA) plots[79].** The keratinocyte subset count matrices were imported into Scanpy version 1.9.3 to

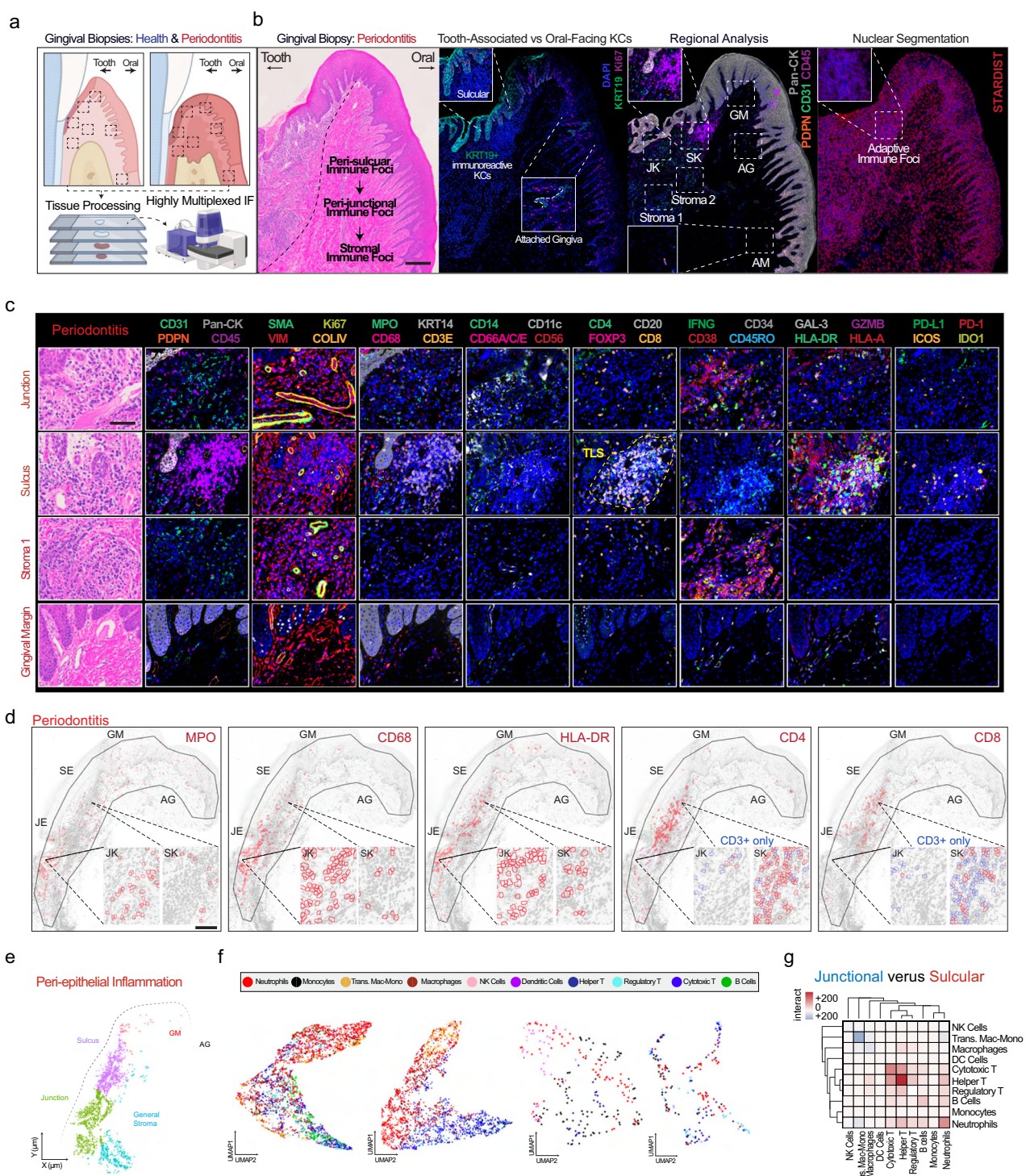

conduct quality control, normalization, and log transformation of the data to control for variability in sequencing depth across cells. To minimize the potential batch effects across the four datasets, a batch correction technique was applied using the Python package HarmonyPy version 0.0.9[73], with the 'sample ID' serving as the batch key. PAGA graphs were constructed using Scanpy's implementation. These graphs were used to explore the relationships between different clusters of cells and to understand the potential developmental trajectories. The coordinates for UMAP[80] were then calculated with the PAGA graph as the initial position, allowing for a visualization that is coherent with the topology of the PAGA graph. To better

understand the developmental progression of cells along these trajectories, pseudotimes were estimated by diffusion pseudotime (DPT) analysis[81] over the PAGA graphs. The DPT is a measure of the transcriptional progression of cells along a trajectory, starting from root cells that were manually selected. Heatmaps were created to visualize gene expression changes along the trajectories, with manually selected start and endpoints, using both Scanpy and Seaborn[82]. To smooth the plots and reduce noise, a moving average of the expression values was used, with a window size of 50 data points along pseudotime. The clustering of the genes in the heatmaps was performed using Ward's method.

**Fig. 7 | Spatial proteomics reveals that peri-epithelial immune microenvironments are uniquely enriched in innate immune populations nearest to junctional keratinocytes. a** The orientation of periodontal tissues is critical to show tooth-facing (sulcular, junctional epithelial keratinocytes; SK/JKs) and oral-facing (gingival margin, GM; attached gingiva, AG; and alveolar mucosa, AM) attachments for highly multiplexed immunofluorescence (mIF) assays of periodontitis. **b** By doing this in sequential sections, we first confirmed orientation and noted highly localized inflammatory profiles near tooth-facing epithelial keratinocytes. As discovered in the initial analysis (Fig. 2), tooth-facing SKs and JKs uniquely express Keratin 19 (KRT19) in every cell type, highlighting the transition zone. An initial analysis of Tier 1 cell assignments using mIF (PhenoCycler Fusion; Akoya Biosciences) revealed adaptive immune foci concentrated near SKs and more diverse, innate immune populated foci near JKs. Cell segmentation was performed using StarDist. **c** The 33-antibody assay revealed more heterogeneity at the cell type and cell state level, including peri-sulcular tertiary lymphoid structures (TLS, yellow) defined by T cell, B cell, and dendritic cell mixed aggregates. Antibodies are grouped and zoomed-in regions from a periodontitis sample are featured. **d** Spatial analysis of peri-epithelial regions was broken into four specific regions as before, highlighting the innate (MPO, CD68, HLA-DR) to adaptive (CD4, CD8) cell transition. **e** Segmented immune cells were assigned identities in health and disease across the four regions. **f** Periodontitis displayed more diverse heterogeneity considering the whole tissue. **g** However, cell–cell interactions among immune cells revealed diverse enrichment of immune cell types in peri-junctional and peri-sulcular immune foci in periodontitis. Abbreviations: Antibodies (see Methods); also see Fig. 1 legend. Scale bars: (**b**, **d**) 250 μm, (**c**) 50 μm. Illustration from (**a**) created with BioRender.com. For this figure, $n = 6$ for tissues: 3 health, 3 periodontitis).

**Mouse single-cell RNA library preparation, sequencing, processing, and analysis of data.** All the necessary animal procedures were followed according to the UK law, Animals Scientific Procedures Act 1986. The experiments were covered by the necessary project licenses under the Home Office and Queen Mary University of London's institution's Animal Welfare & Ethical Review Body (AWERB). The mouse tissues were obtained at Queen Mary University of London, Barts & The London School of Medicine and Dentistry. Mice from both genders were maintained on the C57BL/6N genetic background and were housed under a 12-h light/12-h dark cycle, at temperatures of 20–24 °C with 45–65% humidity. Single-cell suspensions of gingival tissue were obtained from P28 mice, sacrificed by cervical dislocation. Three biological replicates were pooled together to give one single sample for sequencing. Both males and females were used. Fresh gingival tissues were processed immediately after dissection, cut into smaller pieces in a sterile petri dish with RPMI medium (#11875093, Sigma) and digested for 30 min at 37 °C under agitation using the Miltenyi Mouse-Tumor Dissociation kit (#130-096-730). The resulting cell suspension was consecutively filtered through 100 μm and 70 μm cell strainers and cells were collected by centrifugation. The viability of the cell suspension was determined using a Luna-FL automated cell counter (Logos Biosystems). Single-cell cDNA library was prepared using the 10x Genomics Chromium Single-cell 3′ kit (v3.1 Chemistry Dual Index). The prepared libraries were sequenced on Illumina® NovaSeq™6000 (2 × 150 bp) with a targeted sequencing depth of ~30,000 reads/cell. The cell ranger-6.0.1 pipeline was used for processing the scRNAseq data files before analysis according to the instructions provided by 10x Genomics. Briefly, base call files obtained from each of the HiSeq2500 flow cells used were demultiplexed by calling the "cellranger mkfastq". The resulting FASTQ files were aligned to the mouse reference genome (GRCm38/mm10), filtered, and had barcodes and unique molecular identifiers counted and count files generated for each sample. The raw count matrix output from *CellRanger* was then processed by the ambient RNA removal tool CellBender[83], giving an output-filtered count matrix file. This was used for subsequent preprocessing and data analysis using Python package 3.8.13 with the *Scanpy* pipeline[84]. For basic filtering of our data, we filtered out cells expressing less than 200 genes and less than 100 counts. We filtered out genes expressed in less than three cells and with less than 10 counts. Cells were filtered out by applying the following thresholds: 1) more than 20% mitochondrial reads; 2) ribosomal reads lower or higher than the 5th and 95th percentile; 3) more than 1% of hemoglobin reads and 4) total reads lower than 700 and higher than 50,000. *Scrublet*, a doublet removal tool was applied to further remove predicted doublets[85]. To ensure that the data was comparable among cells, we normalized the number of counts per cell to 10,000 reads per cell. Data were then log-transformed for downstream analysis and visualization. The cell cycle stage was predicted using the *sc.tl.score_genes_cell_cycle* tool[86]. We regressed out cell-to-cell variations driven by mitochondrial, ribosomal, and cell-cycle gene expression and the total number of detected molecules. We then scaled the data to unit variance. The neighborhood graph of cells was computed using PCA presentation ($n$ PCs = 40, $n$ neighbors = 10). The graph was embedded in two dimensions using UMAP as suggested by *Scanpy* developers. Clusters of cell types were defined by the Leiden method for community detection on the generated UMAP graph at a resolution of 0.1. Epithelial clusters were used for the second-level clustering. The respective cell types were identified upon annotation of clusters from first-level clustering. The cluster-specific barcodes were retrieved as a list, which was used to select the cells of interest from the filtered count matrix on a separate Jupyter notebook and were re-analyzed separately. Epithelial cells were filtered and analyzed as previously described, and clustered at resolution 0.5 using Louvain.

**Adapted Single-cell Analysis of Host-Microbiome Interactions (SAHMI)[87].** The standard Kraken2 database (version 2.1.3) was downloaded. To avoid overlooking potential oral microbes, genomes from the Human Oral Microbiome Database (HOMD) (https://homd.org) not present in the standard database ($n = 1,502$ taxIDs) were also downloaded, and this custom database was built using kraken2-build[88] with default parameters. Reads from were taxonomically classified using Kraken 2, with "−use-names" and "−report-minimizer-data" (Kraken2Uniq) but otherwise default parameters. True positives from Kraken2 results were identified using barcode level denoising from the SAHMI pipeline and rRNA enrichment. First, barcode denoising was performed. True taxa were identified by performing Spearman correlations between the number of unique and total k-mers across barcodes in each sample. Taxa found to significantly correlate ($p$ value < 0.05) in at least one sample were retained. For all retained taxa, genomic contigs belonging to the taxa were extracted from the Kraken2 database and the reads that were classified to that specific taxa were then mapped to those genomic contigs using bowtie2 (version 2.2.5)[89] with default parameters. Additionally, rRNAs were annotated along the extracted genomic contigs using barrnap (version 0.8) with default parameters. BEDTools (version 2.30.0)[90] coverage was used to count the number of aligned reads overlapping annotated rRNAs. We then calculated the fold enrichment of reads across rRNAs relative to the entire genome, normalized by rRNA and genome length, respectively. Taxa found to contain at least a fivefold enrichment in rRNA sequences relative to the whole genome, which is expected for bacterial transcriptomics that is not rRNA depleted, were retained. From the human host reads, we previously identified which barcodes corresponded to which cell types. Because the reads that are classified to microbial taxa also contain these same barcodes, we then assigned cell types to the microbial reads. To calculate the relative abundance of taxa in each cell type, we divided the total number of reads classified to those taxa with a barcode assigned to that cell type by the total number of reads in the sample assigned to that cell type.

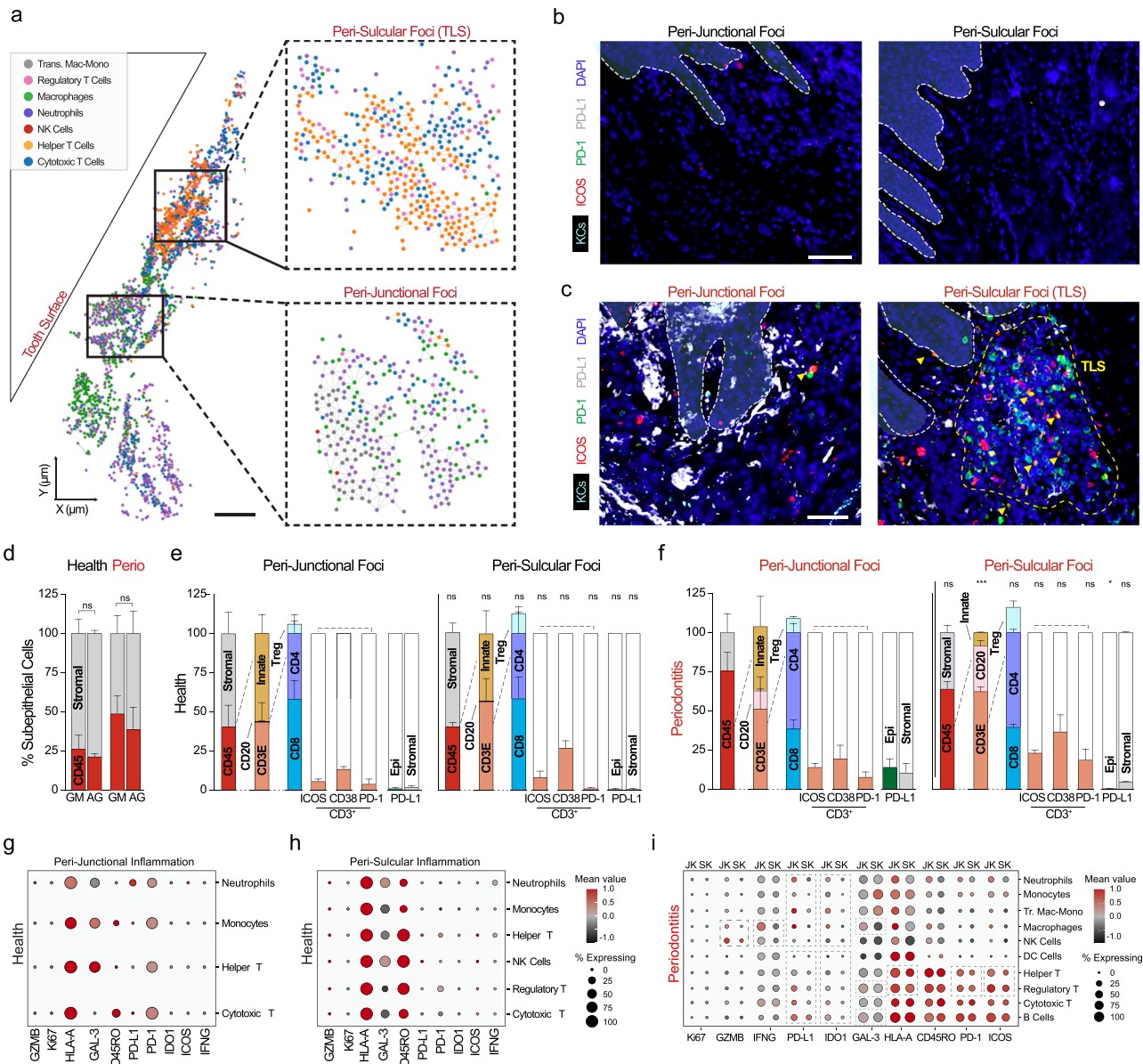

**Fig. 8 | Spatial analyses of peri-junctional and peri-sulcular microniches display regionalized and distinct immunophenotypes. a** Network diagram analysis of only immune-immune cell predicted interactions highlighted regional differences in immune cell aggregation in periodontitis. Both cell identities and their cell states are spatially distinct in periodontitis, with the peri-junctional immune cells expressing more immune exhaustion phenotypes than (**b**) health, contributed by both epithelial cells and stromal-resident cells in (**c**) periodontitis. Peri-sulcular immune cells, including tertiary lymphoid structures (TLS) more often found localized near this zone, contained mixed active (ICOS⁺) and exhausted (PD-1⁺) populations in periodontitis, even when compared to peri-junctional foci. **d** Manual thresholding was performed to show individual marker heterogeneity along the peri-epithelial niches. The cell assignment algorithm for this study is shown in Supplementary Fig. 2. **e** Single marker analysis of CD45⁺ cells revealed general increases in both GM and AG stroma. **f** Comparison of cell identities and cell states

showed minimal differences between peri-junctional and peri-sulcular niches in health. While both peri-junctional and peri-sulcular immune infiltrate increased in disease, peri-junctional foci were biased toward more innate and relatively more CD4⁺ T cells compared to peri-sulcular niches, which were biased toward adaptive immune cells (T and B cells). Both junctional stromal and JKs also expressed more PD-L1 compared to sulcular cell types. Dot plots from total spatial analysis of immune cell states in health considering (**g**) peri-junctional and (**h**) peri-sulcular microenvironments. **i** In periodontitis, differences were quantified between both regions. Abbreviations: Antibodies (see Methods); also see Fig. 1 legend. ns not significant; TLS tertiary lymphoid structure, JK junctional keratinocytes, SK sulcular keratinocytes. Scale bars: (**a**) 100 µm (**b, c**) 50 µm. For this figure, $n = 6$ for tissues: 3 health, 3 periodontitis. Two-sided $t$ tests and chi-square tests were used. *, **, and *** signify less than $p < 0.05$ when comparing healthy JKs to SKs and diseased JKs to SKs.

## Spatial validation and analysis of keratinocytes

**Mouse husbandry.** All mice (Supplementary Fig. 4c) were bred and maintained in an AAALAC-certified animal facility under an IACUC-approved protocol (wild-type to Wallet/ID#20-041.0-B) at the University of North Carolina at Chapel Hill. Each animal was determined to have a healthy body score of at least 3 and had not previously been included in any other panel. Mice were euthanized in accordance with

the Panel on Euthanasia of the American Veterinary Medical Association. In wild-type experiments, animals were only used for validation experiments.

**Tissue preparation, mounting, and sectioning.** Deidentified human gingival tissues were acquired from discarded routine third molar extractions (healthy tissues, Fig. 2 showing gingiva attached to tooth:

NIH/ National Institute of Dental and Craniofacial Research (NIDCR) to LOCI: NCT01805869) or from gingival biopsies (all else: UPenn to LOCI; IRB #844933; MTA #68494). Immediately after extraction, tissues attached to teeth were placed in a 10% solution of N-buffered formalin (NBF) and fixed for a minimum of 24 h in a 4 °C refrigerator. After fixation, the tissues were washed twice in 1X PBS before being placed in 70% EtOH in a 4 °C refrigerator until they were ready to be mounted. Tissues were embedded in paraffin blocks using a Leica system and stored in a 4 °C refrigerator until sectioning using RNAse precautions on a Leica system. Formalin-fixed, paraffin-embedded (FFPE) human gingival tissue on SuperFrost Plus slides was heated to 60 °C on a slide warmer for 30 min. Following deparaffinization for 10 min using HistoChoice Clearing Agent, the tissues were rehydrated using a series of ethanol solutions (100%, 90%, 70%, 50%, and 30% EtOH in nuclease-free water) for 10 min (for 100%) and 5 min each, followed by 2 × 5 min in 100% nuclease-free water. During rehydration, 50 mL of a 1X solution of AR9 buffer (commercially available pH 9 buffer, Akoya Biosciences) in nuclease-free water was prepared and added to a Coplin jar. Following rehydration, the slides were added to the 1X AR9 buffer and covered with aluminum foil. Samples were antigen retrieved in a pressure cooker for 15 min at low pressure. Following antigen retrieval, the Coplin jar was removed from the pressure cooker and cooled for at least 30 min. The slide was then soaked in nuclease-free water for 30 s, followed by soaking in 100% EtOH for 3 min, both in Coplin jars.

**Immunofluorescence on Human or Murine Tissues.** Blocking solution was prepared using gelatin from cold water fish skin (10%; Sigma-Aldrich #G7765-1L), normal donkey serum (10%; Jackson ImmunoResearch #017-000-121), bovine serum albumin (10%; Sigma-Aldrich #A7030-50G), and Triton X-100 (0.2-0.4%; Sigma-Aldrich #T9284-500ML). The sample underwent antigen retrieval as described. A pap-pen was used to draw a hydrophobic barrier around the sample and allowed to dry for 5 min. The slide was then placed in a humidity chamber, and the sample was washed with 1X PBS for 2 × 5 min using a Pipetman (Gibco), then blocked using the blocking solution for 1 h. During blocking, an antibody cocktail containing Rb anti-Hu/Ms KRT19 (clone EP1590Y, 1:500 dilution; Abcam #Ab52625), Ck anti-Hu KRT14 (clone Poly9060, 1:500 dilution; Biolegend #906004), and Rt anti-Hu Ki67 (Clone SolA15, 1:500 dilution; Invitrogen #14-5698-82) in blocking solution dilution was prepared. Following the removal of the blocking solution, this antibody cocktail was added to samples, which were stored overnight in a 4 °C refrigerator. The next day, a donkey anti-species secondary antibody cocktail from was prepared (AlexaFluor 488 [AF488], anti-Rb, 1:1000 dilution, Jackson #711-545-152; Rhodamine Red-X [RRX], anti-Rt, 1:500 dilution, Jackson #712-295-153; and Cyanine 5 [Cy5], anti-Ck, 1:400 dilution, Jackson #703-175-155). The primary antibody cocktail solution was removed, and the samples were washed with 1× PBS for 2 × 5 min. The secondary antibody cocktail was added to samples and left to hybridize for 2 h at rt. This cocktail was then removed, and the samples were washed with 1× PBS for 2 × 5 min. Then, a solution of DAPI (1:1000 dilution in 1× PBS) was added to the sample for 5 min. This solution was removed, and the sample was washed with 1× PBS for 2 × 5 min before mounting with ProLong Gold Antifade. Imaging was performed using a Leica DMi8 with THUNDER Imager (Leica Microsystems) using a 20× 0.8 NA air or 40× 1.35 NA oil objective.

**PhenoCycler-Fusion (PCF) on human tissues.** All reagents in this section were purchased and used as received from Akoya Biosciences unless otherwise noted. Samples underwent deparaffinization, rehydration, and antigen retrieval as described above. Following sample immersion in EtOH, the sample was immersed in Akoya Hydration Buffer for 2 min, followed by Akoya Staining Buffer for 20 min. While the sample cooled to rt, the antibody cocktail was prepared. To 362 µL

**Table 1 | Conjugated antibodies for use in the PhenoCycler Fusion assay**

| Antibody | Clone | Barcode/Reporter | Wavelength | Catalog Number |
|---|---|---|---|---|
| CD20 | L26 | BX/RX020 | AF750 | 4450018 |
| CD8 | C8/144B | BX/RX026 | Atto550 | 4250012 |
| CD4 | EPR6855 | BX/RX003 | AF647 | 4550112 |
| GZMB | D6E9W | BX/RX041 | Atto550 | 4250055 |
| FOXP3 | 236A/E7 | BX/RX031 | AF647 | 4550071 |
| Ki67 | B56 | BX/RX047 | Atto550 | 4250019 |
| HLA-A | EP1395Y | BX/RX004 | AF750 | 4450046 |
| Galectin-3 | M3/38 | BX/RX035 | Atto550 | 4450034 |
| CD3e | EP449E | BX/RX045 | AF647 | 4550119 |
| CD45RO | UCHL1 | BX/RX017 | Atto550 | 4250023 |
| CD45 | D9M81 | BX/RX021 | AF647 | 4550121 |
| PD-L1 | 73-10 | BX/RX043 | AF647 | 4550072 |
| CD14 | EPR3653 | BX/RX037 | Atto550 | 4450047 |
| PD-1 | D4W2J | BX/RX046 | AF647 | 4550038 |
| MPO | AKYP0113 | BX/RX098 | Atto550 | 4250083 |
| CD68 | KP1 | BX/RX015 | AF647 | 4550113 |
| IDO1 | V1NC3IDO | BX/RX027 | AF647 | 4550123 |
| CD31 | EP3095 | BX/RX001 | AF750 | 4150017 |
| KRT14 | Poly19053 | BX/RX002 | Atto550 | 4450031 |
| ICOS | D1K2T | BX/RX054 | AF647 | 4550117 |
| SMA | AKYP0081 | BX/RX013 | AF750 | 4450049 |
| PDPN | NC-08 | BX/RX023 | Atto550 | 4250094 |
| COL_IV | EPR20966 | BX/RX042 | AF647 | 4550122 |
| CD34 | AKYP0088 | BX/RX025 | Atto550 | 4250057 |
| HLA-DR | EPR3692 | BX/RX033 | AF647 | 4550095 |
| CD38 | E7Z8C | BX/RX089 | Atto550 | 4250080 |
| Bcl2 | EPR17509 | BX/RX085 | AF647 | 4250098 |
| VIM | O91D3 | BX/RX022 | AF750 | 4450050 |
| IFNG | AKYP0074 | BX/RX020 | Atto550 | 4250062 |
| CD66A/C/E | ASL-32 | BX/RX016 | AF647 | 4550001 |
| PanCK | AE-1/AE-3 | BX/RX019 | AF750 | 4450020 |
| CD56 | CAL53 | BX/RX028 | Atto550 | 4250087 |
| CD11c | 118/A5 | BX/RX024 | AF647 | 4550114 |

of Akoya Staining Buffer was added 9.5 µL of N, J, G, and S blockers. Then, 157 µL of blocking solution was pipetted into a 1.5 mL vial, and 1 µL of barcoded antibody (Table 1) was added to the vial such that the final volume of antibody cocktail was 190 µL. After immersion in Staining Buffer, the slide was removed, the back and area around the sample were wiped dry, and the slide was added to a humidity chamber. *As a modification to the manufacturer's instructions*, the antibody cocktail was added to the sample, and the humidity chamber was placed in a 4 °C refrigerator overnight. After removal of the antibody cocktail, the slide was placed in staining buffer for 2 min, followed by post-stain fixing solution (10% PFA in staining buffer) for 10 min. Following 3 × 2 min washing in 1X PBS, the slide was immersed in ice-cold MeOH for 5 min. While the slide was immersed, the final fixative solution was prepared by adding 1 vial of fixative (approximately 20 µL) to 1 mL of staining buffer. The slide was removed from MeOH and placed in the humidity chamber, and 200 µL of the final fixative solution was added to the sample. This was left in place for 20 min. Then, the final fixative solution was removed, and the slide was washed in 3 × 2 min in 1× PBS. To convert the slide into a flow cell for use in the PCF experiment, the back of an Akoya flow cell top was removed, and the top was placed adhesive face-up in the Akoya-provided impressing

device. The slide was removed from the 1× PBS, and the edges around the slide that matched where the top of flow cell adhesive would adhere were dried using a micro-squeegee tool (Essential Bangdi). Then, the slide which formed the bottom of the flow cell was placed sample-side down on the top of the flow cell without applying pressure to the adhesive. The tray of the impressing device was inserted into the device, and the lever was gently pulled to adhere to the top and bottom of the flow cell. After 30 s, the lever was depressed, the tray was pulled out, and the flow cell was removed. This flow cell was placed in 1× PCF buffer without buffer additive for a minimum of 10 min before any PCF experiment to allow for improved adhesion between the top and bottom of the flow cell. To prepare the PCF reporter wells, a 15 mL Falcon tube was first wrapped with aluminum foil. To this Falcon tube was added 6.1 mL of nuclease-free water, 675 μL 10× PCF buffer, 450 μL PCF assay reagent, and 4.5 μL of in-house prepared concentrated DAPI such that the final DAPI concentration was 1:1000. Then, this reporter stock solution was pipetted to 18 amber vials, with the volume in each vial being 235 μL. To each vial was added 5 μL of reporter per cycle. The total volume per vial was either 245 μL for a cycle with two reporters or 250 μL for a cycle with three reporters; to optimize reagents and reporters, no cycles contained only one reporter. Only one criterion was used to create a cycle: each cycle could contain a maximum of 3 reporters, corresponding to 1 of Atto550, AlexaFluor 647, and AlexaFluor 750 (where appropriate; see below for more information). A separate pipet tip was used to pipet the contents of each amber vial to a 96-well plate. DAPI-containing vials were pipetted into a well in the H-row, whereas vials containing reporters were pipetted into wells in other rows. Once all wells were filled, Akoya-provided foil was used to seal the wells. Imaging was performed using a PhenoImager Fusion connected to a PhenoCycler i.e., the PhenoCycler Fusion system (Akoya Biosciences) using a 20 × 0.8 NA air objective (Olympus). Requisite solutions for this instrument include ACS-grade DMSO (Fisher Chemical), nuclease-free water, and 1× PCF buffer with buffer additive, the latter of which was prepared by adding 100 mL of 10X PCF buffer and 100 mL of buffer additive to 800 mL of nuclease-free water.

**RNAscope HiPlex 12 V1 or V2 on human or murine tissues.** All reagents in this section were purchased and used as received from ACD unless otherwise noted. The sample underwent deparaffinization and rehydration as described. 1 drop of RNAscope hydrogen peroxide was added to the slides, and the samples were left for 10 min at rt. The hydrogen peroxide was tapped off the slides, and the samples were antigen retrieved and dried as described. During this time, the HybEZ oven was turned on and set to RNAscope (40 °C). The hydration paper was wetted with nuclease-free water to prepare the humidity chamber in the slide tray. An Immedge pen was used to draw a tight hydrophobic barrier around the tissues, then dried at rt for 5 min. The slides were then placed in the slide holder. One drop of Protease IV reagent was added to each contained region. The slide holder was placed in the tray, and the tray was placed in the HybEZ oven for 30 min. During this time, 1X RNAscope wash buffer was prepared in nuclease-free water. The RNAscope hybridization solutions were prepared by adding 1 μL of T probe (Table 2) to 100 μL of probe diluent. The tray was removed, and the slide carrier was immersed in the wash buffer for 2 × 2 min. The carrier was removed and dried, and a paper towel was used to dry the area around the barrier. Then, 20 μL of hybridization solution or 1 drop of positive or negative control probe mix was added to each spot. The slide holder was replaced in the tray, and the tray was placed in the HybEZ oven for 2 h. After 2 h, the slide holder was washed in wash buffer for 2 × 2 min. Signal amplifiers were added to the samples by hybridization of AMP1, AMP2, and AMP3 for 30 min each, with washing in-between steps. After signal amplifiers, T1–T4 fluorophores were added to each spot, with 15 min hybridization and washing. Then, RNAscope DAPI was added to each sample for 30 s. Following this, the DAPI was tapped off the slides, which were immediately mounted with

Prolong Gold Antifade. Imaging was performed using a Leica DMi8 with THUNDER Imager (Leica Microsystems) using a 40 × 1.35 NA oil or 63 × 1.35 NA oil objective.

After imaging acquisition had been completed for T1–T4 probes, the sample necessitated the removal of the first probes for imaging T5–T8 and T9–T12 probes. After completed imaging of T1–T4, slides containing samples were placed in 4X saturated sodium citrate (SSC) at minimum overnight until the cover glass could be gently removed. During this time, an ampule of RNAscope cleaving solution was opened, and a 10% solution of cleaving solution in 4× SSC was prepared. After slide immersion 4× SSC and removal of the cover glass, the slides were added to the slide holder, and one drop of cleaving solution was added to each region containing the sample. The slide holder was loaded into the tray, and the tray was loaded into the HybEZ oven for 15 min. Then, the slide holder was washed in 0.5% Tween for 2 × 2 min. This process was repeated. Following this, T5–T8 probes were added to the sample in the same manner, and the sample was imaged. Once the imaging of T5–T8 probes was completed, their reporters were cleaved as described, and the T9–T12 probes were hybridized and imaged.

For HiPlex V2, no hydrogen peroxide was used, and Protease III was used instead of Protease IV. Additionally, for HiPlex V2, between the AMP3 step and the addition of T1–T4, the addition of FFPE reagent was required as follows. To 100 μL of 4× SSC was added 2.5 μL of FFPE reagent, resulting in a 1:40 solution of FFPE reagent. Following washing slides using 1× wash buffer, the FFPE reagent was added to each slide and incubated at RT for 30 min. Following this, the slide holder was removed from the tray and immersed in 1× wash buffer before proceeding to fluorophore addition.

## Human primary cell culture and analysis

**Human gingival keratinocyte (HGK) culture passaging, cryopreservation, and fixation.** All reagents in this section were purchased and used as received from Lifeline Cell Technology or ATCC. DermaLife K Keratinocyte Medium (HGK medium), the keratinocyte growth kit, and a six-well plate (ThermoFisher Scientific) were brought into a biosafety cabinet using aseptic techniques. The keratinocyte growth kit reagents were added to the dermal basal cell medium. To three of the wells was added 1.5 mL of warmed media. The HGKs (Passage (P)2: Lifeline Cell Technology product #FC-0094, lot #05390; ATCC product #PCS-200-014, lot #80523333) were thawed and aliquoted into 100 μL portions in cryovials (ThermoFisher). One of the 100 μL vials was diluted to 1.5 mL by adding media; then, 500 μL of this diluted media was added to each well, and the six-well plate was placed in a tissue cabinet at 5% CO$_2$ and 37 °C. After 24 h, the media was removed and replaced every 48 h until the cells reached 70% confluence, at which point they were passaged using 0.05% trypsin-EDTA, neutralized, and pelleted using a Sorvall Legend X1R centrifuge (Thermo Scientific) at 1232 RCF for 5 min, resulting in P3 HGKs. Some cells were re-plated in HGK medium and grown using the same procedure until P4 and P5 HGKs were obtained.

Cells that were not plated or re-plated were cryopreserved. P2 HGKs were cryopreserved using the solution in which they were delivered. For in-house passaged HGKs, the excess trypsin/neutralizing solution was removed, and the HGKs were resuspended in 1 mL of Frostalife (Lifeline). The cell suspensions were aliquoted into 1 mL cryovials, which were then placed into a Nalgene Freezing Container that was pre-loaded with 250 mL of ACS Reagent Grade 2-propanol (Sigma-Aldrich). The cells were cooled to -80 °C in an ultra-low-temperature freezer for at least 2 h before immersion in liquid nitrogen. Instead of cryopreservation, some cells were fixed for downstream analysis.

The cells were passaged and either cryopreserved or fixed. Following centrifugation and removal of the excess solution, the cells were suspended in 4% paraformaldehyde (PFA) for a minimum of 24 h.

**Table 2 | RNA in situ hybridization T probes for use in the RNAscope assay**

| Target | Channel | Wavelength | NBCI ref sequence | Chemistry | Cat. # |
|---|---|---|---|---|---|
| KRT19 | T1 | 488 | NM_002276 | V1 | 310221-T1 |
| CCL28 | T1 | 488 | NM_148672.2 | V2 | 418001-T1 |
| Polr2a (+control) | T1 | 488 | NM_000937.4 | V1 | 324311 |
| Krt19 | T1 | 488 | NM_008471.2 | V2 | 402941-T1 |
| LAMB4 | T2 | 550 | NM_001318048.1 | V1 | 530651-T2 |
| CXCL17 | T2 | 550 | NM_198477.2 | V2 | 513241-T2 |
| P. gingivalis | T2 | 550 | NR_040838.1 | V2 | 532132-T2 |
| PPIB (+control) | T2 | 550 | NM_000942.4 | V1 | 324311 |
| CXCL8 | T3 | 647 | NM_000584.3 | V2 | 573621-T3 |
| Fusobacterium | T3 | 647 | NR_026083.1 | V2 | 433881-T3 |
| UBC (+control) | T3 | 647 | NM_021009 | V1 | 324311 |
| CXCL14 | T4 | 750 | NM_004887.4 | V1 | 425291-T4 |
| EB-16S | T4 | 750 | * | V2 | 464461-T4 |
| HPRT1 (+control) | T4 | 750 | NM_000194.2 | V1 | 324311 |
| SAA1/2 | T5 | 488 | NM_000331.6 | V1 | 813351-T5 |
| IL36G | T5 | 488 | NM_001278568.1 | V2 | 424791-T5 |
| COL7A1 | T6 | 550 | NM_000094.3 | V1 | 803381-T6 |
| RHOV | T6 | 550 | NM_133639.3 | V1 | 532931-T6 |
| LGR6 | T6 | 550 | NM_001017404.1 | V1 | 410461-T10 |
| IL1B | T6 | 550 | NM_000576.3 | V2 | 310361-T6 |
| NPPC | T7 | 647 | NM_024409.2 | V1 | 431221-T7 |
| CCL20 | T7 | 647 | NM_001130046.1 | V2 | 409611-T7 |
| ODAM | T8 | 750 | NM_001385579.1 | V1 | 300040-T8 |
| PAPPA | T8 | 750 | NM_002581.5 | V1 | 582481-T8 |
| IL1A | T8 | 750 | NM_000575.3 | V2 | 556791-T8 |
| SFRP1 | T9 | 488 | NM_003012.4 | V1 | 429381-T9 |
| TNFSF15 | T9 | 488 | NM_005118.3 | V2 | 403121-T9 |
| IL18 | T10 | 550 | NM_001562.3 | V1 | 400301-T10 |
| IL6 | T10 | 550 | NM_000600.3 | V2 | 310371-T10 |
| RHCG | T11 | 647 | NM_001321041.2 | V1 | 834511-T11 |
| FDCSP | T11 | 647 | NM_152997.3 | V1 | 444231-T11 |
| CXCL3 | T11 | 647 | NM_002090.3 | V2 | 1002151-T11 |
| NEAT1 | T12 | 750 | NR_131012.1 | V1 | 411541-T12 |
| SOX6 | T12 | 750 | NM_017508.2 | V1 | 524791-T12 |
| CXCL1/2 | T12 | 750 | NM_001511.4x | V2 | 427151-T12 |
| DapB (- control) | T1-T4 | various | EF191515 | V1 | 324341 |

Following fixation, the PFA solution was removed, and the cells were resuspended in 70% ethanol (EtOH) in water. These were stored in a 4 °C refrigerator until future use. SuperFrost Plus (Fisher Scientific) slides were immersed in a solution of 0.1% poly-L-lysine (Sigma-Aldrich) in a Coplin jar for a minimum of 24 h. The slides were rinsed and dried at 37 °C for a minimum of 10 min before use. Cell suspensions were added to 1.5 mL Eppendorf tubes and diluted to 500 μL using 70% EtOH. Cytospin funnels (Fisher Scientific) were rinsed by adding an uncoated slide to the Cytospin clip and adding 70% EtOH to the funnel. The rotor was spun at 1600 RPM for 15 min, resulting in the evaporation of EtOH. Then, a poly-L-lysine slide was added to the Cytospin clip. The funnel was charged with the appropriate cell suspension in 70% EtOH and spun at 1600 RPM for 15 min to obtain cells in a circular spot on the slide. Multiple funnels were used, with rearrangement of the slide and funnels, to obtain four spots per microscope slide. Cells were processed for immunofluorescence and RNA ISH (HiPlex 12) as outlined in the above sections.

Mycotesting was performed using a MycoAlert Plus sample kit (Lonza). Luminescence testing was performed on a Spark (TECAN). Control testing was performed on Lonza MycoAlert Positive Control and deionized water from a Milli-Q IQ7003 (Millipore Sigma). Samples tested included sterile media, media collected prior to passaging cells to P5, and media collected from LPS-challenged and unchallenged cells prior to passaging cells to P4. All test reagents were brought up to rt. The MycoAlert PLUS reagent and MycoAlert PLUS Substrate were each dissolved in 1.2 mL MycoAlert PLUS Assay Buffer. Samples were centrifuged at $200 \times g$ for 5 min. 100 μL of each sample was added to a Hard-Shell PCR 96-Well Plate (Bio-Rad), followed by 100 μL of MycoAlert PLUS Reagent. After 5 min, the first luminescence measurement (Reading A) was taken. Following this, 100 μL of MycoAlert PLUS Substrate was added to each sample. After 10 min, the second luminescence measurement (Reading B) was taken. The ratio of B to A was then calculated (Supplementary Data 1). All cells were found to be mycoplasma negative.

## Statistical methods
**General methods.** All non-sequencing-based data were analyzed in Fiji, QuPath, and/or Prism 9. The selection of statistical tests is described in the text and figure legends; all statistical tests were two-sided. The graphs supporting each figure and the Supplementary Fig.

were generated using Prism 9/10 and a community instance of Celle-nics® (hosted by Biomage) unless otherwise specified. Venn Diagrams were generated using http://www.interactivenn.net/.

**Spatial proteomic cell assignment and analysis.** The determination of marker positivity in each cell was based on comparing its fluorescence intensity to predefined thresholds. We carefully selected thresholds for each marker to ensure accurate cell type classification. A marker was positive if its fluorescence intensity exceeded its threshold and negative if it did not. By applying this criterion, each cell was associated with positive/negative signals for each marker. The assignment of cell types for individual cells was made using consistent, predefined cell type signatures consisting of multiple markers. Cells considered fibroblast/stroma (PanCK⁻, CD45⁻, and CD31⁻), vascular cells (CD31⁺), and epithelia (PanCK⁺) were removed from further analysis. Only cells that were identified as immune cells (CD45⁺, PanCK⁻, and CD31⁻) were kept in the downstream analysis. In most cells, a unique cell type was confidently assigned based on the presence of the positive markers consistent with its signature. However, in some instances, a cell exhibited positive markers from more than one cell type. To resolve this ambiguous cell-type assignment, we implemented a deconvolution approach (adapted from Celesta[91]). The intent was to assign the most likely cell type to each cell in the presence of cell-type mixtures. As such, for each mixed cell type group, we extracted a feature intensity submatrix consisting of those cells and the features relevant to the cell type mixtures they represented. We then utilized a Louvain clustering method[92] implemented in the Seurat toolkit version 3[93] to re-cluster those cells. The clusters were then subjected to cell type assignment based on their enriched markers. Dot plots were used to visualize the marker intensity exhibited across those clusters. The identity of each cluster was then determined by inspecting highly expressed markers, which we determined by the size and intensity of dots in the output matrix and compared with our DEG lists. To infer cellular interactions from the spatial data annotated with cell types, we employed the Squidpy library in Python[94]. Each cell on the slide was represented as a node in the cellular interaction graph. Edges connecting the nodes were created using Delaunay triangulation and were assumed to represent two interacting cells. To remove excessively long edges i.e., unlikely cell-to-cell interactions, a 99th percentile distance threshold was applied to the edges. From the cellular interaction graph, immune cells located in the junctional and sulcular stroma were extracted for further analysis. Interaction matrices were constructed to quantify the number of edges shared between each immune cell type within the junctional and sulcular stroma. Each entry in the interaction matrices represented the number of edges shared between the respective pair of immune cell types. To investigate the variation in immune cell interactions between the junctional and sulcular stroma, focusing on the levels of interaction between each immune cell type, we computed the difference in their respective interaction matrices. Specifically, we subtracted the interaction matrix of the junctional stroma from that of the sulcular stroma. The resulting matrix represented the difference in immune cell type interactions between the two regions, where more positive values represented greater interaction in the sulcular stroma and more negative values represented greater interaction in the junctional stroma. This subtraction matrix was then plotted as a hierarchically clustered heatmap using the Seaborn library in Python. This process was performed for each of the present slides to visualize the differences in immune cell interactions between the sulcular and junctional stroma across the various diseased patients. The resulting matrices were averaged to provide an aggregated view of the variations in the type of cells interacting with the immune cells.

**Quantification and plotting of in situ hybridization.** A multi-step approach to analyze RNA multiplex images was utilized after the image

acquisition was performed using Fiji (ImageJ). The acquired.lif files were subsequently converted into 8-bit images, as an ome.tiff file (scripts available: see Code Availability). The RNA multiplex images were acquired at three different time points, each utilizing four probe sets on the same sample section. To ensure accurate representation, the three images from the same sample were overlaid using the Warpy and image combiner tools[95], generating a 12-plex image represented in different channels. Subsequently, the images were subjected to segmentation using a pretrained model based on Cellpose 2.0[96] (Fig. 5; 12-plex ISH in vivo) or on StarDist[97] (other analyses). The segmentation process was iteratively refined using images from 80 fluorescent images from post-mortem biopsies. The model was trained multiple times until achieving accurate delineation of cell expansion in both the basal and suprabasal layers. The gingival biopsy was segmented separately for each of the areas of interest—junctional epithelia, sulcular epithelia, and gingival margin—with each area divided into basal and suprabasal layers. Following the nuclei-based cell segmentation, subcellular analysis was performed using QuPath, where the number of RNA spots within each cell was detected based on the fluorescence intensity per spot. An individual spot was positive if its fluorescence intensity exceeded its threshold and negative if it did not. The raw data was then exported, and the number of RNA spots was detected and quantified per cell and channel. Subsequently, the raw data was processed as an input file through log scripts to rank the most highly expressed RNA transcripts and extract the field-of-view (FOV) from the X and Y coordinates of each cell. Manual thresholding was applied to define the positivity of the transcripts. To further analyze the RNA transcript data, we transformed the matrix into a Seurat object. The data were treated, and raw clustering was performed in the UMAP projection, as well as PCA evaluation, for each of the twelve probes analyzed, adding the spatial information regarding RNA transcript expression in each patient sample and providing RNA expression patterns within the basal and suprabasal layers from each of the ROIs in the tissue. Individual cell quantification was performed using a similar process (segmentation, manually-thresholded dot quantification) but was conducted independently to validate findings.

**Reporting summary**
Further information on research design is available in the Nature Portfolio Reporting Summary linked to this article.

## Data availability
All data, including links to original raw data from each of the four studies[9,14–16] (GSE152042, GSE161267, GSE164241) can be found at: https://cellxgene.cziscience.com/collections/71f4bccf-53d4-4c12-9e80-e73bfb89e398. Original raw data from the COVID-19 Cell Atlas was uploaded to GEO (GSE266897). Additional raw data available includes the mouse gingival keratinocyte single-cell data (GSE267511).

## Code availability
Analysis notebooks are available at: http://github.com/LOCI/periodontitis https://github.com/sequeira-science/Quinn_et_al_2024_NatComms_mousegingiva.

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

## Acknowledgements

K.M.B. firstly wants to acknowledge all the brilliant and generous teachers in the oral health research field he has had the privilege to learn from over the last 15 years. Our teams further want to acknowledge the tremendous efforts and support of the Human Cell Atlas, specifically

Aviv Regev and Sarah A. Teichmann, for supporting the Oral & Craniofacial Bionetwork these past three years. We want to acknowledge generous and extensive intellectual contributions made by Blake M. Warner to data analysis strategies stemming from collaborations we are actively pursuing to understand the spatial and single-cell impacts of SARS-CoV-2 in salivary glands. Furthermore, we acknowledge that this article has become stronger and more comprehensive through early conversations with Steve Offenbacher, Scott Williams, Julie Marchesan, Karen Swanson, Adam Kurkiewicz, Vicky Morrison, Oliver Gibson, Justin Duplantis, and Peter Kharchenko. We further want to acknowledge the fantastic efforts of the NIH/NIDCR Dental Clinic, specifically Rachel Adam and Danielle Elangue, for their generous support to provide human tissues for testing and validation. Figure 1c, Fig. 2g, Fig. 3a, Fig. 4b, Fig. 7a, and Supplemental Fig. 4a were created with BioRender.com and released under a Creative Commons Attribution-NonCommercial-NoDerivs (CC-BY-NC-ND) 4.0 International license. Services in support of the research project were provided by the VCU Massey Comprehensive Cancer Center Bioinformatics Shared Resource. Massey is supported in part with funding from NIH-NCI Cancer Center Support Grant P30CA016059. Z.R. is supported by the National Institute of Dental & Craniofacial Research (NIDCR) of the National Institutes of Health under Award Number 1K99DE033428. Z.R. was supported by the NIDCR Postdoctoral Training Program under Award Number R90DE031532 (H.K. and Kathleen J. Stebe). D.P. is supported by Fundação para a Ciência e a Tecnologia (2020.08715.BD). IS is funded by Barts Charity (MGU045), the Royal Society (RGS/R2/202291), and the British Skin Foundation (004/RA/23). This work was supported by generous start-up funds from the ADA Science & Research Institute (Volpe Research Scholar Award), the Chan Zuckerberg Initiative/Foundation program Pediatric Networks for the Human Cell Atlas, and the Large Research Grant from the American Academy of Implant Dentistry Foundation (AAID-F) to K.M.B. This publication is part of the Human Cell Atas: http://www.humancellatlas.org/publications.

## Author contributions

For this study, K.M.B. and Q.T.E. conceptualized the project. B.F.M., G.B.S., C.L.W., A.V.P., B.F., K.H., V.R., D.P., K.M., T.W., P.P., X.Q., K.M.T., Z.R., B.T.R., H.K., and K.M.B. developed methods for project analysis. J.L. (NIH) and K.I.K. supported the recruitment of patients and collected data. Q.T.E., Z.C., D.P., I.S., M.B., S.M.W., and K.M.B. collected samples for analysis. Q.T.E., B.F.M., G.B.S., C.L.W., A.V.P., B.F., K.H., V.R., D.P., Z.R., B.T.R., S.H.R., K.M.T., H.K., J.L. (VCU) and K.M.B. performed experimental and/or bioinformatic analysis. Q.T.E., B.F.M., A.H., and K.M.B. performed additional data analysis. K.M.B. and Q.T.E. wrote the original draft; Q.T.E., B.F.M., C.L.W., Z.R., A.H., I.S., H.K., K.M.T., J.L. (VCU), K.I.K., S.A.T., and K.M.B. critically reviewed and edited the final manuscript.

## Competing interests

The authors had access to the study data and reviewed and approved the final manuscript. Although the authors view each of these as non-competing financial interests, K.M.B., Q.T.E., B.F.M., D.P., T.W., A.H., I.S., J.L., and S.A.T. are all active members of the Human Cell Atlas; furthermore, K.M.B. is a scientific advisor at Arcato Laboratories; I.S. a consultant for L'Oréal Research and Innovation; and S.A.T. has consulted for Roche and Genentech and is a scientific advisor for Biogen, GlaxoSmithKline, and Foresite Labs. All other authors declare no competing interests.

## Additional information

[1]Lab of Oral & Craniofacial Innovation (LOCI), Department of Innovation & Technology Research, ADA Science & Research Institute, Gaithersburg, MD, USA. [2]Parse Biosciences, Seattle, WA, USA. [3]The Bioinformatics CRO, Orlando, FL, USA. [4]Wellcome Sanger Institute, Wellcome Genome Campus, Hinxton, Cambridge, UK. [5]Department of Biostatistics, Virginia Commonwealth University, Richmond, VA, USA. [6]School of Computer Science, Carnegie Mellon University, Pittsburgh, PA, USA. [7]Biofilm Research Laboratories, Center for Innovation & Precision Dentistry, School of Dental Medicine, University of Pennsylvania, Philadelphia, PA, USA. [8]Center for Oral Immunobiology and Regenerative Medicine, Barts Centre for Squamous Cancer, Institute of Dentistry, Barts and the London School of Medicine and Dentistry, Queen Mary University of London, London, UK. [9]Akoya Biosciences Inc., Marlborough, MA, USA. [10]Salivary Disorders Unit, National Institute of Dental and Craniofacial Research, National Institutes of Health, Bethesda, MD, USA. [11]Department of Periodontology, Nihon University School of Dentistry, Tokyo, Japan. [12]Department of Periodontics, School of Dental Medicine, University of Pennsylvania, Philadelphia, PA, USA. [13]Respiratory TRACTS Core, Marsico Lung Institute, University of North Carolina at Chapel Hill, Chapel Hill, NC, USA. [14]VCU Massey Comprehensive Cancer Center, Bioinformatics Shared Resource Core, Virginia Commonwealth University, Richmond, VA, USA. [15]Craniofacial Anomalies & Regeneration Section, National Institute of Dental and Craniofacial Research, National Institutes of Health, Bethesda, MD, USA. [16]Marsico Lung Institute, University of North Carolina at Chapel Hill, Chapel Hill, NC, USA. [17]Division of Oral and Craniofacial Health Sciences, Adams School of Dentistry, University of North Carolina at Chapel Hill, Chapel Hill, NC, USA. [18]Department of Oral Biology, College of Dentistry, University of Florida, Gainesville, FL, USA. [19]Department of Physics, Cavendish Laboratory, Cambridge, UK. ✉e-mail: kevinmbyrd@gmail.com

