## [Peer Review File · Nature Communications]

Single-cell and spatially resolved interactomics of tooth-associated keratinocytes in periodontitisREVIEWER COMMENTS

Reviewer #1 (Remarks to the Author):

This manuscript was a comprehensive spatial and multi-omic study pathogenesis of periodontitis. The authors used an array of high-dimensional spatial imaging approaches to understand the cellular immune composition in periodontitis tissues, single and multiplex RNAscope in situ hybridization for host genes and bacteria, as well as single-cell transcriptomic approaches to define inflammation within the oral mucosa. While I do not have major issues or concerns with the manuscript and methods used, I do think some modifications are needed to make this a more accessible manuscript.

1. I think the manuscript in its current form is very dense and difficult to digest the data in its entirety.

2. On Page 7, under the section "Spatial proteomic analysis of periodontitis reveals distinct peri-epithelial immune foci" I believe the authors both misinterpret the CD45+ immune cell clusters and the biologic role of KRT19+ cells with these cells. These cell clusters appear by phenotype, location and composition to be tertiary lymphoid follicles that are found throughout mucosal tissues. In addition, other than their spatial proximity to SK cells, there is no direct evidence that these cells are "inherently chemotactic" or play a direct role in the formation of these cellular structures. The authors should be careful in over interpreting their images.

3. Generally, there are numerous times throughout the manuscript in which the authors make statements of changes in cell population abundance, functional states, etc without providing valid statistical support.

4. On line 147, it is unclear if the "higher innate immune cell concentrations" are specifically in the disease vs healthy state or normally at the junctional region.

5. On lines 154-157, the authors state the "proportion of PD-L1+ cells was nearly 10x higher in health and 2x in disease in the peri-junctional stroma, yet nearly 3X higher in health and 100x higher in disease when analyzing JKs compared to SKs" is a bit confusing. Do the authors mean in healthy peri-junctional stroma there are 10x more JKs than SKs? In addition, where is the statistical analysis? Fig 2e-g has not statistical analysis.

6. Lines 163-168, the statement that "the junction supported interactions between CD68/CD14+ transitioning monocytes/macrophages, CD68+ macrophages and MPO+ neutrophils, highlighting the chronic peri-junctional immunosuppressive niche impact, which validated manual thresholding approaches" does not make sense. How does the presence of these myeloid cells in proximity show an immunosuppressive niche? Did these cells express higher levels of TGFb, IL-10, IDO, etc? Also the comment on about validating manual thresholding does not make sense.

7. On line 179 the authors say they used a "pseudobulk RNAseq analysis" however I cant find reference for this in the Methods section. In addition, based on this description of "pseudobulk RNAseq analysis", it is unclear how the authors are making comparisons in DEG in specific cell populations.

8. The transitions between RNAseq to CODEX on page 9 is abrupt with no lead in or background explanation. The authors should consider a better transition.

9. Throughout the manuscript, the authors jump from spatial imaging approaches, to RNAseq, back to spatial imaging, from human tissue to mouse models, etc., which is quite confusing. It is often not clear which data set the authors are describing in the results section, and it is not clarified when you read the figure legends either. More care should be taken to clearly describe the various data sets.

10. On page 13, lines 277-280, it seems the changes in the cell junction proteins and various cell

receptors could be a function of mucosal barrier defects, which would lead to microbial translocation and the associated inflammatory responses described. In fact, the 16S rRNA and specific bacteria in situ hybridization data would support this, however, this is not mentioned in the manuscript.

11. On line 284, it is important not to over interpret the innate plots in Fig 3. These plots show putative interactions and potential signaling, but without confirmatory direct spatial data, these plots are supportive at best, not definitive proof. Similarly, on line 328, the data do not show "residency" of pathogens, just localization of these bacteria on or in these cells.

12. In the Methods section, under PhenoCycler-Fusion, lines 1213-1219, it is confusing what the composition is for staining buffer, antibody blocking solution and blocking solutions. N, J, G and S blockers needs to be explained. I recommend distinguishing the "antibody blocking solution" from the "blocking solution" and perhaps rename the antibody blocking solution to antibody diluent.

13. In the Methods section, under Spatial Proteomic cell Assignment and Analysis, please explain how threshold was determined? Was it based on isotype control Ab staining or some other method? Threshold will change based on Ab concentration, secondary Ab concentration and fluorophore, exposure time and laser power and gains. More detail is needed on how threshold was set. Also, please explain why threshold changed between slides for the same markers. On lines 1341-1343, how were cell types assigned from spatial transcriptomic (multiplex RNAscope) stained slides?

14. In the Methods section, under Quantification and plotting of in situ hybridization, on line 1382 please explain the threshold method and criterion used.

15. Figure 4a, the presented 16s rRNA ISH data is not sufficient to determine if the authors' claims are correct. First, there are not negative controls presented. What is the false detection rate? Second, providing the cell segmentation in the absence of nuclear staining and membrane staining markers is insufficient. Third, the authors do not provide imaging in 3-D (x/y/z) to demonstrate that the signal is all intracellular. Fourth, a low-to-high mag panel would be highly informative in interpreting these results.

16. Line 114, please define what "Tier 1 resolution" means.

Reviewer #2 (Remarks to the Author):

This study seeks to provide a robust understanding of the microbe-host interface relevant to periodontitis. As the subgingival and supragingival biofilms intimately interact with the gingival epithelium, this study seeks to build an atlas of the epithelial cells that sense and respond to changes in the microbiome and environment surrounding the gingiva. Major findings of this study include the identification and localization of unique keratinocyte cell populations, with the identification that JK and SK populations represent unique immune foci. The characterization of intracellular polymicrobial conditions is exciting, and the observation of multiple species co-infecting cells is a valuable new insight. The abundance calculations of bacteria between health and disease are routinely published, and those presented here are limited due to the small sample sizes analyzed. However, as this is not a significant focus of the study, this reviewer doesn't see it as a major limitation as the overall conclusions are well supported. Overall this is a very well-designed that, despite the density of data and information, is friendly to the reader. The results are impactful, and the conclusions are well justified.

Line 109-110: I know the information is presented in Supp. Table 1, but it would be beneficial to elaborate here on how many samples represent each condition (i.e., healthy, gingivitis, periodontitis)

Line 386: References Fig 8f. Clarify if this is extended data 8f?

Reviewer #3 (Remarks to the Author):

Easter, Byrd and colleagues have performed a landmark study in the field of periodontology. I congratulate them for their fantastic work. Using previously published datasets, the authors have harmonized the metadata (an essential and difficult feat in-and-of itself), and developed a comprehensive (1st version) meta-atlas for the human periodontium. Building from this dataset, and in combination with refined spatial analysis (multiplexed immunofluorescence and in situ hybridization) the authors refine and further resolve the cellular heterogeneity in the periodontal space. Authors introduce the concept of 'epi-kines' or 'keratokines', and evaluate and map how certain epithelial subtypes are skewed toward pro-inflammatory phenotypes, are targeted by multiple intracellular bacteria at the same time, and respond to this bacterial challenge. Finally, I also wish to commend the authors for their extensive methodological section. Below, I include some general remarks and concerns, methodological concerns/questions, as well as line-by-line and panel-by-panel comments, queries and corrections.

General remarks and concerns:

Regarding prior scRNA-seq analysis of periodontal (and dental tissues) not all relevant studies were included in the manuscript (e.g. lines 106-107; 'first' on line 42). Takahashi et al. (2019; PMID: 30509999) and Hemeryck et al. (2022; PMID: 35217915) studied mouse and human dental follicle respectively (preceding the periodontal ligament; admittedly not the most relevant for this study – but should be included in the summary here). Zhao et al. (2021; PMID: 33038290) studied mouse periodontal ligament, including cementoblasts. And although this study is indeed the first meta-atlas focused on the periodontium, including non-dental epithelium (DE) derived lineages, Hermans et al. (2022; PMID: 36299483) developed the first meta-atlas of odontogenic tissue-derived lineages; including analysis of junctional epithelium (JE).

Regarding the discussion of the different subtypes of epithelial cells, I feel there is some discussion lacking on their respective developmental origins. Lineage tracing in mouse has shown that JE is derived from the odontogenic DE lineage (Yajima-Himuro et al. 2014, PMID: 24785116; Yuan et al. 2021, PMID: 32985318; and Tanaka et al. 2021, PMID: 34552180). Perhaps this can explain some of the differences seen between JK and SK (e.g. marker profile; JK express many enamel matrix protein related genes/proteins, e.g. ODAM, FDCSP, ODAPH – as also shown in Hermans et al. (2022, PMID: 36299483 and 2023: 37084723). Interestingly, another study reported that JE might be eventually overgrown by oral gingival epithelium, replacing all odontogenic DE-derived JE (Kato et al., 2019; PMID: 37084723) – I am curious what the authors think about this, and if their data may support these findings or not.

Although figures are highly complex, and well designed, in many instances essential information is missing from the figure, and not included in the legends. In fact, the figure legends (admittedly very well written!) are formulated in such a way to tell the reader how to interpret the figure. In many panels it is not really described in the figure legend what we are looking at, merely how we should interpret it. I do not think this is good practice. This is something that should be thoroughly revised and double-checked for all figures and panels. Below I include some examples of missing information as well:

- Fig 1g: no legend included for dot size and color scale...
- Fig 2d: what exactly are we looking at? Unclear.
- Fig 2k: legend does not help explain what is shown here at all. What do values of +200 and -200 mean? Arbitrary?
- Ext Fig 3c-e: no information on color code, dot sizes, etc.
- ...

Additionally, in figure 1 and 2 the tooth-facing side is positioned on the left; for other figures this is flipped (e.g. Fig 3a, Ext. Fig 4c 2nd row). I would recommend making this consistent throughout the manuscript to prevent any confusion for the reader.

From lines 173-240 the entire content is placed in extended figures 2-5. Perhaps a redesign is needed; this corresponds to half the manuscript up to this point. If these concepts are so essential to the story, perhaps they can also be included in main figures..?

In lines 294-296 the authors conclude that because basal JK have limited interactions, they might be more affected in periodontitis? I feel that this could use more clarification for readers not as versed in immunology – why would they have less interactions with immune cells if they are more affected? The next sections nicely show that they are indeed more affected by intracellular pathogens, which supports these statements – however they also show more expression of 'epikines', so I would expect more interactions to be detected? How can this 'paradox' be explained? Perhaps these are novel signaling interactions (e.g. previously unidentified L-R combinations?) that are not included in the used databases?

Authors use primary human gingival keratinocytes (HGKs) for in vitro studies, and detect intracellular bacteria present for many passages. To me, this is a tremendous finding. Would this also be the case in oral epithelial organoids? E.g. from gingiva or tongue? Identifying intracellular pathogens in organoid models would be an incredible finding!

Can some close-up examples of cell-cell interactions (eg Ext Fig 7) be visualized using (m)IF or (m)ISH? It would be beautiful, and highly supportive of the (bioinformatics) data, to show such cell-cell interactions in situ; e.g. of JK with immune cells...

In Extended Figure 8d-f the response seems stronger in GM and AG cells than in JG and SG? Doesn't this contradict previous data..?

The sections from lines 375-383 and 384-394 are really unclear. The text and figure 6e are confusing. How are the authors making conclusions about stem cells and differentiated progeny? No stem cell markers are used in Figure 6f. It is also not clear what exactly we are looking at. I would recommend to revise this section.

The authors describe increased presence of periopathogens as a consequence of body aging? But is the concept of accumulated exposure the same as aging? This would also indicate overall less presence of intracellular bacteria in basal layers of periodontium in children? Do authors see and correlations with the age of the sample donors?

Methodological concerns/questions:

- scRNA-seq analysis and establishment of meta-atlas: as I understand from the methods, authors employed a 'one size fits all' approach to quality control (QC) of these datasets (line 968)? Is this really suitable? Datasets of different origins will often have widely different ranges regarding QC parameters. Some additional plots of the QC parameters seem warranted. I would also recommend to include additional plots with metadata, also for subclustered data. Certain cell types should not be recovered from certain anatomical positions (e.g. JE from palatal region (Ext Fig 1a seems to show some samples from this region?)).

- Some of the dotplots presented (E.g. Ext Fig 4b) seem to have a lot of background/overlap between annotated cell types. Is this biological or due to background/ambient RNA expression? This can be easily checked using the SoupX package (Young et al., 2020, PMID: 33367645) (or CellBender as performed for mouse scRNA-seq data). Performing this analysis can help clean up the overall picture.

- Why was a different analysis pipeline used for the mouse data? Couldn't this also be analyzed using the same software tools?

- Have authors also performed cell cycle regression? To remove this effect? Perhaps this could further improve clustering of Fig 3b and Ext Fig 4a?

- How was the mIF panel informed by the meta-atlas (line 139; Extended data 1c,d = tier 3 and 4 markers?)? Conceptually this is a very interesting point, that I feel could be discussed more. Which considerations were taken to select the markers employed? Adding to this, it would be beneficial if in the tables (also for mISH panel) the cell type labelled by the marker could be added – this is not always clear.

- Regarding collection of tissues for histology and mISH studies, I am concerned about the methodology employed to remove tissues attached to the enamel surface (lines 1131-1135). Authors don't really describe how tissues were removed. How can we be sure that those tissues

closest to the enamel surface (JE/JK) are not damaged by the process, and are as such accurately represented in this study? Do some remain attached to the enamel surface? These cells are very tightly attached to the tooth to form a microbiological barrier.

- Line 254: 'pseudotime analysis confirmed differentiation': I do not agree that pseudotime analysis alone is sufficient to confirm differentiation. What about cell plasticity that is niche/migration dependent..?

- The authors use the proportion of bacterial reads to the total number of human reads to quantify 'bacterial load' per cell. I am wondering how suitable this is? Diseased states will alter the cells' transcriptome, perhaps reducing (or increasing) the amount of total transcripts per cell – this will of course affect the identified proportions (e.g. Fig 4c,d), right?

Line-by-line feedback:

- The presentation of supplementary tables is very confusing, wouldn't it be better to split the Excel sheets into separate, properly labelled tables? Sometimes Extended Data refers to the tables, and sometimes the files; this is confusing as well.

- Line 74: it should be tissue's instead of tissues

- Line 131: similar proportions of Ki67 is not clear from figure 1J – this could be due to low resolution of image provided for review? Alternatively authors could quantify this.

- Lines 169-170: are GZMB, IFN-g, Galectin-3 etc. signs of the immunosuppressive niche in the junctional zone? (these genes should also be in italics, as it is gene expression)

- Line 186 and extended figure 2: not all of the genes described here are shown in the figure. E.g. CXCL3, CCL20 and IL36G can't be found...

- Line 193: 'in concert with stromal populations'? What do authors refer to here? This is ambiguous... The other forthcoming work referred to in lines 435-436? No data is shown related to this claim in this report.

- Lines 197-199: this phrasing is confusing, making epithelial-xxx crosstalk sound much more generalized/abundant than Ext Fig 3c shows. It is JK/SK with the majority of interactions – which is than more accurately reported in line 200.

- Lines 213; 218: referred to panels of both Ext Fig 4 and 5. Perhaps redesigning the figures (see previous comment as well) is warranted? This is annoying and confusing for the reader.

- Lines 232-233: Did authors use Krt19 antibodies and antigen retrieval methods to be absolutely sure? This could also be included in the methods. This is useful for other researchers in the field. So, how sure can we be of this? And what could be the functional reason behind this suggested disconnect between transcription/translation here? Perhaps the authors can use ISH for Krt19 to confirm specificity of the transcripts for mouse JE at least.

- Line 236: it should be Il1b instead of il1b

- Line 245: '17 keratinocyte populations': phrasing is confusing; because there are other keratinocyte populations (KRT19-) not included in this analysis.

- Lines 257-258: I don't see it like this; SPRR2A also seems to decrease in SK??

- Line 279s and 574: different pathways discussed in text and legend? Why?

- Lines 284-288: cDC1cs is not defined in the text; abbreviations added for some, but not all cell types (e.g. NKT, but not gdT?).

- Line 291: Also CD44 according to plot in Ext Fig 7a?

- Line 292: What about NKT and gdT? Those also seem strong in the plots..?

- Line 301: there is an excessive point before 'infection profiles'.

- Line 319: does this mean 0.5-2% of all cells in the meta-atlas have intracellular bacteria? What does the range refer to? Different cell types? Datasets? Unclear.

- Line 360 and Fig 6a: IL1A and CXCL1 were added to the keratokine panel; but why? (and they are not mentioned in the text)

- Lines 369, 371, 374, 386: wrong figure reference; should be '6' instead of '8'?

- Lines 406 and 407: phrasing?

- Lines 433-434: I agree, but I think the authors can include more references to this 'historic knowledge', and relate their findings more.

- Lines 449-450: is these really an accurate reflection of the data? JK and SK also seem to signal to many other cell types. Rephrase?

- Line 459: 'this study': which study do authors refer to? There are 4 references in this sentence.
- Line 472: replace approached with 'anticipated'?
- Line 495: it should be (f-g) instead of (f-h).
- Lines 501; 1150: it should be immunofluorescence (IF) instead of IHC.
- Line 549: in situ should be in italics; double-check for other instances (e.g. line 583)
- Line 552: in legend SK and JK; in figure 3a JE and SE used. Make these annotations consistently?
- Line 639: (e) should be (d)
- Line 641: (e) is missing
- Line 646: the description for (f) is unclear.
- Line 872: 'as in Figure 1': unclear what is meant here.
- Line 880: unclear why authors refer to figure 5 here?
- Line 904 'PAGA broken out': unscientific language
- Line 1134: NBF abbreviation should be explained
- Line 1143: AR9 buffer should be explained
- I would change the order of mISH and mIF in the methods – as of now mIF is positioned last, whereas it is employed first in the text. I had almost missed it.

Panel-by-panel feedback:

- Fig 1e: Why are Langerhans cells and melanocytes included in the epithelial cell types (line 119)? Neither is derived from epithelium (as Merkel cells)?
- Fig 1g: gene names are too small and close together.
- Fig 2c: Some colors are really hard to distinguish; the panel from the supplementary with info on the markers and which cell type they label would be useful here – or if some kind of annotation could be made.
- Fig 2e: Why are JK/SK not included here? This should be done.
- Fig 2f: sometimes there is no space between CD38 and PD-L1 in the figure labels. A table with numerical values could be interesting to many readers. Also very unclear to see 10x, 2x, 3x and 100x increases as described on lines 154-156 without this information.
- Fig 2h and i: could be merged into one panel? They are basically one panel already given the 'periodontitis' line below combining them.
- Fig 2i: I think this is an 'l' instead of 'i'
- Fig 2i: really hard to identify the colors belonging to each cell on the plots; very blurry.
- Fig 2j: should be 'j' instead of 'i'
- Fig 2l: JK SK are not always positioned the same over the dots in the plot.
- Fig 3d: unclear what the percentages relate to? E.g. 43.9%? of what?
- Fig 3f-i: the up/down are hard to read, and not added for each panel. This makes it confusing to interpret; perhaps a different color scale can be used to indicate downregulation more clearly? E.g. blue to red?
- Fig 3k: TGF-B: which TGF-B gene? There are multiple.
- Fig 3k: What do these values mean? 0-100? Information flow? What does this signify?
- Fig 3j: these plots are almost impossible to untangle and interpret. I don't see some of the interpretation the authors make on lines 281-288. E.g. 'basal cell signaling decreased': I agree this is maybe the case for JK, but this not seem the case for SK? 'In adaptive plots, basal SK decreased': I don't agree? This also seems more or less the same..? The authors conclusion in lines 294-296 matches better with what the figure shows, and not what is claimed in these lines of text.
- Fig 3l,m: adding an indication of 'Innate' and 'Adaptive' to the panels themselves would be useful. The text discussing these panels is difficult to understand. Does each panel check the interactions of the 5 different KC populations vs all innate or adaptive immune cell types?
- Fig 4a: Are these all suprabasal (SB) keratinocytes? Based on lines 294-296 we would also expect more intracellular bacteria to present in basal cells, right? Can the authors quantify 16S+ cells in basal vs suprabasal cells, and include images of both regions?
- Fig 4b: add 'SAHMI' on the scheme.
- Fig 4c-e: what does the red font for some bacteria signify? It is not explained anywhere in the legend, and it doesn't seem to correspond to periopathogens? (e.g. *T. vincentii* is not in red, but described as periopathogen on line 598).
- Fig 4e: what is the color code? Fold change? Not labelled.

- Fig 4e: strange output: e.g. Mast cells have low proportions in both panel c,d, but incredibly high fold change in panel e. Is this relevant then?
 - Fig 4g: different colors and cells are almost impossible to discern.
 - Fig 4f: unclear what this means? Is this the percentage of keratinocytes with intracellular bacteria? Which keratinocytes? The KRT19+ positive ones analyzed in figure 3; or all?
 - Fig 4i,j and lines 334-335: quantify this?
 - Fig 4k: significance is not clear? Why is this relevant? I can't see the fluorescence signal in this panel...
 - Fig 5: Are panels i,j quantifications of images such as in b-h? This isn't really clear to me.
 - Fig 5f: unclear what this panel is? What do arrows signify?
 - Fig 5j: What if binning is performed differently? Instead of >100, in steps of 100? IL6 doesn't really seem to correlated with 16S load in panel i, although the binning in j does make it look like this... Or perhaps this reaches an early plateau, but doesn't really increase anymore as the 16S load increases..?
 - Fig 5i: different scale used for bottom row.
 - Fig 6d: so both suprabasal and basal layers are summed in these plots? Unclear.
-
- Ext Fig 1a: what do these numbers at each reference mean? They don't seem to refer to a specific reference?
 - Ext Fig 1h: I think this is an essential image. Could there also be markers included for Tier 2 epithelial and mesenchymal cell classifications? Can KRT19 be included?
 - Ext Fig 2: wouldn't it be more intuitive if the comparison is flipped? So that the positive LFC corresponds to overexpression in periodontitis?
 - Ext Fig 3c,e: no explanation for color scale
 - Ext Fig 3d: no legend for color code or size of dots
 - Ext Fig 3f: very blurry, I can't read it.
 - Ext Fig 4b: genes are too close together and too small to read.
 - Ext Fig 4c: unclear why some markers are used; and they are not referred to at all anywhere else besides this figure. E.g. LAMB4, COL7A1, RHOV. Further, the legend (line 875) mentions KRT15 and LGR6 which I do not see in this figure..?
 - Ext Fig 5f: can we somehow link these clusters (numerical entities only at the moment) to human cell types (or at least some of them)? This would make the study much more complete!
 - Ext Fig 6d,e: labels are not really that clear/obvious. What is what? Legend is insufficient
 - Ext Fig 7: add labels for 'innate' and 'adaptive' to make this immediately clear for the reader.
 - Ext Fig 8a: increase brightness? Hard to see signal.
 - Ext Fig 8d-f: I would rearrange these panels into 3 rows (e.g. such as f). Will make it easier for the reader.

Reviewer #4 (Remarks to the Author):

Tooth function relies on various cell types. While hundreds of diseases can affect teeth, the specific contribution of each cell type to these diseases has not been adequately explored. This study has comprehensively analysed single-cell RNA sequencing (scRNAseq) on tooth-supporting structures, followed by considerable downstream work to identify previously undescribed keratinocytes. Using this approach, the study has, for the first time, created an integrated periodontitis meta-atlas of human tissues through open-source single-cell analysis. The authors validated these cell identities using multiplexed in situ hybridization (mISH). Subsequently, by utilizing highly multiplexed immunofluorescence assays (mIF), they demonstrated the differences in immune foci between the sulcular keratinocyte (SK) and junctional keratinocyte (JK) niches.

The authors noted that, transcriptionally, SK and JK microniches respond differently in the presence of disease, generally leading to alterations in cell differentiation and the upregulation of effector cytokines which they referred to as 'epi-kines/keratokines'. This effect is predominantly observed in JKs. Furthermore, the authors predicted that foregoing cells differentially regulate innate and adaptive immune cell subpopulations, even in a healthy state (Line 85-90). The findings and interpretations presented in the preceding sentences lack clarity. Authors should revise these sentences. Moreover, detailed discussion in comparison to existing literature should be included in

the discussion section.

As the next step, the authors combined unmapped reads from the periodontal integrated atlas, mIF, and mISH, revealing that some SKs and JKs tolerate intracellular pathogenesis. The analysis identified several distinct intracellular species (Line 91-94). However, it is uncertain whether the authors have conducted a sufficient number of downstream experiments to establish compelling evidence.

The authors mentioned that the junctional region consistently showed higher concentrations of innate immune cells, while the sulcular region revealed distinct adaptive immune foci. Is there sufficient evidence from the conducted experiments to support this statement? There should be more downstream work.

Line 150-153 authors stated that "We quantified this by region using single markers and manual thresholding, revealing more infiltrate in peri-epithelial stroma and higher innate peri-junctional immune foci frequency in disease". The message of this sentence is not clear. Please revise.

Line 157-158, authors stated that "Thus, the periodontal niche may potentially support immunosuppressive microenvironments nearest to the tooth-soft tissue interface, but the reason for this was unknown". It's essential to note that the present study is primarily observation-based, lacking a comprehensive functional analysis of the cell types involved. Therefore, the available evidence might not be sufficient to draw the conclusion that the periodontal niche actively supports immunosuppressive microenvironments.

Line 163 to 168 Authors stated that "Considering local neighborhoods, the sulcus supported more immune-immune predicted "interactions" within cellular neighborhoods, favoring both innate and adaptive immune cell types; however, the junction supported interactions between CD68/CD14+ transitioning monocytes/macrophages, CD68+ macrophages, and MPO+ neutrophils, highlighting the chronic peri-junctional immunosuppressive niche impact, which validated manual thresholding approaches". Once again, the authors' interpretation remains unclear. What exactly is meant by 'chronic peri-junctional immunosuppressive niche impact'? It is preferable to describe the findings clearly without using technical terms that lack a clear explanation.

Line 170-171 "Overall, the junctional zone structural niche appeared to be more immunosuppressive than other peri-epithelial niches, but the underlying reason remained unclear". Similar to previous comments, this statement should make clearer and more detailed, rather than stating "remained unclear". This point can also be elaborated in the discussion section.

Line 201-202 What is the reason that SK/JKs are enriched for cytokines and matrix metalloproteinases even in health? The authors suggest active immune roles of these cells (line 206-207). However, this point can be elaborated in the discussion section.

The authors have proposed that SK/JKs may represent new human cell types. To explore this, they used primary human gingival keratinocytes (HGKs) and discovered the presence and maintenance of KRT19+ cells. Furthermore, through ISH, they identified cell subpopulation-specific markers, such as FDCSP (line 212-222). However, the mere presence of these markers may not be sufficient to definitively establish the existence of new cell types. The authors should consider demonstrating the functional relevance of the proposed new cell type in comparison to other cell types.

From the murine gingival keratinocyte study explained in lines 226-239, it becomes evident that there are notable differences between the mouse and human junctional niche. Furthermore, the IHC analysis of healthy mouse gingiva around M2 and M3 clearly demonstrated the presence of Krt14 and Ki67 but not Krt19, suggesting a lack of gene-to-protein translation. This point could be further elaborated in the discussion section.

The studies described under 'Periodontitis affects SK/JK stem/progenitor differentiation to upregulate inflammatory signatures' (lines 241-251) need to provide better clinical relevance. It's important to recognize that periodontitis is not a single entity. Therefore, authors should provide clear evidence on the reference.

The polybacterial coinfection of human keratinocytes is an interesting and novel aspect in the field. The authors have done commendable work in this area. However, there are a couple of questions that need clarification, and these points should be elaborated in the discussion section.

The authors observed that at least four pathogens can coexist within the same cell, termed 'polybacterial intracellular coinfection phenotype.' Some of these organisms, such as *P. aeruginosa* and *T. denticola*, are anaerobic in nature. How do four microorganisms survive in the same human cell? What is the authors' explanation for their survival mechanisms within the host cell? Why have the authors not detected commonly known oral commensals intracellularly?

Line 405-407 There is an error in this sentence with repeated words. Please correct.

Overall, the manuscript presents interesting and novel findings. However, the discussion section

currently appears to be vague and does not critically analyse the findings. The discussion should focus on specific findings rather than providing an overview of the current status in the field. Therefore, the authors should consider a comprehensive revision of the manuscript.

February 19th, 2024

Dear Reviewers,

Thank you for providing the reviews of our manuscript, "Polybacterial intracellular coinfection of epithelial stem cells in periodontitis", now entitled "Single-Cell and Spatially Resolved Interactomics of Tooth-Associated Keratinocytes in Periodontitis". We found your positive comments both exciting and insightful. We greatly appreciate the breadth of expertise and agree with the important points raised by each of you regarding our work. To address these comments, we have made direct changes to the text or figures or performed additional experiments. We are confident that this comprehensively revised manuscript addresses and accounts for all reviewer concerns from the initial submission. Below are the specific revisions we have made to address the comments from all the reviewers.

Reviewer #1 (Remarks to the Author):

This manuscript was a comprehensive spatial and multi-omic study pathogenesis of periodontitis. The authors used an array of high-dimensional spatial imaging approaches to understand the cellular immune composition in periodontitis tissues, single and multiplex RNAscope in situ hybridization for host genes and bacteria, as well as single-cell transcriptomic approaches to define inflammation within the oral mucosa. While I do not have major issues or concerns with the manuscript and methods used, I do think some modifications are needed to make this a more accessible manuscript.

• We thank the reviewer for their appreciation of the comprehensive array of high-dimensional spatial imaging, in situ hybridization, and transcriptomic approaches we used in this manuscript. We also believe our approach was comprehensive but could also be improved through focused edits on concepts and content.

1. I think the manuscript in its current form is very dense and difficult to digest the data in its entirety.

• We have adjusted the text to try to make it more accessible, in addition to restructuring part of the manuscript to introduce Extended Figures into the main text as well as reorder the manuscript to increase readability.

2. On Page 7, under the section "Spatial proteomic analysis of periodontitis reveals distinct peri-epithelial immune foci" I believe the authors both misinterpret the CD45+ immune cell clusters and the biologic role of KRT19+ cells with these cells. These cell clusters appear by phenotype, location and composition to be tertiary lymphoid follicles that are found throughout mucosal tissues. In addition, other than their spatial proximity to SK cells, there is no direct evidence that these cells are "inherently chemotactic" or play a direct role in the formation of these cellular structures. The authors should be careful in over interpreting their images.

• We thank the reviewer for this insightful comment. We agree that these cell clusters appear to be tertiary lymphoid structures (TLS) or follicles. Periodontitis is a chronic inflammatory disease, and TLS generally form near chronic inflammatory sites i.e., tumors, suggesting the proper formation of one is required for disease resolution. This is the one of the first examples that we know of in the literature of a TLS in gingiva (see: DOI: 10.1111/cei.13584), and we now highlight this. TLS have also been found in other oral sites (see: <https://doi.org/10.1016/j.joen.2023.06.006>). We further understand the reviewer's concern in overinterpreting our images as it relates to the SKs being "inherently chemotactic", though we have performed orthogonal work to suggest these cells at least support the idea of chemotaxis. Regardless, we have limited our discussion of inherent chemotaxis by these cells.

3. Generally, there are numerous times throughout the manuscript in which the authors make

statements of changes in cell population abundance, functional states, etc. without providing valid statistical support.

- We agree. We have made revisions throughout the text to ensure accuracy in describing our findings. We have quantified some of the single marker assays; see Figure 8.

4. On line 147, it is unclear if the "higher innate immune cell concentrations" are specifically in the disease vs healthy state or normally at the junctional region.

- We agree. We revised the text to state "...in periodontitis".

5. On lines 154-157, the authors state the "proportion of PD-L1+ cells was nearly 10x higher in health and 2x in disease in the peri-junctional stroma, yet nearly 3X higher in health and 100x higher in disease when analyzing JKs compared to SKs" is a bit confusing. Do the authors mean in healthy peri-junctional stroma there are 10x more JKs than SKs? In addition, where is the statistical analysis? Fig 2e-g has not statistical analysis.

- We agree that the text is confusing as written. We revised to instead focus on the disease differences between SK/JKs in periodontitis.

6. Lines 163-168, the statement that "the junction supported interactions between CD68/CD14+ transitioning monocytes/macrophages, CD68+ macrophages and MPO+ neutrophils, highlighting the chronic peri-junctional immunosuppressive niche impact, which validated manual thresholding approaches" does not make sense. How does the presence of these myeloid cells in proximity show an immunosuppressive niche? Did these cells express higher levels of TGFb, IL-10, IDO, etc.? Also, the comment on about validating manual thresholding does not make sense.

- We thank the reviewer for pointing out this confusing comment. Since we found PD-L1 expression in the JKs, we wanted to convey that the PD-L1/PD-1 axis may be immunosuppressive. We removed the text regarding the chronic immunosuppression and revised the text to "through PD-L1/PD-1 interactions" but recognize this alone does not imply immunosuppression. We have also clarified what is meant by and manual thresholding, which is currently used in the field to determine cells, similar to a flow cytometry experiment.

7. On line 179 the authors say they used a "pseudobulk RNAseq analysis" however I cant find reference for this in the Methods section. In addition, based on this description of "pseudobulk RNAseq analysis", it is unclear how the authors are making comparisons in DEG in specific cell populations.

- We agree. We described this in the "In DEG Analysis using Cellenics and g:Profiler" section, which we renamed to reflect where this information can be found, including a description of using Cellenics to group cells of varying resolutions for DEG analyses available in the software.

8. The transitions between RNAseq to CODEX on page 9 is abrupt with no lead in or background explanation. The authors should consider a better transition.

- We agree. The order and flow have been changed to emphasize the immune microenvironments at the end.

9. Throughout the manuscript, the authors jump from spatial imaging approaches, to RNAseq, back to spatial imaging, from human tissue to mouse models, etc., which is quite confusing. It is often not clear which data set the authors are describing in the results section, and it is not clarified when you read the figure legends either. More care should be taken to clearly describe the various data sets.

- We thank the reviewer for drawing attention to this, which correlates with comment 1. We tried to clarify this where possible by rearranging data and focusing the story in the order of 1) atlas, 2) keratinocytes, 3) impacts on keratinocytes from microbes, 4) signaling from keratinocytes to immune cells, and 5) peri-epithelial immune microenvironments.

10. On page 13, lines 277-280, it seems the changes in the cell junction proteins and various cell receptors could be a function of mucosal barrier defects, which would lead to microbial translocation and the associated inflammatory responses described. In fact, the 16S rRNA and specific bacteria in situ hybridization data would support this, however, this is not mentioned in the manuscript.

- We agree. The text now reads “supporting a coordinated response, likely in conjunction with mucosal barrier defects.” Throughout the text, we have added the concept of barrier defects to discussion of findings.

11. On line 284, it is important not to over interpret the innate plots in Fig 3. These plots show putative interactions and potential signaling, but without confirmatory direct spatial data, these plots are supportive at best, not definitive proof. Similarly, on line 328, the data do not show "residency" of pathogens, just localization of these bacteria on or in these cells.

- We thank the reviewer for this insightful comment. Indeed, these plots show potential signaling. We added the text throughout about these predictions, and how they may correlate with the in situ multiplex protein data. Regarding line 328, we also agree the data do not definitively show pathogen residency. We point the reviewer to the custom-generated bacterial database, where bacterial RNA was captured with human RNA – either in or on cells.

12. In the Methods section, under PhenoCycler-Fusion, lines 1213-1219, it is confusing what the composition is for staining buffer, antibody blocking solution and blocking solutions. N, J, G and S blockers needs to be explained. I recommend distinguishing the "antibody blocking solution" from the "blocking solution" and perhaps rename the antibody blocking solution to antibody diluent.

- We thank the reviewer for pointing this out. The blockers are proprietary solutions and are used as received from Akoya. We changed antibody blocking solution to antibody diluent.

13. In the Methods section, under Spatial Proteomic cell Assignment and Analysis, please explain how threshold was determined? Was it based on isotype control Ab staining or some other method? Threshold will change based on Ab concentration, secondary Ab concentration and fluorophore, exposure time and laser power and gains. More detail is needed on how threshold was set. Also, please explain why threshold changed between slides for the same markers. On lines 1341-1343, how were cell types assigned from spatial transcriptomic (multiplex RNAscope) stained slides?

- We thank the reviewer for this attention to detail. We clarified the text as follows: “The determination of marker positivity in each cell was based on comparing its fluorescence intensity to predefined thresholds. We carefully selected thresholds for each marker to ensure accurate cell type classification. A marker was positive if its fluorescence intensity exceeded its threshold and negative if it did not.” For lines 1341-1343, we clarified as follows: “The identity of each cluster was then determined by inspecting highly expressed markers, which we determined by the size and intensity of dots in the output matrix and comparing with our DEG lists.”

14. In the Methods section, under Quantification and plotting of in situ hybridization, on line 1382 please explain the threshold method and criterion used.

• We clarified the text as follows: "Following the nuclei-based cell segmentation, subcellular analysis was performed using QuPath, where the number of RNA spots within each cell was detected based on the fluorescence intensity per spot. An individual spot was positive if its fluorescence intensity exceeded its threshold and negative if it did not."

15. Figure 4a, the presented 16s rRNA ISH data is not sufficient to determine if the authors' claims are correct. First, there are not negative controls presented. What is the false detection rate? Second, providing the cell segmentation in the absence of nuclear staining and membrane staining markers is insufficient. Third, the authors do not provide imaging in 3-D (x/y/z) to demonstrate that the signal is all intracellular. Forth, a low-to-high mag panel would be highly informative in interpreting these results.

• Thanks for this suggestion; we have pulled back some of the conclusions of this submission and feel that it is important to prove intracellular pathogenesis in multiple orthogonal assays. Here, we suggest bacterial interactions are what we are observing. To address these comments, firstly, we have assessed the negative control probes which show little signal. We think these do not fully allow for the false detection rate as requested by the reviewer because they are not scrambled sequences for each probe as would typically be used for siRNA or other targeting assay. This 16S probe is widely used and, in our study, it appears specific, including colocalizing in cells that also express periopathogen mRNAs. We don't believe that every signal is truly specific as cross reactivity occurs with qPCR and ISH probes. We think the signals here demonstrate an appropriate level of specificity. Cell segmentation was done using DAPI/nuclei but was not included in the image. The field has been working on membranous segmentation for years and has not solved this issue yet. In the bacterial figure (now Figure 5), in the original submission and here in Figure 5, xyz images are taken of these 16S and mRNA signals. Throughout Figure 5 and Figure 6, there are examples of low and high-mag images of the signal.

16. Line 114, please define what "Tier 1 resolution" means.

We clarified the text as follows: Tier 1 resolution (epithelial, stromal, endothelial, neural, and immune).

Reviewer #2 (Remarks to the Author):

This study seeks to provide a robust understanding of the microbe-host interface relevant to periodontitis. As the subgingival and supragingival biofilms intimately interact with the gingival epithelium, this study seeks to build an atlas of the epithelial cells that sense and respond to changes in the microbiome and environment surrounding the gingiva. Major findings of this study include the identification and localization of unique keratinocyte cell populations, with the identification that JK and SK populations represent unique immune foci. The characterization of intracellular polymicrobial conditions is exciting, and the observation of multiple species co-infecting cells is a valuable new insight. The abundance calculations of bacteria between health and disease are routinely published, and those presented here are limited due to the small sample sizes analyzed. However, as this is not a significant focus of the study, this reviewer doesn't see it as a major limitation as the overall conclusions are well supported. Overall, this is a very well-designed that, despite the density of data and information, is friendly to the reader. The results are impactful, and the conclusions are well justified.

• We are grateful for the reviewer's enthusiasm for our findings and for the well-justified conclusions we provided in the manuscript to identify the localization of unique keratinocyte cell populations, unique immune foci, and polybacterial interactions. We also appreciate the comment that despite the density of data and information, this is friendly to the reader.

1. Line 109-110: I know the information is presented in Supp. Table 1, but it would be beneficial to elaborate here on how many samples represent each condition (i.e., healthy, gingivitis, periodontitis).

• We agree. We revised the text as follows: '3 states [20 health, 4 gingivitis, 10 periodontitis];'.

2. Line 386: References Fig 8f. Clarify if this is extended data 8f?

• We fixed this to refer to current figures.

Reviewer #3 (Remarks to the Author):

Easter, Byrd and colleagues have performed a landmark study in the field of periodontology. I congratulate them for their fantastic work. Using previously published datasets, the authors have harmonized the metadata (an essential and difficult feat in-and-of itself), and developed a comprehensive (1st version) meta-atlas for the human periodontium. Building from this dataset, and in combination with refined spatial analysis (multiplexed immunofluorescence and in situ-hybridization) the authors refine and further resolve the cellular heterogeneity in the periodontal space. Authors introduce the concept of 'epi-kines' or 'keratokines', and evaluate and map how certain epithelial subtypes are skewed toward pro-inflammatory phenotypes, are targeted by multiple intracellular bacteria at the same time, and respond to this bacterial challenge. Finally, I also wish to commend the authors for their extensive methodological section. Below, I include some general remarks and concerns, methodological concerns/questions, as well as line-by-line and panel-by-panel comments, queries and corrections.

We are very grateful for these kind remarks and overall enthusiasm from the reviewer. We are humbled that this reviewer found our work to be a landmark study in periodontology. This is the exact kind of in-depth review that improves manuscripts. We recognize that every initial submission is not perfect, yet the comments from this reviewer have resulted in a dramatic improvement to our manuscript. We want to recognize this reviewer for their collaborative effort with us in their comments to improve the manuscript.

General remarks and concerns:

1. Regarding prior scRNA-seq analysis of periodontal (and dental tissues) not all relevant studies were included in the manuscript (e.g. lines 106-107; 'first' on line 42). Takahashi et al. (2019; PMID: 30509999) and Hemeryck et al. (2022; PMID: 35217915) studied mouse and human dental follicle respectively (preceding the periodontal ligament; admittedly not the most relevant for this study – but should be included in the summary here). Zhao et al. (2021; PMID: 33038290) studied mouse periodontal ligament, including cementoblasts. And although this study is indeed the first meta-atlas focused on the periodontium, including non-dental epithelium (DE) derived lineages, Hermans et al. (2022; PMID: 36299483) developed the first meta-atlas of odontogenic tissue-derived lineages; including analysis of junctional epithelium (JE).

• We thank the reviewer for directing our attention to these studies. As discussed in the text, our draft of the periodontal atlas is V1, and more studies such as these will be added to future drafts. We now mention the additional studies will be integrated as part of future efforts in the Discussion.

2. Regarding the discussion of the different subtypes of epithelial cells, I feel there is some discussion lacking on their respective developmental origins. Lineage tracing in mouse has shown that JE is

derived from the odontogenic DE lineage (Yajima-Himuro et al. 2014, PMID: 24785116; Yuan et al. 2021, PMID: 32985318; and Tanaka et al. 2021, PMID: 34552180). Perhaps this can explain some of the differences seen between JK and SK (e.g. marker profile; JK express many enamel matrix protein related genes/proteins, e.g. ODAM, FDCSP, ODAPH – as also shown in Hermans et al. (2022, PMID: 36299483 and 2023: 37084723). Interestingly, another study reported that JE might be eventually overgrown by oral gingival epithelium, replacing all odontogenic DE-derived JE (Kato et al., 2019; PMID: 37084723) – I am curious what the authors think about this, and if their data may support these findings or not.

• We are grateful for the reviewer's suggestions to consider the developmental origins of the epithelial cells. Indeed, in line with the previous comment on odontogenesis, we expect that JK cells in particular may have originated from odontogenic lineages, especially given their markers, and acknowledge this in the text. Their maintenance after disease, surgery, and in aging is still to be investigated in future work. In the case of the JE being overgrown by gingival epithelium, we are intrigued by this possibility and agree that it may be possible. This has been acknowledged with the limitations of space in the discussion.

3. Although figures are highly complex, and well designed, in many instances essential information is missing from the figure, and not included in the legends. In fact, the figure legends (admittedly very well written!) are formulated in such a way to tell the reader how to interpret the figure. In many panels it is not really described in the figure legend what we are looking at, merely how we should interpret it. I do not think this is good practice. This is something that should be thoroughly revised and double-checked for all figures and panels. Below I include some examples of missing information as well:

- **Fig 1g: no legend included for dot size and color scale...**
- **Fig 2d: what exactly are we looking at? Unclear.**
- **Fig 2k: legend does not help explain what is shown here at all. What do values of +200 and -200 mean? Arbitrary?**
- **Ext Fig 3c-e: no information on color code, dot sizes, etc.**

• These are great suggestions and have been addressed in figures, the text, and legends.

4. From lines 173-240 the entire content is placed in extended figures 2-5. Perhaps a redesign is needed; this corresponds to half the manuscript up to this point. If these concepts are so essential to the story, perhaps they can also be included in main figures..?

• The original paper was designed for the six-figure maximum at another *Nature*-branded journal; this manuscript was directly transferred without reformatting. We agree that a redesign is needed. Extended Data/Figures have been converted into main text figures where possible to support the story.

5. In lines 294-296 the authors conclude that because basal JK have limited interactions, they might be more affected in periodontitis? I feel that this could use more clarification for readers not as versed in immunology – why would they have less interactions with immune cells if they are more affected? The next sections nicely show that they are indeed more affected by intracellular pathogens, which supports these statements – however they also show more expression of 'epikines', so I would expect more interactions to be detected? How can this 'paradox' be explained? Perhaps these are novel signaling interactions (e.g. previously unidentified L-R combinations?) that are not included in the used databases?

• We thank the reviewer for their insight here and their comment on the "paradox". While the interactions appear more limited in basal JKs, this is independent of the interaction significance. Indeed, we see more peri-JK innate immune cells (and epikines/keratokines signaled by JKs), which could be a preferred response to polybacterial

interactions yet not intracellularly. We think this also could be due to comparison of baseline inflamed JKs in “healthy” tissues due to biofilms/subgingival plaque more often to be affecting these cell types. We revised these lines as follows: Overall, basal JK interactions decreased, suggesting these cells are especially affected in periodontitis; however, why these keratinocytes might be affected—and the significance of the remaining interactions—over nearby cells remained unclear.

6. Authors use primary human gingival keratinocytes (HGKs) for in vitro studies, and detect intracellular bacteria present for many passages. To me, this is a tremendous finding. Would this also be the case in oral epithelial organoids? E.g. from gingiva or tongue? Identifying intracellular pathogens in organoid models would be an incredible finding!

• We appreciate the reviewer’s enthusiasm for this tremendous finding. Developing oral epithelial organoids is very difficult, especially because of the inherent heterogeneity in the gingival keratinocytes as we describe, which likely would result in heterogeneous organoids as well. We think the importance of this finding in 2D and 3D models warrants more investigation using super-resolution microscopy, TEM, or direct 16s/metagenomic sequencing of these cell types, etc. so we have removed the in vitro studies from this manuscript and are currently doing the hard work to characterize this phenomenon in numerous ways.

7. Can some close-up examples of cell-cell interactions (eg Ext Fig 7) be visualized using (m)IF or (m)ISH? It would be beautiful, and highly supportive of the (bioinformatics) data, to show such cell-cell interactions in situ; e.g. of JK with immune cells...

• We don’t have specific cell-cell interactions for these but think soon assays like the 500-5000plex ISH experiments (Xenium, MERSCOPE, CosMx) will allow for validating these interactions at scale.

8. In Extended Figure 8d-f the response seems stronger in GM and AG cells than in JG and SG? Doesn’t this contradict previous data..?

• We understand the reviewer’s concern. Rather than showing a regional strong response, this data instead shows polybacterial interaction burden drives expression of keratokines. These are downsampled to show this effect and has been updated in the text. The keratinocytes with the highest bacterial burden are also consistently the cells with the highest keratokine expression, independent of their location in the gingiva.

9. The sections from lines 375-383 and 384-394 are really unclear. The text and figure 6e are confusing. How are the authors making conclusions about stem cells and differentiated progeny? No stem cell markers are used in Figure 6f. It is also not clear what exactly we are looking at. I would recommend to revise this section.

• We thank the reviewer for their concern in these sections. We are making this conclusion because we see the polybacterial interaction phenotype in the basal epithelial cells (progenitors), which give rise to differentiated cells that form the suprabasal layer. We revised the text to reflect this tissue phenomenon.

10. The authors describe increased presence of periopathogens as a consequence of body aging? But is the concept of accumulated exposure the same as aging? This would also indicate overall less presence of intracellular bacteria in basal layers of periodontium in children? Do authors see and correlations with the age of the sample donors?

• The reviewer raises a good point about aging and increased periopathogen presence, and we are currently unclear on this point. We revised the text to “body-wide phenomenon associated with exposures over the lifespan”.

11. scRNA-seq analysis and establishment of meta-atlas: as I understand from the methods, authors employed a ‘one size fits all’ approach to quality control (QC) of these datasets (line 968)? Is this really suitable? Datasets of different origins will often have widely different ranges regarding QC parameters. Some additional plots of the QC parameters seem warranted. I would also recommend to include additional plots with metadata, also for subclustered data. Certain cell types should not be recovered from certain anatomical positions (e.g. JE from palatal region (Ext Fig 1a seems to show some samples from this region?)).

• We agree that a ‘one size fits all’ approach to QC can be difficult. While we acknowledge different datasets will have different ranges, we kept our approach consistent regardless of dataset to account for these differences. Regarding Ext Fig 1a, there was no palatal information – it possibly appears this way because of the line.

12. Some of the dot plots presented (E.g. Ext Fig 4b) seem to have a lot of background/overlap between annotated cell types. Is this biological or due to background/ambient RNA expression? This can be easily checked using the SoupX package (Young et al., 2020, PMID: 33367645) (or CellBender as performed for mouse scRNA-seq data). Performing this analysis can help clean up the overall picture.

• We appreciate the reviewer’s attention to detail here. This “background/overlap” is a biological phenomenon, The host/microbe interactions, as well as cell location, introduce additional complexity, because the polybacterial interaction phenomenon may drive keratokine expression either independent of or linked to identity markers. In our current version 2 data and integration of oral sites with the rest of the body, further parameters are being discussed for better integration approaches generally, especially considering single nucleus seq data as well as other modalities (Parse tech, sci-RNA-seq, etc.).

13. Why was a different analysis pipeline used for the mouse data? Couldn’t this also be analyzed using the same software tools?

• The reviewer is right that a different analysis pipeline was used here. This data was provided by a collaborator and was included as received. The goal of this data was not to harmonize across analysis pipelines but rather to observe commonality between human and mouse scRNAseq data.

14. Have authors also performed cell cycle regression? To remove this effect? Perhaps this could further improve clustering of Fig 3b and Ext Fig 4a?

• We removed cell cycles from the clusters to remove this effect. Despite doing so, we still observed these patterns (CCNB1+).

15. How was the mIF panel informed by the meta-atlas (line 139; Extended data 1c,d = tier 3 and 4 markers)? Conceptually this is a very interesting point, that I feel could be discussed more. Which considerations were taken to select the markers employed? Adding to this, it would be beneficial if in the tables (also for mISH panel) the cell type labelled by the marker could be added – this is not always clear.

• We agree. The markers we used were available from Akoya and were purchased to support a robust characterization of the immune and stromal tissues related to SK and JKs. This has been updated in the text.

16. Regarding collection of tissues for histology and mISH studies, I am concerned about the methodology employed to remove tissues attached to the enamel surface (lines 1131-1135). Authors don’t really describe how tissues were removed. How can we be sure that those tissues closest to the

enamel surface (JE/JK) are not damaged by the process, and are as such accurately represented in this study? Do some remain attached to the enamel surface? These cells are very tightly attached to the tooth to form a microbiological barrier.

• The reviewer is correct about the collection of tissues and potential loss of cells. Because sample collection is not perfect, some cells may be left behind during collection. We unfortunately are unable to be certain that these tissues are not damaged and that no or few cells were left behind. However, we would like to point out that JE/JK (and SE/SK) were represented in health and disease despite low sample numbers. Further, our mIHC and mIHC analyses pinpointed at least some of these cells, which correlated with our scRNAseq data.

17. Line 254: 'pseudotime analysis confirmed differentiation': I do not agree that pseudotime analysis alone is sufficient to confirm differentiation. What about cell plasticity that is niche/migration dependent..?

• We agree that pseudotime analysis alone would not answer this question. From the scRNAseq data, we find differentiated keratinocyte subpopulations trend toward effects and have changed the text.

18. The authors use the proportion of bacterial reads to the total number of human reads to quantify 'bacterial load' per cell. I am wondering how suitable this is? Diseased states will alter the cells' transcriptome, perhaps reducing (or increasing) the amount of total transcripts per cell – this will of course affect the identified proportions (e.g. Fig 4c,d), right?

• We thank the reviewer for this comment. This of course occurs for SARS-CoV-2 as well, where transcription is altered and often downregulated across many genes. This observed phenomenon requires further investigation to understand how to correct for it. For now, we are reporting the atlas and data as observed.

Line-by-line feedback:

19. The presentation of supplementary tables is very confusing, wouldn't it be better to split the Excel sheets into separate, properly labelled tables? Sometimes Extended Data refers to the tables, and sometimes the files; this is confusing as well.

• The submission guidelines state that multi-sheet Excel pages should be combined into a single document, so we tried to do this. We clarified Extended Data table and file references where possible.

20. Line 74: it should be tissue's instead of tissues

• Fixed.

21. Line 131: similar proportions of Ki67 is not clear from figure 1J – this could be due to low resolution of image provided for review? Alternatively, authors could quantify this.

• We appreciate the reviewer's attention to detail. The image compression for peer review resulted in low resolution despite our best efforts. Future work will address epithelial stem cell kinetics from these tissues.

22. Lines 169-170: are GZMB, IFN-g, Galectin-3 etc. signs of the immunosuppressive niche in the junctional zone? (these genes should also be in italics, as it is gene expression)

- We agree. We removed PD-L1 from the list because only that protein, in combination with PD-1, would enable an immunosuppressive axis. We now mention this axis multiple times in the text when we refer to immunosuppression.

23. Line 186 and extended figure 2: not all of the genes described here are shown in the figure. E.g. CXCL3, CCL20 and IL36G can't be found...

- We thank the reviewer for this clarification request. We have rearranged the text and we direct the reviewer to CELLxGENE website for examining this data.

24. Line 193: 'in concert with stromal populations'? What do authors refer to here? This is ambiguous... The other forthcoming work referred to in lines 435-436? No data is shown related to this claim in this report.

- We agree. We revised the text to state "This phenomenon, which likely occurs in concert with stromal populations, may help reshape the periodontal niche through immune recruitment."

25. Lines 197-199: this phrasing is confusing, making epithelial-xxx crosstalk sound much more generalized/abundant than Ext Fig 3c shows. It is JK/SK with the majority of interactions – which is than more accurately reported in line 200. +

- We agree. We revised the text as follows: "We found expected fibroblast-vasculature and fibroblast-immune interactions. As predicted, we also discovered many interactions between SK/JKs and many other cell types, suggesting regional epithelia-stromal-immune communication axes in periodontitis."

26. Lines 213; 218: referred to panels of both Ext Fig 4 and 5. Perhaps redesigning the figures (see previous comment as well) is warranted? This is annoying and confusing for the reader.

- We thank the reviewer for the comment for clarification. We have redesigned figures as mentioned in an earlier comment by the reviewer.

27. Lines 232-233: Did authors use Krt19 antibodies and antigen retrieval methods to be absolutely sure? This could also be included in the methods. This is useful for other researchers in the field. So, how sure can we be of this? And what could be the functional reason behind this suggested disconnect between transcription/translation here? Perhaps the authors can use ISH for Krt19 to confirm specificity of the transcripts for mouse JE at least.

- The Krt19 antibody we used reacts with both human and mouse, and we used the same antigen retrieval method for all IHC to be confident in our conclusions. While the functional reason remains unclear, it may be that the lack of protein transcription may be unique to mouse, and other model species would be better for periodontitis studies. Indeed, forced expression (see PMID 31809739) is needed to definitively detect Krt19 in mice. We have removed the protein data from the manuscript and will need to follow up with multiple antibodies to be sure that the mRNA is not transcribed.

28. Line 236: it should be Il1b instead of il1b

- Fixed.

29. Line 245: '17 keratinocyte populations': phrasing is confusing; because there are other keratinocyte populations (KRT19-) not included in this analysis.

- We agree. We revised this to “subpopulations”.

30. Lines 257-258: I don't see it like this; SPRR2A also seems to decrease in SK?? +

- We agree. We revised the text to highlight clear pathways as follows: i.e., JUND increase in both, yet TGM decrease in JKs, SPRR3 decrease in JKs and SKs, and SPINK7 decrease in SKs.

31. Line 279s and 574: different pathways discussed in text and legend? Why?

- We harmonized lines 279 and 574 by bringing the information in line 574 to line 279.

32. Lines 284-288: cDC1cs is not defined in the text; abbreviations added for some, but not all cell types (e.g. NKT, but not gdT?).

- We agree. We defined cDC1s and added abbreviations where necessary.

33. Line 291: Also CD44 according to plot in Ext Fig 7a?

- We agree and added this to the text.

34. Line 292: What about NKT and gdT? Those also seem strong in the plots..?

- We agree and added this to the text.

35. Line 301: there is an excessive point before 'infection profiles'.

- Fixed.

36. Line 319: does this mean 0.5-2% of all cells in the meta-atlas have intracellular bacteria? What does the range refer to? Different cell types? Datasets? Unclear.

- We agree. We revised the text to “Using healthy and diseased keratinocytes, we gave each keratinocyte a barcode and first looked at the average number of cells harboring at least one bacterial read per barcode, finding enrichment of key periopathogens in 0.5-2% of all barcodes”.

37. Line 360 and Fig 6a: IL1A and CXCL1 were added to the keratokine panel; but why? (and they are not mentioned in the text)

- We mentioned previously that CXCL1 and IL1A had large differences in cell signaling changes and were able to add them to the panel for this experiment. We now mention them in the text.

38. Lines 369, 371, 374, 386: wrong figure reference; should be '6' instead of '8'?

- Fixed.

39. Lines 406 and 407: phrasing?

- We revised as follows: Our study highlights the lack of precision periodontal medicine advances may partially arise from an incomplete pathophysiological complexity understanding at a single-cell and spatial biological level.

40. Lines 433-434: I agree, but I think the authors can include more references to this 'historic knowledge', and relate their findings more.

- We wanted to include many references, but we have already reached the max number of references. We think a review article on this topic would be a great way to address this.

41. Lines 449-450: is these really an accurate reflection of the data? JK and SK also seem to signal to many other cell types. Rephrase?

- We agree. We revised to "While these new cell types signaled many cell types, JKs predominantly signaled macrophages and neutrophils (*CXCL1*, *CXCL3*, *CXCL8*), whereas SKs predominantly signaled T/NK and B-cells (*CCL20*, *CCL28*), correlating with the polybacterial intracellular coinfection phenotype and likely with structural immunity correlations with peri-junctional and -sulcular stromal foci and peri-vascular microniches, which warrants further investigation."

42. Line 459: 'this study': which study do authors refer to? There are 4 references in this sentence.

- We agree and revised to "other studies found" to clarify.

43. Line 472: replace approached with 'anticipated'?

- Replaced.

44. Line 495: it should be (f-g) instead of (f-h).

- Fixed.

45. Lines 501; 1150: it should be immunofluorescence (IF) instead of IHC.

- Fixed.

46. Line 549: in situ should be in italics; double-check for other instances (e.g. line 583)

- Fixed.

47. Line 552: in legend SK and JK; in figure 3a JE and SE used. Make these annotations consistently?

- We thank the reviewer for this clarification comment. SKs and JKs comprise the SE and JE, respectively. We have changed this in the text and figures.

48. Line 639: (e) should be (d)

- Fixed.

49. Line 641: (e) is missing

- Fixed.

50. Line 646: the description for (f) is unclear.

- We agree and revised this.

51. Line 872: ‘as in Figure 1’: unclear what is meant here.

- We agree and revised this.

52. Line 880: unclear why authors refer to figure 5 here?

- We fixed this.

53. Line 904 ‘PAGA broken out’: unscientific language

- We agree and revised to “Expanded PAGA plots”.

54. Line 1134: NBF abbreviation should be explained

- Defined.

55. Line 1143: AR9 buffer should be explained

- We revised to state that this is a pH 9 buffer available from Akoya Biosciences.

56. I would change the order of mISH and mIF in the methods – as of now mIF is positioned last, whereas it is employed first in the text. I had almost missed it.

- We agree and changed the order.

Panel-by-panel feedback:

57. Fig 1e: Why are Langerhans cells and melanocytes included in the epithelial cell types (line 119)? Neither is derived from epithelium (as Merkel cells)?

- We understand the reviewer’s concern. We classified these in the “other” epithelial cell types because these cells are epithelial resident, rather than being derived from it.

58. Fig. 1g: gene names are too small and close together.

- We agree and made font sizes as large as we can. The marker genes are also in Supplementary Table 1.

59. Fig 2c: Some colors are really hard to distinguish; the panel from the supplementary with info on the markers and which cell type they label would be useful here – or if some kind of annotation be made.

- We have labeled colors/antibody to give an indication of cell types. Individual markers are in Supplementary Table 1.

60. Fig 2e: Why are JK/SK not included here? This should be done.

- We understand the reviewer's concern for not including JK/SK. The mIF panel we used does not have any markers for JK/SK specifically; the KRT19 ab used to detect these cells by standard IF is not commercially available through Akoya. We kept the orientation of the tissue the same and used sequential sections to detect KRT19 in the periodontitis samples.

61. Fig 2f: sometimes there is no space between CD38 and PD-L1 in the figure labels. A table with numerical values could be interesting to many readers. Also very unclear to see 10x, 2x, 3x and 100x increases as described on lines 154-156 without this information.

- We agree and added a space between CD38 and PD-L1. Due to space limitations, we could not include a table, but we designed the bar charts in such a way to reflect the small differences in information. Based on this design, the 100x increase in PD-L1 is evident to the reader, highlighting the differences in PD-L1 relative to other markers.

62. Fig 2h and i: could be merged into one panel? They are basically one panel already given the 'periodontitis' line below combining them.

- We agree and have condensed the figures.

63. Fig 2i: I think this is an 'l' instead of 'i'

- Fixed.

64. Fig 2i: really hard to identify the colors belonging to each cell on the plots; very blurry.

- Due to space limitations, we could not make these larger. We will ensure the publication figure is clear.

65. Fig 2j: should be 'j' instead of 'i'

- Fixed.

66. Fig 2l: JK SK are not always positioned the same over the dots in the plot.

- Fixed.

67. Fig 3d: unclear what the percentages relate to? E.g. 43.9%? of what?

- The percentages signify the number of either shared or uniquely upregulated genes. For example, 43.9% signifies that from the analysis we performed, 43.9% of the genes in question were uniquely upregulated by JKs, which we then further investigated by basal & suprabasal.

68. Fig 3f-i: the up/down are hard to read, and not added for each panel. This makes it confusing to interpret; perhaps a different color scale can be used to indicate downregulation more clearly?

- We thank the reviewer for requesting clarification on these panels. We added up/down so that it is consistent within the panels.

69. Fig 3k: TGF-B: which TGF-B gene? There are multiple.

- These are related to pathways and not genes. We did not do a deeper dive into specific genes in this figure unless one specifically stood out. We instead kept our analysis to overall signaling i.e., increase in CCL or decrease in laminin signaling.

70. Fig 3k: What do these values mean? 0-100? Information flow? What does this signify?

- We thank the reviewer for their attention to detail here. The information flow is from keratinocytes to innate and adaptive cells. Blue indicates enriched pathways in health; red indicates enriched pathways in periodontitis. A value of 0% red means the pathway is not enriched in periodontitis; similarly, a value of 100% red means the pathway is highly enriched. We now clarify this in the figure legend.

71. Fig 3j: these plots are almost impossible to untangle and interpret. I don't see some of the interpretation the authors make on lines 281-288. E.g. 'basal cell signaling decreased': I agree this is maybe the case for JK, but this not seem the case for SK? 'In adaptive plots, basal SK decreased': I don't agree? This also seems more or less the same..? The authors conclusion in lines 294-296 matches better with what the figure shows, and not what is claimed in these lines of text.

- We thank the reviewer for requesting clarification on these plots. We revised the text to note that JK basal cell signaling decreased but SK basal cell signaling did not.

72. Fig 3l,m: adding an indication of 'Innate' and 'Adaptive' to the panels themselves would be useful. The text discussing these panels is difficult to understand. Does each panel check the interactions of the 5 different KC populations vs all innate or adaptive immune cell types?

- We added innate and adaptive immune signaling to the figure.

73. Fig 4a: Are these all suprabasal (SB) keratinocytes? Based on lines 294-296 we would also expect more intracellular bacteria to present in basal cells, right? Can the authors quantify 16S+ cells in basal vs suprabasal cells, and include images of both regions?

- We chose to include both basal and SB keratinocytes in this figure. Based on data we collected in the manuscript, we would expect more intracellular bacteria in SB cells because there is more cytoplasm for bacteria to replicate.

74. Fig 4b: add 'SAHMI' on the scheme.

- Added.

75. Fig 4c-e: what does the red font for some bacteria signify? It is not explained anywhere in the legend, and it doesn't seem to correspond to periopathogens? (e.g. *T. vincentii* is not in red, but described as periopathogen on line 598).

- The red font signifies bacteria we investigated by mRNA ISH. We now clarify this in the text.

76. Fig 4e: what is the color code? Fold change? Not labelled.

- We now ensure the color code is included in the figure.

77. Fig 4e: strange output: e.g. Mast cells have low proportions in both panel c,d, but incredibly high fold change in panel e. Is this relevant then?

- The reviewer raises a great point about the relevance of these findings. Indeed, we were surprised at the amount of fold change of some species. Given that some of the percentages of total bacteria could be 0.5-2% as we previously discussed, we do believe these are relevant. To the best of our knowledge, studies have not considered the impact of cell-specific tropism of certain bacteria, so this discovery opens the doors to more questions than answers, which we look forward to exploring in the future.

78. Fig 4g: different colors and cells are almost impossible to discern.

- We agree and note this is due to the resolution of the image uploaded for review, not the publication-quality image.

79. Fig 4f: unclear what this means? Is this the percentage of keratinocytes with intracellular bacteria? Which keratinocytes? The KRT19+ positive ones analyzed in figure 3; or all?

- We thank the reviewer for requesting clarification. This is the percentage of all keratinocytes with polybacterial interaction.

80. Fig 4i,j and lines 334-335: quantify this?

- The quantification of this is complex. Within the samples, there are KRT19+ and KRT19- basal and SB cells all with different levels of polybacterial interaction. Our goal for these panels was to show that not only was there polybacterial coinfection but that in SB cells there were generally more bacterial mRNA signals than in basal cells.

81. Fig 4k: significance is not clear? Why is this relevant? I can't see fluorescence signal in this panel...

- Deleted.

82. Fig 5: Are panels i,j quantifications of images such as in b-h? This isn't really clear to me.

- We thank the reviewer for this clarification question. These panels are quantifying the number of keratokines in a minimum of 100 cells per sample type.

83. Fig 5f: unclear what this panel is? What do arrows signify?

- Fig 5f in the original submission shows the range of keratokines + 16s in LPS-challenged HGKs. We have removed this data to streamline the story.

84. Fig 5j: What if binning is performed differently? Instead of >100, in steps of 100? IL6 doesn't really seem to correlated with 16S load in panel i, although the binning in j does make it look like this... Or perhaps this reaches and early plateau, but doesn't really increase anymore as the 16S load increases..?

- We think that IL6 is not well associated with the direct counts of 16S (see Figure 5b).

85. *Fig 5i: different scale used for bottom row.*

Fixed.

86. *Fig 6d: so both suprabasal and basal layers are summed in these plots? Unclear.*

- Yes, these plots are summed. We now clarify this point in the legend.

87. *Ext Fig 1a: what do these numbers at each reference mean? They don't seem to refer to a specific reference?*

- We thank the reviewer for requesting clarification. The numbers reference the different sites from which the biopsies were acquired for single-cell analysis. In total, biopsies were taken at 9 different sites, hence the numbering scheme.

88. *Ext Fig 1h: I think this is an essential image. Could there also be markers included for Tier 2 epithelial and mesenchymal cell classifications? Can KRT19 be included?*

- We have rearranged the text and figures for clarifying issues like this one.

89. *Ext Fig 2: wouldn't it be more intuitive if the comparison is flipped? So that the positive LFC corresponds to overexpression in periodontitis?*

- We understand the reviewer's concern that the data might be more intuitive if the comparison is flipped and the positive LFC corresponds to overexpression in periodontitis. We designed the figure this way to ensure we adhered to previously published methods in the field, as other groups designed their figures such that negative LFC corresponds to upregulation (or overexpression, as pointed out by the reviewer).

90. *Ext Fig 3c,e: no explanation for color scale*

- We agree. We added the following line to the legend: The most frequent inferred receptor-ligand interactions are in red; the least frequent are in blue.

91. *Ext Fig 3d: no legend for color code or size of dots*

- We appreciate the reviewer's attention to detail. We have added a color legend to the figure, showing the range of colors. We also added the following text to the caption: Larger dots signify more cells upregulating a specific gene. Yellow represents a higher log fold change of cells upregulating the gene compared to other cell types.

92. *Ext Fig 3f: very blurry, I can't read it.*

- We ensured this is clear in the publication-quality image.

93. *Ext Fig 4b: genes are too close together and too small to read.*

- We made the genes further apart and the text larger.

94. *Ext Fig 4c: unclear why some markers are used; and they are not referred to at all anywhere else*

besides this figure. E.g. LAMB4, COL7A1, RHOV. Further, the legend (line 875) mentions KRT15 and LGR6 which I do not see in this figure..?

• Indeed, these markers serve only to pinpoint gene expression heterogeneity, which underlies our claim that there are different cell types. We removed KRT15 and LGR6 because those are in the heatmaps rather than detected in situ.

95. Ext Fig 5f: can we somehow link these clusters (numerical entities only at the moment) to human cell types (or at least some of them)? This would make the study much more complete!

• We agree that linking human to mouse cell types would be exciting. Our next papers will include full atlas cross-species annotation to support future work and modeling.

96. Ext Fig 6d,e: labels are not really that clear/obvious. What is what? Legend is insufficient

• We agree. We expanded the legend as follows: Trajectory analysis of (d) and (e) showed that terminally differentiated SK/JK SB cells were present in health and periodontitis. (f-g) Using CellTypist, we created a Tier 4 UMAP (f) to reflect the new cell types we validated. We further refined the first draft annotation of innate and adaptive immune cell subpopulations (g).

97. Ext Fig 7: add labels for 'innate' and 'adaptive' to make this immediately clear for the reader.

• Done.

98. Ext Fig 8a: increase brightness? Hard to see signal.

• We removed this figure in revision.

99. Ext Fig 8d-f: I would rearrange these panels into 3 rows (e.g. such as f). Will make it easier for the reader.

• We agree and rearranged the panels.

Reviewer #4 (Remarks to the Author):

Tooth function relies on various cell types. While hundreds of diseases can affect teeth, the specific contribution of each cell type to these diseases has not been adequately explored. This study has comprehensively analysed single-cell RNA sequencing (scRNAseq) on tooth-supporting structures, followed by considerable downstream work to identify previously undescribed keratinocytes. Using this approach, the study has, for the first time, created an integrated periodontitis meta-atlas of human tissues through open-source single-cell analysis. The authors validated these cell identities using multiplexed in situ hybridization (mISH). Subsequently, by utilizing highly multiplexed immunofluorescence assays (mIF), they demonstrated the differences in immune foci between the sulcular keratinocyte (SK) and junctional keratinocyte (JK) niches.

We thank the reviewer for their appreciation of our approach and for our work to create a periodontitis meta-atlas and validate the cell types and immune foci.

1. The authors noted that, transcriptionally, SK and JK microniches respond differently in the presence of disease, generally leading to alterations in cell differentiation and the upregulation of effector

cytokines which they referred to as 'epi-kines/keratokines'. This effect is predominantly observed in JKs. Furthermore, the authors predicted that foregoing cells differentially regulate innate and adaptive immune cell subpopulations, even in a healthy state (Line 85-90). The findings and interpretations presented in the preceding sentences lack clarity. Authors should revise these sentences. Moreover, detailed discussion in comparison to existing literature should be included in the discussion section.

- We agree and revised this sentence.

2. As the next step, the authors combined unmapped reads from the periodontal integrated atlas, mIF, and mISH, revealing that some SKs and JKs tolerate intracellular pathogenesis. The analysis identified several distinct intracellular species (Line 91-94). However, it is uncertain whether the authors have conducted a sufficient number of downstream experiments to establish compelling evidence. The authors mentioned that the junctional region consistently showed higher concentrations of innate immune cells, while the sulcular region revealed distinct adaptive immune foci. Is there sufficient evidence from the conducted experiments to support this statement? There should be more downstream work.

- While we appreciate the reviewer's insight into our intracellular pathogenesis finding, we believe our orthogonal methods show compelling evidence for this phenomenon. We found KRT19+ SK and JK cells upregulate proinflammatory cytokines by scRNAseq. We used highly multiplexed IHC to determine adaptive and innate immune foci; highly multiplexed ISH showed that SK and JK upregulate adaptive and innate markers, respectively. Because we don't have multiple orthogonal validations of intracellular pathogenesis, we have tempered language as we suspect some cells have been harboring outer membrane vesicles and others true intracellular pathogens. Methods will need to be developed in situ to prove this phenomenon, hence we are currently terming this polybacterial interaction to address the potential scope.

- The reviewer also raises the question of whether there is sufficient evidence from the conducted experiments to support the statement of innate vs adaptive immune foci. Indeed, with periodontitis not being a singular entity (as raised by the reviewer in another point), we understand the reviewer's concern, and it is difficult to say whether every biopsy would have the same composition of immune cells. However, we would like to point out that another reviewer indicated the composition of the adaptive foci we profiled is consistent with a tertiary lymphoid structure (TLS), meaning compositions would likely be similar across biopsies but differing based on disease severity.

3. Line 150-153 authors stated that "We quantified this by region using single markers and manual thresholding, revealing more infiltrate in peri-epithelial stroma and higher innate peri-junctional immune foci frequency in disease". The message of this sentence is not clear. Please revise.

- We revised this. Great suggestion.

4. Line 157-158, authors stated that "Thus, the periodontal niche may potentially support immunosuppressive microenvironments nearest to the tooth-soft tissue interface, but the reason for this was unknown". It's essential to note that the present study is primarily observation-based, lacking a comprehensive functional analysis of the cell types involved. Therefore, the available evidence might not be sufficient to draw the conclusion that the periodontal niche actively supports immunosuppressive microenvironments.

- We agree. We revised the text to point out that this may be through a PD-L1/PD-1 axis.

5. Line 163 to 168 Authors stated that “Considering local neighborhoods, the sulcus supported more immune-immune predicted “interactions” within cellular neighborhoods, favoring both innate and adaptive immune cell types; however, the junction supported interactions between CD68/CD14+ transitioning monocytes/macrophages, CD68+ macrophages, and MPO+ neutrophils, highlighting the chronic peri-junctional immunosuppressive niche impact, which validated manual thresholding approaches”. Once again, the authors' interpretation remains unclear. What exactly is meant by 'chronic peri-junctional immunosuppressive niche impact'? It is preferable to describe the findings clearly without using technical terms that lack a clear explanation.

• We thank the reviewer for pointing out this confusing comment. Since we found PD-L1 expression in the JKs, we wanted to convey that PD-L1/PD-1 axis may be immunosuppressive. We removed the text regarding the chronic immunosuppression and manual thresholding and revised the text to “through PD-L1/PD-1 interactions” but recognize this alone does not imply immunosuppression.

6. Line 170-171 “Overall, the junctional zone structural niche appeared to be more immunosuppressive than other peri-epithelial niches, but the underlying reason remained unclear”. Similar to previous comments, this statement should make clearer and more detailed, rather than stating “remained unclear”. This point can also be elaborated in the discussion section.

• We agree and revised to state, “Overall, the junctional zone structural niche appeared to be more immunosuppressive through a PD-L1/PD-1 axis than other peri-epithelial niches, but the underlying reason remained unclear”.

7. Line 201-202 What is the reason that SK/JKs are enriched for cytokines and matrix metalloproteinases even in health? The authors suggest active immune roles of these cells (line 206-207). However, this point can be elaborated in the discussion section.

• We agree that this is surprising. In our view, the SK/JKs are constantly exposed to bacteria in the pocket and play a major role in presenting disease pathogenesis. Thus, they are primed to respond to dysbiosis and signal the immune system, in addition to playing their own roles in preventing pathogenesis. We added a brief discussion of this to the manuscript.

8. The authors have proposed that SK/JKs may represent new human cell types. To explore this, they used primary human gingival keratinocytes (HGKs) and discovered the presence and maintenance of KRT19+ cells. Furthermore, through ISH, they identified cell subpopulation-specific markers, such as FDCSP (line 212-222). However, the mere presence of these markers may not be sufficient to definitively establish the existence of new cell types. The authors should consider demonstrating the functional relevance of the proposed new cell type in comparison to other cell types.

• We thank the reviewer for suggesting the cell type's functional relevance. Using the RNAscope (and odontogenic) marker *FDCSP* allowed us to distinguish between tooth- and oral-facing keratinocytes, which correlated with our in situ data in the tissues. We agree that identifying these markers does not definitively establish these as new cell types, but in conjunction with the tissue and single-cell data, this orthogonal confirmation allowed us to pinpoint these cell subpopulations. These cells were also able to be cultured and retained their markers in vitro. More work is needed to validate their function and true identify. The language of “new” cell types has been tempered throughout the manuscript and acknowledges past work in this field.

9. From the murine gingival keratinocyte study explained in lines 226-239, it becomes evident that there are notable differences between the mouse and human junctional niche. Furthermore, the IHC analysis of healthy mouse gingiva around M2 and M3 clearly demonstrated the presence of Krt14 and Ki67 but

not Krt19, suggesting a lack of gene-to-protein translation. This point could be further elaborated in the discussion section.

- We thank the reviewer for pointing this out. More work will be done in the future to understand cross-species annotation of how and why these niches are different.

10. The studies described under 'Periodontitis affects SK/JK stem/progenitor differentiation to upregulate inflammatory signatures' (lines 241-251) need to provide better clinical relevance. It's important to recognize that periodontitis is not a single entity. Therefore, authors should provide clear evidence on the reference.

- We agree that periodontitis is not a single entity. We added clarifying sentences.

11. The polybacterial coinfection of human keratinocytes is an interesting and novel aspect in the field. The authors have done commendable work in this area. However, there are a couple of questions that need clarification, and these points should be elaborated in the discussion section. The authors observed that at least four pathogens can coexist within the same cell, termed 'polybacterial intracellular coinfection phenotype.' Some of these organisms, such as P. aeruginosa and T. denticola, are anaerobic in nature. How do four microorganisms survive in the same human cell? What is the authors' explanation for their survival mechanisms within the host cell? Why have the authors not detected commonly known oral commensals intracellularly?

- We thank the reviewer for this insightful question and for their feedback on the commendable work. Indeed, it was similarly surprising to us that we could detect these intracellular signals. The literature on this topic is complex but suggests that bacteria play both mutualistic and antagonistic roles depending on the scenario. In our view, the intracellular environment is more anaerobic than the extracellular environment, which may in part explain why there are more bacteria. These bacteria are also facultative anaerobes that would thrive in such an environment, bolstering our claim. While these intracellular bacteria would ordinarily have a difficult time surviving within the same cell, we believe external factors result in the cell having an outward-focused rather than inward-focused response, enabling the bacteria to grow. This also may be reflected in intracellular host-microbe interactions of which we have not yet explored – and for which there are only now emerging methods i.e., spatial meta-transcriptomics to profile. Further, these may be many more outer membrane vesicles than microbes. At a single cell and spatial cell resolution, more work is needed to untangle this.

12. Line 405-407 There is an error in this sentence with repeated words. Please correct.

- We agree and removed the repeated word.

13. Overall, the manuscript presents interesting and novel findings. However, the discussion section currently appears to be vague and does not critically analyse the findings. The discussion should focus on specific findings rather than providing an overview of the current status in the field. Therefore, the authors should consider a comprehensive revision of the manuscript.

- We thank the reviewer for recognizing the interesting and novel findings we described. We appreciate the opportunity to use the reviewer's comments to improve the manuscript.

REVIEWERS' COMMENTS

Reviewer #1 (Remarks to the Author):

This is a very comprehensive study, utilizing an array of high-dimensional approaches, including spatial analysis. While this is an incredibly dense and data intensive paper, it is necessary to describe periodontitis. The authors have sufficiently answered my questions and concerns.

Reviewer #3 (Remarks to the Author):

The authors have suitably addressed my concerns and comments in their revised manuscript and rebuttal. The manuscript has been strongly improved. Important elements of structure, readability and contextualization have been enhanced. Given that some elements have been strongly revised, I do have numerous, mostly minor, suggestions, comments and questions for the authors, mostly with regard to design of the figures and missing information for readers.

Comments and suggestions:

- Lines 51, 52-53: *in situ* is not consistently italicized.
- Figure 1C,D: I would recommend to indicate 'Tier 1' and 'Tier 2' in the panels, as done for 'Tier 3' and panel 1E. This will make it immediately clear for the reader.
- Extended Data 1D: the dot size in this figure is smaller than for the other Volcano Plots in this figure. Can this be harmonized?
- Extended Data 1E: there is no x-axis for this panel containing the log fold change.
- Extended Data 1D,E,F: I would label the x-axis for all panels, rather than only panel F. I assume this is log₂ fold change (as correctly labeled in Figure 1H)?
- Line 189: "To validate K19/KRT19 spatial localization": I feel that for a reader this section starts out of the blue. KRT19 as marker for SK/JK has only very briefly been mentioned on line 142. So I am missing some contextualization for the reader here, or perhaps foreshadowing near line 142. In addition, why use K19 instead of KRT19? For all other cytokeratins you used KRT5, KRT14, KRT20... Please harmonize this.
- Extended Data 2A,B: it is unclear from the figure/legend what the color codes indicate? This information should be included somewhere. Can Panel C not be formatted in the same way for consistency too, as it does show the information (Term Name, Term ID, Padj and associated genes; and perhaps -log₁₀ Padj is not necessary)?
- Figure 2F: There are two scale bars in the top left figure? I believe one of these should be for the closeups? This scale should also be indicated in the figure legend too then.
- Extended 3B: I notice that in panel A there seems to be a scale bar over the feature plot of LGR6. I am guessing this is the rogue scale bar that is missing for the closeups?
- Figure 2H,I: what do '+' icons indicate in the figure? This is not mentioned in the figure legend.
- Figure 2j: I don't clearly see blue LGR6 spots in the RNAscope image? Can we really call this "LGR6+ high" cells (as labelled on the figure; but not referenced in the manuscript)? If I zoom in very closely I can see a couple of faint spots.
- Figure 3B: KRT19 should be italicized as it concerns gene expression.
- Figure 3E: All genes should be italicized as it concerns gene expression. Also here: log₂? This panel is very small and hard to read, with much overlap between genes. In addition, the panel is not immediately clear to interpret: there is a disconnect between the title above the figures and the order in which the data is shown in the Volcano plots. For instance, top left, genes upregulated in basal JK are shown on the left-hand side of the plot, whereas in the title Basal JKs are on the right-hand side. Harmonizing the titles to the figures would make it much more clear.
- Figure 3F-I: unclear what the difference is between the left and right-handed heatmaps is? I assume left = healthy, right = periodontitis? This is not mentioned in the figure and/or legend.
- Lines 237-239: the phrasing is very confusing and hard to follow. I recommend rephrasing this sentence for better clarity for the reader. I don't directly see how the next sentence (lines 239-241) relates, this seems like a large jump/overstatement and perhaps better suited to the discussion following further clarification/contextualization.
- I don't understand Extended Figure 5 and the legend. The annotations on the figure (e.g. panels A,B) don't match with the legend. And how do panels D,E relate to this? Are the healthy vs

periodontitis subsets separately clustered again with PAGA?

- Figure 4A: I apologize for not noticing these aspects in the previous round of review – but what do the arrowheads mean? They are only included for healthy tissue, and not mentioned in the legend. In addition, for the periodontitis images, I assume the regions are the same as indicated above the columns? Why is it then included on the figure itself, rather than the directionality (i.e. towards tooth/oral)? Moreover, cells are fully colored in red for periodontitis samples if 5 or more 16S molecules, right? This seems also to be the case for some healthy cells? Why not colored there too then? This also obscures the actual signal in these cells as well...
- Figure 4E: wouldn't it make more sense to show these microbial species in the same order as for panels C,D? I am surprised that *S. maltophilia* does not show any large differences in ratio when (4E) when comparing the differences in panels C,D?
- Lines 282, 387: *in situ* should be italicized.
- Figure 4G: Unclear from the figure/legend what the significance is of the boxed area and the arrows (are these to indicate which region of the image is SB/Basal/stromal cells? – perhaps this can also be done for sulcular epi and gingival margin?).
- Figure 4H: Unclear what the three larger panels indicate? Suprabasal, Basal and Stroma? From Junctional Epi image in 4G? I am confused by the presentation of these panels. Moreover, can you indicate to which cells in these images the close-ups on the left correspond to?
- Figure 4G,H: I am missing a (supplemental) figure with healthy controls to compare with periodontitis-affected tissue shown.
- Figure 5E: I would recommend flipping GM and JK keratinocyte panels, this will better correspond to their location in panel 5D. Perhaps it is an idea to also show separately the segmentation with 16S only; to further cement the finding by this study that keratinocytes with highest bacterial burden are these cells with the consistently highest 'keratokine' expression (also referring to the author's reply in point 8.)
- Figure 5G: not clear from figure/legend what the 'red overlay' over certain cells is in the top panel?
- Lines 320,330: I do not agree with the author's interpretation for IL6: it does not really seem associated with microbial burden given the dot plots in 5B and the analyses presented in 5G-I.
- Line 332: 16S should be italicized.
- Extended Data 6C: I would enlarge this panel, it is too small to read the genes used for annotation.
- Figure 7: not all regions indicated in panels A,B are further included in panel C (or as supplemental). Either they should be included, or they should not be indicated in panels A,B...
- Extended Data 7C: I think you should also refer to this figure in lines 395-398. Before, it is not really clear what is meant with 'cell states'; somewhere in the panel it should also be indicated what cell states these markers correlate with.
- Figure 8G-I: I would harmonize these panels: use the same order for the cell state markers, and combine G,H as done for panel I. This will make it easier to compare health vs periodontitis for the reader.
- Line 443: missing words 'that it', between is/has.
- Line 495: missing 'a' between are/known; and 'of' between cause/multiple?
- Line 498: it should be a ' '. Instead of a ', '?
- Line 502: provide a reference if possible? This sentence is out of the blue.
- Line 630: excessive comma between *gingivalis* and *fimA*?
- Formatting references is inconsistent: abbreviated journal names or not; Capital letters or not etc. I would recommend closely checking these.
- Line 1449: it should be 'in'. In addition, do authors refer to figure 7 here instead of 2?

Reviewer #3 (Remarks on code availability):

I could not access the repository.

Reviewer #4 (Remarks to the Author):

Revision is satisfactory.

April 15th, 2024

Dear Reviewers,

Thank you for providing the reviews of our manuscript, "Single-Cell and Spatially Resolved Interactomics of Tooth-Associated Keratinocytes in Periodontitis". We agree with the important points raised by each of you regarding our work. To address these comments, we have made direct changes to the text and figures. We are confident that this revised manuscript addresses and accounts for all reviewer concerns from the resubmission. Below are the specific revisions we have made to address the comments from all the reviewers.

Reviewer #1 (Remarks to the Author):

This is a very comprehensive study, utilizing an array of high-dimensional approaches, including spatial analysis. While this is an incredibly dense and data intensive paper, it is necessary to describe periodontitis. The authors have sufficiently answered my questions and concerns.

- We thank the reviewer for their kind words regarding the breadth and depth of our study and for their time to review and re-review the manuscript.

Reviewer #3 (Remarks to the Author):

The authors have suitably addressed my concerns and comments in their revised manuscript and rebuttal. The manuscript has been strongly improved. Important elements of structure, readability, and contextualization have been enhanced.

- We are grateful to the reviewer for taking the time to provide such insightful feedback. We appreciate the opportunity to revise our manuscript.

Given that some elements have been strongly revised, I do have numerous, mostly minor, suggestions, comments and questions for the authors, mostly with regard to design of the figures and missing information for readers.

- We thank the reviewer for providing additional suggestions to strengthen the manuscript.

1. Lines 51, 52-53: in situ is not consistently italicized.

- We standardized this here and throughout the manuscript.

2. Figure 1C,D: I would recommend to indicate 'Tier 1' and 'Tier 2' in the panels, as done for 'Tier 3' and panel 1E. This will make it immediately clear for the reader.

- We added Tier 1/2 Resolution to the Figure.

3. Extended Data 1D: the dot size in this figure is smaller than for the other Volcano Plots in this figure. Can this be harmonized?

- Not without extensive edits. We believe this formatting change is unnecessary for supplemental information.

4. Extended Data 1E: there is no x-axis for this panel containing the log fold change.

- We added the x-axis and labeled this panel.

5. Extended Data 1D,E,F: I would label the x-axis for all panels, rather than only panel F. I assume this is log₂ fold change (as correctly labeled in Figure 1H)?

- We ensured all x-axes are labeled.

6. Line 189: “To validate K19/KRT19 spatial localization”: I feel that for a reader this section starts out of the blue. KRT19 as marker for SK/JK has only very briefly been mentioned on line 142. So I am missing some contextualization for the reader here, or perhaps foreshadowing near line 142. In addition, why use K19 instead of KRT19? For all other cytokeratins you used KRT5, KRT14, KRT20... Please harmonize this.

- We harmonized all K19 to KRT19.

7. Extended Data 2A,B: it is unclear from the figure/legend what the color codes indicate? This information should be included somewhere. Can Panel C not be formatted in the same way for consistency too, as it does show the information (Term Name, Term ID, Padj and associated genes; and perhaps -log₁₀ Padj is not necessary)?

- We obtained the charts in this Extended Data from g:Profiler and inserted them as generated. We

8. Figure 2F: There are two scale bars in the top left figure? I believe one of these should be for the closeups? This scale should also be indicated in the figure legend too then.

- We moved the scale bar and now mention the scale in the figure legend.

9. Extended 3B: I notice that in panel A there seems to be a scale bar over the feature plot of LGR6. I am guessing this is the rogue scale bar that is missing for the closeups?

- We removed the rogue scale bar.

10. Figure 2H,I: what do ‘+’ icons indicate in the figure? This is not mentioned in the figure legend.

The + icons indicate KRT19⁺ primary gingival keratinocytes. This is now mentioned in the figure legend.

11. Figure 2j: I don’t clearly see blue LGR6 spots in the RNAscope image? Can we really call this “LGR6+ high” cells (as labelled on the figure; but not referenced in the manuscript)? If I zoom in very closely I can see a couple of faint spots.

- While we understand and agree that there is low *LGR6* signal, especially compared to *FDCSP* in the same cell, not all RNA transcripts are upregulated similarly. In this panel, we note that despite the low expression of *LGR6*, another cell has no transcripts, allowing us to distinguish between “high” and “low”.

12. Figure 3B: *KRT19* should be italicized as it concerns gene expression.

- Fixed.

13. Figure 3E: All genes should be italicized as it concerns gene expression. Also here: log₂? This panel is very small and hard to read, with much overlap between genes. In addition, the panel is not immediately clear to interpret: there is a disconnect between the title above the figures and the order in which the data is shown in the Volcano plots. For instance, top left, genes upregulated in basal JK are shown on the left-hand side of the plot, whereas in the title Basal JKs are on the right-hand side. Harmonizing the titles to the figures would make it much more clear.

- We italicized the genes. Due to space constraints, we tried to make the text as little overlapping as possible. We left Figure 3E as-is because we wanted the reader to immediately focus on upregulated genes by JKs then SKs in health and in periodontitis as we see immunosuppression at and adjacent to the former.

14. Figure 3F-I: unclear what the difference is between the left and right-handed heatmaps is? I assume left = healthy, right = periodontitis? This is not mentioned in the figure and/or legend.

- Yes, left is health and right is periodontitis. This is now clarified in the figure.

15. Lines 237-239: the phrasing is very confusing and hard to follow. I recommend rephrasing this sentence for better clarity for the reader. I don't directly see how the next sentence (lines 239-241) relates, this seems like a large jump/overstatement and perhaps better suited to the discussion following further clarification/contextualization.

- We revised this sentence to "This finding uncovered potential therapeutic avenues for periodontitis niche restoration to health since keratinocytes closest to the tooth surface drove most effector cytokine expression compared to all keratinocytes." This correlates with our manuscript revision to highlight the role of the epithelium in periodontitis pathogenesis.

16. I don't understand Extended Figure 5 and the legend. The annotations on the figure (e.g. panels A,B) don't match with the legend. And how do panels D,E relate to this? Are the healthy vs periodontitis subsets separately clustered again with PAGA?

- Thank you for bringing our attention to Extended Figure 5. The legend prior to the panel description has been revised as follows: "PAGA analysis of *KRT19*-high keratinocytes shows trajectory analysis towards terminally differentiated cells."

17. Figure 4A: I apologize for not noticing these aspects in the previous round of review – but what do the arrowheads mean? They are only included for healthy tissue, and not mentioned in the legend. In addition, for the periodontitis images, I assume the regions are the same as indicated above the columns? Why is it then included on the figure itself, rather than the directionality (i.e. towards tooth/oral)? Moreover, cells are fully colored in red for periodontitis samples if 5 or more 16S molecules, right? This seems also to be the case for some healthy cells? Why not colored there too then? This also obscures the actual signal in these cells as well...

- The arrows are included here to indicate the directionality to the reader; the text in the periodontitis images directly correlates with the health orientation, rendering the need for the arrows irrelevant. We chose to show the

examples of the signal in health, then demonstrated that clearly in the junctional epithelium, we saw more cells with 5 or more 16S rRNA transcripts.

18. Figure 4E: wouldn't it make more sense to show these microbial species in the same order as for panels C,D? I am surprised that *S. maltophilia* does not show any large differences in ratio when (4E) when comparing the differences in panels C,D?

• In this manuscript, we focused on the epithelium, and for this reason, we ordered the ratio change of mapped microbiome reads according to the greatest fold change in the epithelium. There is a ratio change of *S. maltophilia* as seen in Figure 4E but not at the level of other microbes.

19. Lines 282, 387: *in situ* should be italicized.

• Fixed as mentioned in #1.

20. Figure 4G: Unclear from the figure/legend what the significance is of the boxed area and the arrows (are these to indicate which region of the image is SB/Basal/stromal cells? – perhaps this can also be done for sulcular epi and gingival margin?) and 21. Figure 4H: Unclear what the three larger panels indicate? Suprabasal, Basal and Stroma? From Junctional Epi image in 4G? I am confused by the presentation of these panels. Moreover, can you indicate to which cells in these images the close-ups on the left correspond to?

• Figures 4G and 4H show complementary information. In these panels, we reused the same slide from the PhenoCycler Fusion assay by removing the flow cell and hybridizing the probes to detect the bacteria. We showed in these panels that there is more enrichment of known periopathogen *P. gingivalis* and genus *Fusobacterium* in the junctional basal, suprabasal, and stromal cells compared to sulcular and gingival margin (GM) equivalents. Further, this correlated with the innate immune niche in the junction, compared to adaptive (sulcus) and less immune infiltrate (GM).

22. Figure 4G,H: I am missing a (supplemental) figure with healthy controls to compare with periodontitis-affected tissue shown.

• We performed this experiment on the sample indicated because of technical challenges. Despite storing the sample in the refrigerator in Akoya storage buffer, over a month had passed since we performed the PhenoCycler Fusion experiment, with no guarantee of RNA integrity. Future work will explore this in these tissues with more quantitative instead of a qualitative approach. Here, we simply wanted to provide an orthogonal approach and think no further follow up is warranted.

23. Figure 5E: I would recommend flipping GM and JK keratinocyte panels, this will better correspond to their location in panel 5D. Perhaps it is an idea to also show separately the segmentation with 16S only; to further cement the finding by this study that keratinocytes with highest bacterial burden are these cells with the consistently highest 'keratokine' expression (also referring to the author's reply in point 8.)

• We flipped the panel.

24. Figure 5G: not clear from figure/legend what the 'red overlay' over certain cells is in the top panel?

• The red overlay is to show the segmentation region within the cell and to help with visualization of the intracellular 16S signal. We updated this in the figure legend.

25. Lines 320,330: I do not agree with the author's interpretation for IL6: it does not really seem associated with microbial burden given the dot plots in 5B and the analyses presented in 5G-I.

- We removed IL6 from line 320 but kept it in line 330 as some of our data suggested at least some correlation with microbial burden.

26. Line 332: 16S should be italicized.

- Fixed.

27. Extended Data 6C: I would enlarge this panel, it is too small to read the genes used for annotation.

- We tried to increase the font size to make the genes more legible.

28. Figure 7: not all regions indicated in panels A,B are further included in panel C (or as supplemental). Either they should be included, or they should not be indicated in panels A,B...

- We removed some regions we did not discuss from the panels. We kept the AG panel to show a lack of immune recruitment to that site as opposed to other sites with clear immune infiltrate.

29. Extended Data 7C: I think you should also refer to this figure in lines 395-398. Before, it is not really clear what is meant with 'cell states'; somewhere in the panel it should also be indicated what cell states these markers correlate with.

- We revised this to "including cell states and immune checkpoint expression (Extended Data 7c)." We also now mention in the associated caption "All cell states except Ki67 (cycling) relate to immune activation."

30. Figure 8G-I: I would harmonize these panels: use the same order for the cell state markers, and combine G,H as done for panel I. This will make it easier to compare health vs periodontitis for the reader.

- We chose to keep the panel as-is because we wanted to make immediately clear to the reader what peri-junctional and -sulcular inflammatory signatures we identified in health and how the mean value shifted in periodontitis.

31. Line 443: missing words 'that it', between is/has.

- We revised to "The limitation of this study is that it has likely missed as-of-yet undiscovered cell types or states in periodontitis."

32. Line 495: missing 'a' between are/known; and 'of' between cause/multiple?

- Fixed.

33. Line 498: it should be a '.' Instead of a ','?

- Fixed.

34. Line 502: provide a reference if possible? This sentence is out of the blue.

- Removed.

35. Line 630: excessive comma between gingivalis and fimA?

- We revised as follows: (*P. gingivalis*, *fimA*; *Fusobacterium*, *fadA*).

36. Formatting references is inconsistent: abbreviated journal names or not; Capital letters or not etc. I would recommend closely checking these.

- Fixed.

37. Line 1449: it should be 'in'. In addition, do authors refer to figure 7 here instead of 2?

- Fixed.

Reviewer #4 (Remarks to the Author):

Revision is satisfactory.

- We thank the reviewer for their approval of our revision after taking the time to re-read it.